# Towards Complete Multi-Agent Coordination Policy Learning via Denoising Maximum Entropy Optimization

**Guanghao Li** [1]  **Lei Yuan** [1 2]  **Ruiqi Xue** [1]  **Hengchang Zhang** [1]  **Jianhong Wang** [3]  **Yi-Chen Li** [1 2]  **Yang Yu** [1 2]

## Abstract

Parameter sharing is a widely used technique in Multi-Agent Reinforcement Learning (MARL) that enhances sample efficiency by equipping agents with a unified policy. While effective in homogeneous settings, it often struggles in heterogeneous environments where agents possess diverse capabilities. Conversely, learning customized policies for agents can resolve knowledge conflicts but significantly hinders knowledge transfer, thereby reducing learning efficiency. Existing approaches attempt to balance this trade-off using clustering or agent-specific masks, but they typically rely on strong environment-specific priors and struggle in settings where the team exhibits multi-modal policies. To address these limitations, we propose Dspic, an efficient shared-policy algorithm grounded in the maximum entropy framework. Specifically, Dspic employs self-supervised learning to extract discriminative role embeddings for each agent. These embeddings guide a complete division of the observation space, providing a theoretical guarantee for the optimality of parameter sharing. Furthermore, to handle the increased observation complexity and diversity resulting from this division, Dspic incorporates a diffusion policy, enhancing the capacity to model complex action distributions while enabling efficient learning. Extensive experiments on MaMuJoCo, SMAC, SMACv2, and LBF demonstrate that Dspic achieves superior sample efficiency while maintaining asymptotic optimality. The code of our algorithm is available on https://github.com/tlxxx/Dspic_code.

## 1. Introduction

Multi-agent reinforcement learning (MARL) (Hernandez-Leal et al., 2019) has attracted widespread attention via training multi-agent systems in a trial-and-error way, demonstrating great success in multiple scenarios like autonomous driving (Zhang et al., 2024), financial trading (Sarin et al., 2024), and embodied intelligence (Feng et al., 2025). To mitigate the challenges like non-stationary and partially observable environments (Yuan et al., 2023), typical works have adopted the *Centralized Training and Distributed Execution* (CTDE) paradigm (Foerster et al., 2016), including both value-based methods (Sunehag et al., 2017; Rashid et al., 2018; Wang et al., 2021a), policy gradient methods (Lowe et al., 2017; Foerster et al., 2018; Yu et al., 2022), complex policy structure design (Wen et al., 2022), efficient agent exploration (Wang et al., 2020b), etc.

To promote knowledge sharing and accelerate learning, widely used MARL approaches typically train a unified policy for all agents, distinguishing individuals via appended indicators such as one-hot IDs (Lowe et al., 2017; Sunehag et al., 2017; Rashid et al., 2018). While effective in simple scenarios, these homogeneous methods struggle in complex environments composed of agents with unique characteristics (Wooldridge, 2009). To address this, some works have introduced architectural or objective-based differentiations. For instance, ROMA (Wang et al., 2020a) utilizes a shared supernetwork to embed roles into value network parameters, while SePS (Christianos et al., 2021) clusters agents by encoding identities into a latent space to determine policy sharing groups. Furthermore, CDS (Li et al., 2021) fosters diversity through intrinsic rewards, while Kaleidoscope (Li et al., 2024c) achieves specialization by applying distinct masks within a shared network. However, these approaches often struggle in settings where the team exhibits multi-modal policies, or rely on intricate, domain-specific designs, thereby limiting their scalability and practical deployment in resource-constrained or general-purpose settings. These limitations highlight the necessity of Heterogeneous-Agent RL (HARL) (Zhong et al., 2024). In contrast, HASAC extends the maximum entropy (MaxEnt) (Haarnoja et al., 2018) theory to MARL, customizes one policy to each agent and updating policies in a sequential way, showing high adapt-

[1]National Key Laboratory for Novel Software Technology, Nanjing University, China & School of Artificial Intelligence, Nanjing University, China [2]Polixir Technologies [3]INFORMED-AI Hub, University of Bristol, UK. Correspondence to: Lei Yuan <yuanl@lamda.nju.edu.cn>.

*Proceedings of the 43$^{rd}$ International Conference on Machine Learning*, Seoul, South Korea. PMLR 306, 2026. Copyright 2026 by the author(s).

ability, but are limited by decreased sample efficiency due to the lack of knowledge sharing (Liu et al., 2024).

Addressing this limitation, we theoretically identify that the performance bottleneck of shared policies primarily stems from the overlap of observation spaces, meaning that agents are forced to condition their decisions on indistinguishable observations, which leads to suboptimal convergence. To resolve this, we propose that the key solution lies in a *Complete Division* (CD) of the observation space. Specifically, we introduce *Diffusion Soft Policy Iteration with Complete Division* (Dspic), an efficient shared-policy algorithm grounded in the maximum entropy framework. Dspic first leverages a Variational Autoencoder (VAE) to generate discriminative role embeddings, which subsequently guide the learning of role-based division for the observation space. Furthermore, to capture the highly complex and multimodal policies arising from the unified observation space, we incorporate a diffusion policy. This enhances the model's expressivity while maintaining sample efficiency. Theoretically, we prove that Dspic achieves the same asymptotic optimality as HARL methods, despite sharing parameters. Extensive experiments on MaMuJoCo (de Witt et al., 2020), SMAC (Samvelyan et al., 2019), SMACv2 (Ellis et al., 2023), and LBF (Christianos et al., 2020) demonstrate that our algorithm significantly outperforms baselines and verifies the efficacy of our design.

**Conflict of Interest Disclosure.** The authors declare that there is no conflict of interest.

## 2. Related Work

**Multi-agent Reinforcement Learning (MARL).** MARL has witnessed vigorous progress in recent years. The main research paths include multi-agent policy gradients (Lowe et al., 2017; Foerster et al., 2018; Yu et al., 2022; Wen et al., 2022), and value-based MARL (Sunehag et al., 2017; Rashid et al., 2018; Wang et al., 2021a). Notably, policy gradient-based methods often struggle in heterogeneous environments when relying on naive parameter sharing. To address the heterogeneity problem, heterogeneous-agents reinforcement learning (HARL) plays a crucial role. HAPPO/HATRPO (Kuba et al., 2022) provides a sequential update paradigm with monotonic improvement and NE convergence properties. HetGPPO (Bettini et al., 2023) proposes a GNN-based framework for heterogeneous learning. However, these methods risk converging to suboptimal solutions (Liu et al., 2024), which can be solved by introducing MaxEnt reinforcement learning. HASAC (Liu et al., 2024) applies this improvement and uses a sequential update pipeline to improve performance. It also achieves state-of-the-art results in multiple benchmarks.

**Parameter Sharing.** First discussed by Tan (Tan, 1993), parameter sharing has been widely adopted in MARL algo-

rithms due to its simplicity and high sample efficiency (Terry et al., 2023). To balance the flexibility of heterogeneous policies with the advantages of shared policies, some studies focus on partial parameter sharing, such as SePS (Christianos et al., 2021), SNP (Kim & Sung, 2023) and AdaPs (Li et al., 2024a). Furthermore, CDS (Li et al., 2021) encourages agents to learn diverse strategies by adding intrinsic rewards to existing rewards. MADPS (Hu et al., 2024a) introduces a dynamic parameter sharing algorithm that automatically adjusts the sharing scheme based on a novel policy distance metric. In addition, some works consider symmetry of agents (van der Pol et al., 2022), while some works propose methods for providing different network masks for different agents (Li et al., 2024c). The role of agents should also be considered when allocating parameters, for example, ROMA (Wang et al., 2020a) controls the parameters of the value network under different roles through a shared supernetwork, and ACORM (Hu et al., 2024b) enhances policy performance by incorporating contrastive role representations into networks with attention.

**Diffusion Model as Policy.** Diffusion models (Ho et al., 2020; Song et al., 2021) have demonstrated strong competitiveness in modeling policies, thanks to their powerful representation capabilities and learning efficiency. Previous work (Janner et al., 2022; Wang et al., 2023) has demonstrated that diffusion models significantly enhance offline RL. Diffusion models still demonstrate good performance in online scenarios. DIPO (Yang et al., 2023) proposes action gradient to overcome the problem that diffusion policy is difficult to compute gradients within the policy gradient framework. From a different perspective, to promote model exploration, QVPO (Ding et al., 2024) achieves equivalent effects by designing diffusion losses that respectively encourage utilization and exploration, while DIME (Celik et al., 2025) transforms computing entropy into an equivalent optimization of its lower bound. For the same purpose, GenPO (Ding et al., 2025) leverages exact diffusion inversion to construct invertible action mappings for stable on-policy policy optimization. Given its enormous success under single agent tasks, introducing diffusion models into MARL presents a promising avenue. MAD3PG (Zhong et al., 2025) uses diffusion model as the critic to avoid calculating policy gradient when updating actors.

## 3. Preliminaries

### 3.1. Maximum Entropy MARL

We consider a cooperative Markov game (Littman, 1994) formulated by a tuple $\langle \mathcal{I}, \mathcal{S}, \mathcal{A}, r, P, \gamma, d \rangle$. Here, $\mathcal{I} = \{1, \ldots, n\}$ denotes the set of $n$ agents, $\mathcal{S}$ is the finite state space, $\mathcal{A} = \mathcal{A}^1 \times \mathcal{A}^2 \times \ldots \times \mathcal{A}^n$ is the joint action space, where $\mathcal{A}^i$ denotes the finite action space of agent $i$, $r : \mathcal{S} \times \mathcal{A} \to [r_{\min}, r_{\max}]$ is the joint reward function,

$P : \mathcal{S} \times \mathcal{A} \times \mathcal{S} \to [0,1]$ is the transition probability function, $\gamma \in [0,1)$ is the discount factor, and $d$ is initial state distribution. At time step $t \in \{1, \ldots, T\}$, all agents are at states $s_t \in \mathcal{S}$ and take joint actions $\boldsymbol{a}_t \sim \boldsymbol{\pi}(\cdot|s_t)$, where the joint actions can be considered as a concatenation of each agent's action $a_t^i$, i.e., $\boldsymbol{a}_t = (a_t^1, a_t^2, ..., a_t^n)$. The agents receive a joint reward $r_t = r(s_t, \boldsymbol{a}_t)$ and move to the next state $s_{t+1} \sim P(\cdot|s_t, \boldsymbol{a}_t)$. The initial state distribution $d$, the joint policy $\boldsymbol{\pi}$, and the transition kernel $P$ induce a marginal state distribution at time $t$, denoted by $\rho_{\boldsymbol{\pi}}^t$. The goal of maximum entropy MARL is to jointly maximize the sum of expected rewards and entropies of a policy, defined as

$$J(\boldsymbol{\pi}) = \mathbb{E}_{s_{0:T} \sim \rho_{\boldsymbol{\pi}}^{0:T}, \boldsymbol{a}_{0:T} \sim \boldsymbol{\pi}} \left[ \sum_{t=0}^{T} \gamma^t \left( r_t + \alpha \mathcal{H}\left( \boldsymbol{\pi}(\cdot|s_t) \right) \right) \right],$$
(1)

where $\mathcal{H}\left( \boldsymbol{\pi}(\cdot|s) \right) = -\int_{\boldsymbol{a}} \boldsymbol{\pi}(\boldsymbol{a}|s) \log \boldsymbol{\pi}(\boldsymbol{a}|s) \mathrm{d}\boldsymbol{a}$ is the differential entropy and $\alpha > 0$ controls the exploration exploitation trade-off (Haarnoja et al., 2017). To evaluate state-action values under the objective defined in Eqn. 1, we introduce the joint soft $Q$-function as follows:

$$Q_{\boldsymbol{\pi}}(s_t, \boldsymbol{a}_t) = r_t +$$
$$\mathbb{E}_{\boldsymbol{a}_{t+1:T} \sim \boldsymbol{\pi}, s_{t+1:T} \sim P} \left[ \sum_{l=t+1}^{T} \gamma^{l-t} \left( r_l + \alpha \mathcal{H}(\boldsymbol{\pi}(\cdot|s_l)) \right) \right].$$
(2)

To be applicable to heterogeneous scenarios with Max-Ent MARL methods, previous work like HASAC (Liu et al., 2024), trains heterogeneous policy by updating parameters without affecting cross agents, i.e., $\boldsymbol{\pi}(\boldsymbol{a}|s) = \prod_{i=1}^{n} \pi^i(a^i|s)$, where $\pi^i$ denotes the individual policy of agent $i$. These methods show the characteristic of *Quantitative Response Equilibrium* (QRE) (McKelvey & Palfrey, 1995) policy, which guarantees each agent cannot improve the $J(\boldsymbol{\pi})$ by changing its own policy.

To obtain a policy converging to QRE, the HASAC algorithm implements the Heterogeneous-Agent Soft Policy Iteration (HASPI) method, which defines a soft Bellman backup operator $\mathcal{T}^{\boldsymbol{\pi}}$ to update $Q$ with

$$\mathcal{T}^{\boldsymbol{\pi}} Q_{\boldsymbol{\pi}}(s_t, \boldsymbol{a}_t) \triangleq r_t + \gamma \mathbb{E}_{s' \sim P} \left[ V_{\boldsymbol{\pi}}(s') \right], \quad (3)$$
$$V_{\boldsymbol{\pi}}(s) = \mathbb{E}_{\boldsymbol{a} \sim \boldsymbol{\pi}} \left[ Q_{\boldsymbol{\pi}}(s, \boldsymbol{a}) + \alpha \mathcal{H}\left( \boldsymbol{\pi}(\cdot|s) \right) \right], \quad (4)$$

and update joint policy by decomposing it into update of individual policies sequentially. We also provide a detailed description in App. A.1 to avoid confusion.

### 3.2. Diffusion Model as Policy

Denoising diffusion probabilistic models (DDPM) (Ho et al., 2020) are powerful generative models that progressively add noise to data and then recover the original data through a reverse diffusion process. To model the RL policy with DDPM, considering a joint policy modeled by

DDPM, where we denote its forward and reverse processes as $\vec{\boldsymbol{\pi}} \in \vec{\boldsymbol{\Pi}}$ and $\reflectbox{$\vec{\boldsymbol{\pi}}$} \in \reflectbox{$\vec{\boldsymbol{\Pi}}$}$ respectively ($\vec{\boldsymbol{\Pi}}$ and $\reflectbox{$\vec{\boldsymbol{\Pi}}$}$ are their corresponding policy space). In the following derivation, we use $\mathbf{u}_k, k = 0, 1, ..., K$ to represent the $k$-th step in the diffusion process[1], specifically, $\mathbf{u}_0 = \boldsymbol{a}$. Note that the diffusion process of different agents is independent, then we have $\mathbf{u}_k = (u_k^1, u_k^2, ..., u_k^n)$, where $u_k^i$ is $i$-th agent's diffusion latent variable at step $k$. Following the results of DIME (Celik et al., 2025), we can optimize the policy by indirectly maximizing the lower bound of $J(\reflectbox{$\vec{\boldsymbol{\pi}}$})$, i.e.,

$$J(\reflectbox{$\vec{\boldsymbol{\pi}}$}) \geq \hat{J}(\reflectbox{$\vec{\boldsymbol{\pi}}$}) = \mathbb{E}_{s_{0:T}, \boldsymbol{a}_{0:T}} \left[ \sum_{t=0}^{T} \gamma^t \left( r_t + \alpha \ell(\reflectbox{$\vec{\boldsymbol{\pi}}$}(\cdot|s_t)) \right) \right],$$
(5)

where

$$\ell(\reflectbox{$\vec{\boldsymbol{\pi}}$}(\cdot|s)) = \mathbb{E}_{\mathbf{u}_{0:K} \sim \reflectbox{$\vec{\boldsymbol{\pi}}$}_{0:K}} \left[ \log \frac{\vec{\boldsymbol{\pi}}_{1:K|0}(\mathbf{u}_{1:K}|\mathbf{u}_0, s)}{\reflectbox{$\vec{\boldsymbol{\pi}}$}_{0:K}(\mathbf{u}_{0:K}|s)} \right]. \quad (6)$$

Then, we can achieve the goal of maximizing $\hat{J}(\reflectbox{$\vec{\boldsymbol{\pi}}$})$ by

$$\min_{\reflectbox{$\vec{\boldsymbol{\pi}}$} \in \reflectbox{$\vec{\boldsymbol{\Pi}}$}} \mathbb{E}_s \left[ D_{\mathrm{KL}}\left( \reflectbox{$\vec{\boldsymbol{\pi}}$}(\mathbf{u}_{0:K}|s) \| \vec{\boldsymbol{\pi}}(\mathbf{u}_{0:K}|s) \right) \right]. \quad (7)$$

A detailed explanation can be found in App. A.4.

## 4. Method

In this section, we will describe the design of Dspic, a Max-Ent Multi-Agent policy iteration framework with optimality guarantee. Specifically, Sec. 4.1 presents Complete Division (CD) of agents' state space to guarantee optimality, and Sec. 4.2 combines CD with the agents' role within a matrix encoder. To improve the policy's expressiveness and learning efficiency, Sec. 4.3 presents Dspic method based on the diffusion model. Combining all these aspects, we present a practical pipeline in Sec. 4.4.

### 4.1. Optimality Guarantee with Complete Division

First, we will theoretically explore the reasons for the sub-optimality of shared policy to inspire further improvements. For agents with different action spaces, we first align dimensions by zero-padding, denoted as $\mathcal{A}^1 = \cdots = \mathcal{A}^n = \mathcal{A}$. Then we consider the QRE policy under heterogeneous policy and the optimal joint policy under shared policy setting, denoted as $\boldsymbol{\pi}_\star^{\mathrm{HA}}$ and $\boldsymbol{\pi}_\star^{\mathrm{share}}$ respectively. Then, we will explain that the *Homogeneous Setting* is the only case that makes the $\boldsymbol{\pi}_\star^{\mathrm{HA}}$ equal to $\boldsymbol{\pi}_\star^{\mathrm{share}}$.

**Definition 4.1** (**Homogeneous Setting**). A multi-agent system is **homogeneous** if for any joint policy $\boldsymbol{\pi}$, there exists an

---

[1]To distinguish between RL time steps and diffusion steps, the subscript $t$ of notations $a$ represents RL timesteps, while the subscript $k$ of notations $u$ represents diffusion steps.

action-independent bias $b_{\boldsymbol{\pi}}^{ij}(s)$ for any $i \neq j$, such that for $s \in \mathcal{S}, a \in \mathcal{A}, \boldsymbol{a} \in \prod_{k=1}^{n-1} \mathcal{A}$, the joint $Q$-function satisfies:

$$
\begin{aligned}
Q_{\boldsymbol{\pi}}(s, a^i = a, \boldsymbol{a}^{-i} = \boldsymbol{a}) \\
= Q_{\boldsymbol{\pi}}(s, a^j = a, \boldsymbol{a}^{-j} = \boldsymbol{a}) + b_{\boldsymbol{\pi}}^{ij}(s).
\end{aligned}
\tag{8}
$$

We have $D_{\mathrm{KL}}(\boldsymbol{\pi}_{\star}^{\mathrm{HA}}(\cdot|s) \| \boldsymbol{\pi}_{\star}^{\mathrm{share}}(\cdot|s)) > 0$ holds **unless** agents are in a *homogeneous setting* and its proof is in App. G.1. Inspired by the derivation, we believe that such policy deviation actually stems from the overlap in the agents' state space, leading to interference between agents. Previous works attempted to provide identity information for agents by concatenating agent ID, but our experiments in Sec. 5.3 found that it can't achieve the performance of heterogeneous baselines. The failure of these methods is attributed to insufficient identity information solely by concatenation and most dimensions of the state still overlap. To fundamentally address it, we propose *Complete Division* (**CD**) of a space.

**Definition 4.2** (**Complete Division**). Define an injective set denoted by $\{f_i\}_{i=1}^n$ with the same domain $\mathfrak{D}$ and ranges $\{\mathcal{V}_i\}_{i=1}^n$. If for $\forall i \neq j$, $\mathcal{V}_i \cap \mathcal{V}_j = \emptyset$ holds, then we call $\{f_i\}_{i=1}^n$ a Complete Division of $\mathfrak{D}$.

With a CD $\{f_i\}_{i=1}^n$ of $\mathcal{S}$, we can define a joint policy that shares parameters but doesn't share state space, i.e.,

$$
\boldsymbol{\pi}^{\mathrm{CD}}(\boldsymbol{a} \mid s) = \prod_{i=1}^n \pi^{\mathrm{CD}}\left(a^i \mid f_i(s)\right).
\tag{9}
$$

Then we denote the QRE policy under this definition as $\boldsymbol{\pi}_{\star}^{\mathrm{CD}}(\cdot|s)$ with CD $\{f_i\}_{i=1}^n$. With Complete Division satisfied, $\boldsymbol{\pi}_{\star}^{\mathrm{CD}}(\cdot|s)$ does not deviate from $\boldsymbol{\pi}_{\star}^{\mathrm{HA}}$.

**Theorem 4.3** (**Zero Policy Deviation**). *For $\forall s \in \mathcal{S}$, Equation $D_{KL}(\boldsymbol{\pi}_{\star}^{HA}(\cdot|s) \| \boldsymbol{\pi}_{\star}^{CD}(\cdot|s)) = 0$ always holds.*

Its proof is provided in App. G.2. This theorem demonstrates that it is feasible to ensure optimality while sharing parameters by distinguishing the state space of agents.

### 4.2. Role-based Orthogonal Complete Division

For different agents in a team, their role information often has a significant impact on the policy. Thus, the role can provide guidance for constructing CDs over tasks. We use a vector $z_i$ to represent the role of agent $i$. A role embedder based on VAE (Kingma et al., 2019) is applied to obtain it. For a period at the beginning of training, we train it by embedding $i$ into role embeddings $z_i$. Then the decoder attempts to decode $s_{t+1}$ and $r_t$ based on $z_i$, $s_t$, and $a_t^i$. We provide a detailed description in App. B.1. With the role information in $z_i$, we propose the orthogonal construction.

**Proposition 4.4** (**Orthogonal Construction**). *Define a set of matrices $\{\mathbf{P}_i\}_{i=1}^n$, where $\mathbf{P}_i \in \mathbb{R}^{d_s \times d'}$ ($d_s$ is the dimension of the state space $\mathcal{S}$ and $d'$ is the dimension of projection space). If for any $i \neq j$, $\mathbf{P}_i^{\top} \mathbf{P}_j = \mathbf{O}$ holds, then the*

*mapping set $\{f_i(s) = (\mathbf{P}_i s, s)\}_{i=1}^n$ constitutes a Complete Division of space $\mathcal{S} \backslash \{s \mid s \in \mathcal{N}(\mathbf{P}_i^{\top} \mathbf{P}_i) \cap \mathcal{N}(\mathbf{P}_j^{\top} \mathbf{P}_j), \exists i \neq j\}$, where $\mathcal{N}(\mathbf{P}_i^{\top} \mathbf{P}_i) = \{s \mid \mathbf{P}_i^{\top} \mathbf{P}_i s = \mathbf{0}\}$ is the null space of matrix $\mathbf{P}_i^{\top} \mathbf{P}_i$.*

The derivation details are available in App. G.3. We further design a shared supernetwork $g_{\psi}(\cdot)$ with parameters $\psi$ as matrix encoder, i.e., $\mathbf{P}_i = g_{\psi}(z_i)$. We should point out that although there are certain indivisible regions, our division is effective in practice and achieves a **trade-off** between model simplicity and division completeness. Then, we use the Euclidean distance between $z_i$ to measure the role differences between corresponding agents and use it as the weight of the loss value, i.e.,

$$
\mathcal{L}_{\mathrm{ort}}(\psi) = \sum_{i \neq j} \left( \|z_i - z_j\| \left\| g_{\psi}(z_i)^{\top} g_{\psi}(z_j) \right\|_F \right)^2.
\tag{10}
$$

**Discussion.** Previous works have used similar role-based hypernetwork structures, such as ROMA (Wang et al., 2020a). However, it should be noted that spatial division is more fundamental in comparison, and the policy parameters of the agents are completely shared. In order to standardize the role, ROMA considers constraining the role distance between agents, while we encode the role embedding into an orthogonal space. See App. B.1 for more discussion.

### 4.3. Diffusion Soft Policy Iteration with CD

Admittedly, our method increases the density of the policy state space, which increases difficulty for policy expressiveness. To obtain more expressive policy, we introduce a diffusion model to learn the shared policy. We consider any CD $\{f_i\}_{i=1}^n$ for generality, denoting $\vec{\boldsymbol{\pi}}^{\mathrm{CD}}(\boldsymbol{a}|s) = \prod_{i \in \mathcal{I}} \vec{\pi}^{\mathrm{CD}}(a^i|f_i(s))$ and $\bar{\boldsymbol{\pi}}^{\mathrm{CD}}(\boldsymbol{a}|s) = \prod_{i \in \mathcal{I}} \bar{\pi}^{\mathrm{CD}}(a^i|f_i(s))$ as the forward and backward processes of a joint soft diffusion policy. To optimize $J(\bar{\boldsymbol{\pi}}^{\mathrm{CD}})$, we define the corresponding joint soft $Q$-function as

$$
\hat{Q}_{\bar{\boldsymbol{\pi}}^{\mathrm{CD}}}(s_t, a_t) = r_t +
$$
$$
\mathbb{E}_{\boldsymbol{a}_{t+1:T}, s_{t+1:T}} \left[ \sum_{l=t+1}^{T} \gamma^{l-t} \left( r_l + \alpha \ell(\bar{\boldsymbol{\pi}}^{\mathrm{CD}}(\cdot|s_l)) \right) \right].
\tag{11}
$$

Thus, in policy evaluation step, we update it by

$$
\mathcal{T}^{\bar{\boldsymbol{\pi}}^{\mathrm{CD}}} Q_{\bar{\boldsymbol{\pi}}^{\mathrm{CD}}}(s_t, \boldsymbol{a}_t) \triangleq r_t + \gamma \mathbb{E}_{s' \sim P} \left[ V_{\bar{\boldsymbol{\pi}}^{\mathrm{CD}}}(s') \right],
\tag{12}
$$

$$
V_{\bar{\boldsymbol{\pi}}^{\mathrm{CD}}}(s) = \mathbb{E}_{\boldsymbol{a} \sim \bar{\boldsymbol{\pi}}^{\mathrm{CD}}} \left[ Q_{\bar{\boldsymbol{\pi}}^{\mathrm{CD}}}(s, \boldsymbol{a}) + \alpha \ell \left( \bar{\boldsymbol{\pi}}^{\mathrm{CD}}(\cdot|s) \right) \right],
\tag{13}
$$

with a soft diffusion Bellman backup operator $\mathcal{T}^{\bar{\boldsymbol{\pi}}^{\mathrm{CD}}}$. Its convergence is guaranteed, and all the proofs of this section is provided in App. G.4.

**Lemma 4.5** (**Joint Soft Diffusion Policy Evaluation with CD**). *Consider the soft Bellman backup operator $\mathcal{T}^{\overleftarrow{\pi}^{CD}}$ and initial function $\hat{Q}_0 : \mathcal{S} \times \mathcal{A} \to \mathbb{R}$ with $|\mathcal{A}| < \infty$, and define $\hat{Q}_{k+1} = \mathcal{T}^{\overleftarrow{\pi}^{CD}} \hat{Q}_k$. Then the sequence $Q_k$ will converge to the joint soft Q-function of $\overleftarrow{\pi}^{CD}$ as $k \to \infty$.*

Then we use the notation $\mathrm{Sym}(n)$ to denote the set of permutations of integers $\{1, \ldots, n\}$. Let $i_{1:m} \subseteq \mathcal{I}$ denote an ordered subset with $m$ agents and $-i_{1:m}$ be its complementary set. For policy improvement, we show that joint diffusion policy updates in Eqn. 7 can be decomposed into multiplication of sequential local diffusion policy updates.

**Proposition 4.6** (**Joint Soft Diffusion Policy Decomposition with CD**). *Let $\vec{\pi}^{CD}$ and $\overleftarrow{\pi}^{CD}$ be the forward and backward processes of a joint soft diffusion policy with CD $\{f_i\}_{i=1}^n$, and $i_{1:n} \in \mathrm{Sym}(n)$ be a permutation of $\mathcal{I}$. Suppose for each $s \in \mathcal{S}$ and $m \in \mathcal{I}$, let $h^{i_m} = f_{i_m}(s)$, update*

$$
\overleftarrow{\pi}_{new}^{CD}(\cdot|h^{i_m}) = \underset{\overleftarrow{\pi}^{CD}(\cdot|h^{i_m})}{\mathrm{argmin}}
$$
$$
D_{KL}\left( \overleftarrow{\pi}^{CD}(u_{0:K}^{i_m}|h^{i_m}) \| \vec{\pi}_{old}^{CD}(u_{0:K}^{i_m}|h^{i_m}) \right), \tag{14}
$$

*where the forward policy is given by Boltzmann distribution,*

$$
\vec{\pi}_{old}^{CD}(a^{i_m}|h^{i_m}) \propto \exp \mathbb{E}_{\boldsymbol{a}_{new}^{i_{1:m-1}}} \left[ \frac{1}{\alpha} \hat{Q}_{\overleftarrow{\pi}_{old}^{CD}}^{i_{1:m}}(s, \boldsymbol{a}_{new}^{i_{1:m-1}}, \cdot^{i_m}) \right],
$$

$$
\text{with} \quad \boldsymbol{a}_{new}^{i_{1:m-1}} \sim \prod_{j=1}^{m-1} \overleftarrow{\pi}_{new}^{CD}(a^{i_j}|h^{i_j}). \tag{15}
$$

*Then the joint diffusion policy satisfies the following:*

$$
\overleftarrow{\pi}_{new}^{CD} = \underset{\overleftarrow{\pi}^{CD} \in \check{\Pi}}{\mathrm{argmin}}\, D_{KL}\left( \overleftarrow{\pi}^{CD}(\mathbf{u}_{0:K}|s) \| \vec{\pi}_{old}^{CD}(\mathbf{u}_{0:K}|s) \right).
$$

Given this proposition, we propose an effective shared policy method to maximize $\hat{J}(\overleftarrow{\pi}^{CD})$. To demonstrate that our method has theoretical guarantees of monotonic improvement and convergence, we propose the ***Diffusion Soft Policy Iteration with CD*** (Dspic) as formalized below.

**Theorem 4.7** (**Diffusion Soft Policy Iteration with CD**). *For any initial diffusion policy $\overleftarrow{\pi}_0^{CD}$ and $\vec{\pi}_0^{CD}$ with CD $\{f_i\}_{i=1}^n$. Let $i_{1:n} \in \mathrm{Sym}(n)$ be an agent permutation and for every $m \in \mathcal{I}$, we repeatedly use Eqn. 15 in Proposition 4.6 to iterate from $\overleftarrow{\pi}_{old}^{CD}(\cdot|f_{i_m}(s))$ to obtain $\overleftarrow{\pi}_{new}^{CD}(\cdot|f_{i_m}(s))$ for all $s \in \mathcal{S}$. Then we have (1) $\hat{Q}_{\overleftarrow{\pi}_{new}^{CD}}(s, \boldsymbol{a}) \geq \hat{Q}_{\overleftarrow{\pi}_{old}^{CD}}(s, \boldsymbol{a})$ for all $(s, \boldsymbol{a}) \in \mathcal{S} \times \mathcal{A}$ and $\hat{J}\left( \overleftarrow{\pi}_{new}^{CD} \right) \geq \hat{J}\left( \overleftarrow{\pi}_{old}^{CD} \right)$, (2) the diffusion soft policy will eventually converge to a $\overleftarrow{\pi}_\star^{CD}$ as the number of iterations approaches infinity, with $\hat{Q}_{\overleftarrow{\pi}_\star^{CD}}(s, \boldsymbol{a}) \geq \hat{Q}_{\overleftarrow{\pi}^{CD}}(s, \boldsymbol{a})$ and $\hat{J}\left( \overleftarrow{\pi}_\star^{CD} \right) \geq \hat{J}\left( \overleftarrow{\pi}^{CD} \right)$ for any $\overleftarrow{\pi}^{CD} \in \check{\Pi}$.*

### 4.4. Practical Algorithm

For the whole optimization process, considering a partially observable setting for generality, where agent $i$ obtains dif-

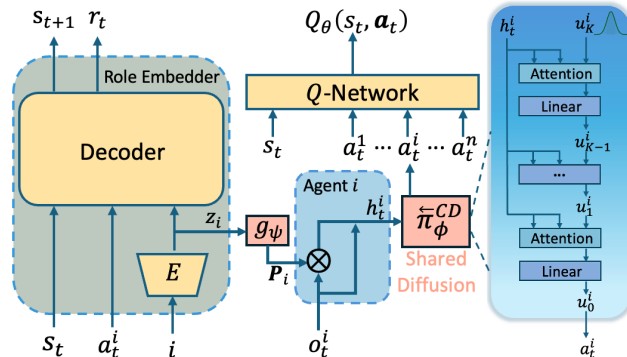

*Figure 1.* Schematics of our approach. The role embedder is pre-trained to generate the agent's role $z_i$, which is encoded into a projection matrix $\mathbf{P}_i$ by the matrix encoder $g_\psi$. Each agent constructs its individual observation space with $\mathbf{P}_i$ and makes decisions through a shared diffusion policy. Finally, we aggregate the joint actions $\boldsymbol{a}_t$ and predict $Q$-value by a centralized critic.

ferent local observations $o_t^i$ in timestep $t$ for $\forall i \in \mathcal{I}$. Note that we can still align their observation dimensions across agents by padding with zeros. Firstly, we use a parameterized function approximation for the $Q$-function, the diffusion policy and the orthogonal matrix generator, denoted as $Q_\theta$, $\overleftarrow{\pi}_\phi^{CD}$ and $g_\psi$ with parameters $\theta$, $\phi$ and $\psi$ respectively. Please note that we use DDPM as policy as described in Sec. 3.2, where $\phi$ actually represents the parameters of the denoising network. And we denote $\bar{\theta}$ as the parameters of the target $Q$-network.

In policy evaluation step, we minimize the Bellman residual

$$
\mathcal{L}(\theta) = \mathbb{E}_{s_t, \boldsymbol{a}_t}\left[ \left( Q_\theta(s_t, \boldsymbol{a}_t) - \left( r_t + \gamma \mathbb{E}_{s_{t+1}}\left[ V_{\bar{\theta}}(s_{t+1}) \right] \right) \right)^2 \right]. \tag{16}
$$

And in policy improvement step, we first pre-train the role representation $\{z_i\}_{i=1}^n$ as Sec. 4.2 by randomly exploring for some episodes (warmup). Then we couple the policy training with the projection matrix set training. Specifically, we draw a random permutation $i_{1:n}$ of $\mathcal{I}$, then for $m = 1, 2, \ldots, n$, following Eqn. 10 and Eqn. 14, we train policy with

$$
\mathcal{L}(\phi, \psi) = \lambda \mathcal{L}_{\mathrm{ort}}(\psi) - \mathbb{E}_{o^{i_{1:m}}}\left[ \alpha \ell\left( \overleftarrow{\pi}_\phi^{CD}(\cdot|h_\psi^{i_m}) \right) + \right.
$$
$$
\left. \mathbb{E}_{\boldsymbol{a}_{new}^{i_{1:m-1}}, a^{i_m} \sim \overleftarrow{\pi}_\phi^{CD}(\cdot|h_\psi^{i_m})}\left[ Q_{\theta_{old}}^{i_{1:m}}(s, \boldsymbol{a}^{i_{1:m}}) \right] \right], \tag{17}
$$

where $h_\psi^{i_m} = (g_\psi(z_{i_m})o^{i_m}, o^{i_m})$, $\lambda > 0$ and we sample $\boldsymbol{a}_{new}^{i_{1:m-1}}$ following Eqn. 15. We also add an attention gate in the denoising net to utilize condition information better. We have provided a detailed description in App. B.3.

**Conservative Temperature Adaptation.** For diffusion policy, over-exploration is dangerous, we propose ***Conservative Temperature Adaptation*** (CTA) accordingly, i.e.,

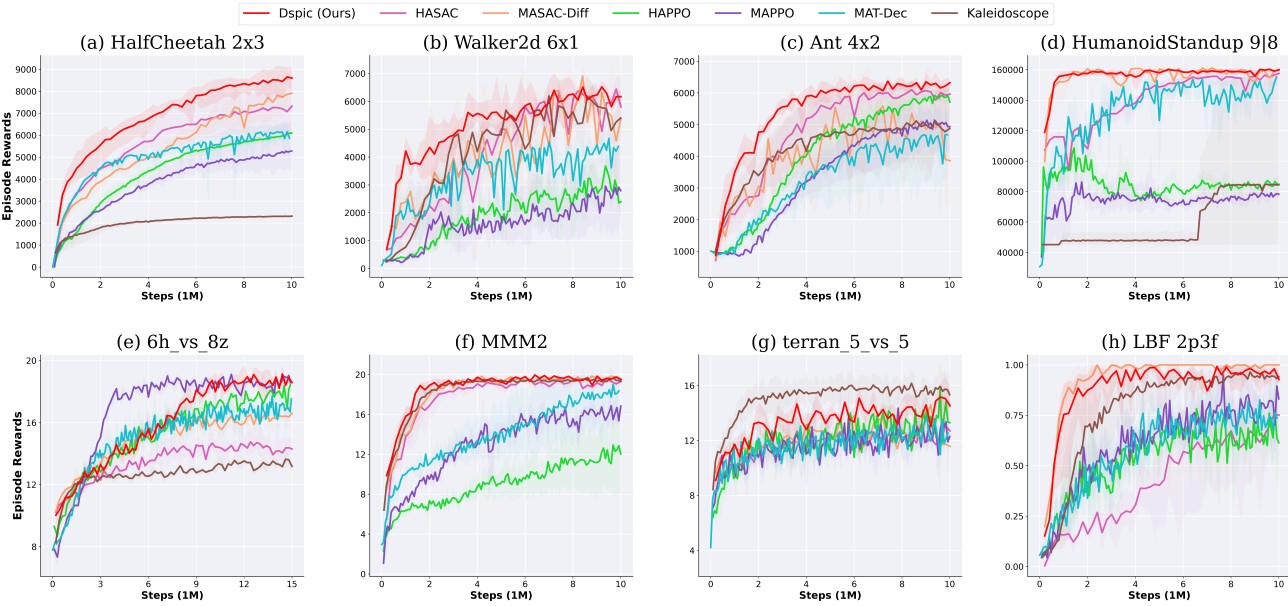

*Figure 2.* Training curves on Benchmarks. We present the average performance of each algorithm across benchmarks under 5 random seeds, and shade the area corresponding to the 95% confidence interval in our experiments.

$$\mathcal{L}(\alpha) = \max\left\{\alpha\left[\ell\left(\bar{\tilde{\pi}}^{\text{CD}}(\cdot|h^{i_m}_{\psi,t})\right) - \tilde{\mathcal{H}}\right], -\epsilon_t\right\}, \quad (18)$$

where $\tilde{\mathcal{H}}$ is the target entropy, $\epsilon_t = \epsilon_0$ when $t \leq t_0$ and $\epsilon_t = 0$ otherwise, and hyperparameters $\epsilon_0, t_0 > 0$ are insensitive to the environment we found in practice. The algorithm procedure is shown in Fig. 1, and the pseudocode and implementation details are provided in App. C and App. B.

## 5. Experiments

In this section, we evaluate Dspic in several complex multi-agent tasks to answer the following questions: **(1)** How does our method compare against multiple baselines in various tasks? **(2)** What are the advantages of our method compared to methods such as direct parameter sharing and ID concatenation? **(3)** What roles do CD and Diffusion play in our method? **(4)** Is it necessary to introduce conservative temperature adaptation (CTA) during training?

### 5.1. Experimental Setups

**Environment.** We tested our proposed method in environments with various control scenarios. Specifically, the environments we selected include **(1)** MaMuJoCo (de Witt et al., 2020), a physically realistic benchmark for cooperative MARL that evaluates decentralized coordination in continuous action spaces by partitioning single-agent robotic joints into multiple independent agents. **(2)** SMAC (Samvelyan et al., 2019) and SMACv2 (Ellis et al., 2023), which provide challenging StarCraft II micromanagement environments for

testing decentralized coordination and scalability in MARL. **(3)** LBF (Christianos et al., 2020) , where agents with diverse skill levels must learn to cooperate in a grid world to maximize rewards by collectively foraging food items. Details are provided in App. D.1.

**Baselines.** We considered different baselines here, where **(1)** HASAC (Liu et al., 2024) proposed heterogeneous-agent soft actor-critic, which trains each agent one specific policy based on sequential policy updating. **(2)** MAPPO (Yu et al., 2022) extended PPO to multi-agent scenarios by sharing policy. **(3)** HAPPO (Kuba et al., 2022) improved upon MAPPO by sequential updates with heterogeneous policies. **(4)** MAT (Wen et al., 2022) modeled MARL as a sequence prediction problem using the Transformer. Note that we selected its decentralized version (i.e., MAT-Dec) to ensure consistency with our method. **(5)**MASAC-Diff optimized the MaxEnt objective using a shared policy and acts with a diffusion policy. **(6)** Kaleidoscope (Li et al., 2024c) is an adaptive shared-policy method that utilized learnable masks to selectively share parameters among agents. The details of hyperparameters for Dspic could be found in App. F.

### 5.2. Results and Analysis

**Performance Comparisons.** We first show how our method performs compared to the baselines, as illustrated in Fig. 2. MAPPO, a shared-policy algorithm, performs poorly across most tasks, primarily because its uniform sharing scheme constrains policy diversity and leads to sub-optimal equilibria. The exception is the 6h_vs_8z task, where agents are homogeneous, making a shared policy inherently suitable. In

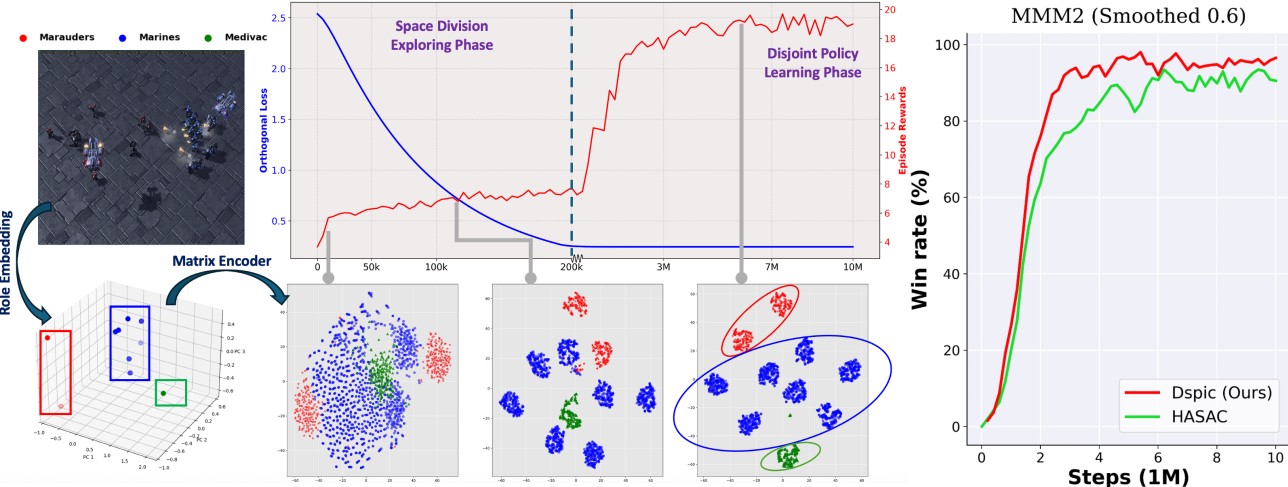

*Figure 3.* A case visualization on MMM2 task. Firstly, we present the clustering results for role embedding $z_i$. For visualizing $\mathbf{P}_i$, we randomly sample 200 points from uniform distribution (corresponding to real data) and project them onto $\mathbf{P}_i$ at different learning stages. We also demonstrate the changes in orthogonal loss and Episode Rewards during training, dividing the training into two phases accordingly. Finally, we present the win rate curves of Dspic and HASAC throughout the entire process.

contrast, HAPPO achieves performance gains over MAPPO in most scenarios by learning individual policies for each agent. However, HAPPO's performance degrades in the MMM2 task, where agents have different division of labor. This is because its purely heterogeneous architecture prevents effective knowledge sharing between agents, thereby hindering joint training. To address the suboptimality of parameter sharing, Kaleidoscope uses masks to provide heterogeneity to the agents with shared policy and demonstrates outstanding results in SMAC and SMACv2, particularly significantly outperforming other methods in the terran_5_vs_5 task. This superior performance can be attributed to that Kaleidoscope is built upon QMIX (Rashid et al., 2018), a value-decomposition MARL algorithm, whereas our method and other baselines learn explicit policies, which are not well-suited for these tasks. Modeling decision-making as time series forecasting, MAT-Dec uses the Transformer to provide a more complex policy model. Consequently, MAT-Dec consistently outperforms both MAPPO and HAPPO in various experiments.

Algorithms that lack encouragement for exploration, such as HAPPO, often get stuck in suboptimal solutions due to insufficient exploration. To mitigate this, HASAC incorporates a maximum entropy objective to encourage exploration, yielding significant improvements over HAPPO in most environments except LBF. The slight decline in LBF is attributed to its simplicity, where intensive exploration is less critical. However, HASAC overlooks inter-agent knowledge sharing, affecting its overall learning efficiency. Dspic addresses this by sharing policy with CD. Experimental results show that our algorithm outperforms all baselines in MaMuJoCo, SMAC, and LBF, achieving a significant ad-

vantage in HalfCheetah 2x3, which implies its suitability for continuous control. In the terran_5_vs_5 task, Dspic also surpasses all policy-based MARL baselines. Finally, we evaluate MASAC-Diff, a diffusion-based variant of our method, to compare against other diffusion policy algorithms. While MASAC-Diff achieves convergence performance comparable to Dspic in tasks like HumanoidStandup 9|8, it exhibits lower stability in Walker2d 6x1 and Ant 4x2 and worse performance in 6h_vs_8z. In contrast, our primary algorithm maintains superior stability throughout the training process.

Furthermore, our method demonstrates significant performance improvements across environments with 2 agents to 10 agents (e.g., HalfCheetah 2x3 and MMM2), indicating that our method is scalable with the number of agents. Simultaneously, our superior performance in the HumanoidStandup 9|8 task demonstrates that Dspic can still perform excellent tasks with heterogeneous agents by simply padding with zeros and sharing the policy. Thus, Dspic doesn't lose the advantage of heterogeneous agent methods.

**Case Study.** We set up a case study in LBF to demonstrate that our algorithm can indeed learn diverse policies for agents with shared policy. Specifically, we initialize two players (colored red and blue) at $(3, 4)$ and randomly initialize one food (apple) in the green and yellow areas of Fig. 5 respectively. The goal of players is to reach the target area and accurately collect the apples. For simplicity, we set all levels in the game as 1 to ensure that each food item can be picked up by a single player. We conduct 200 repeated experiments using Dspic and plot the trajectories of the two agents. Fig. 5 (a) shows that even if the agents use the same parameters, they can still collect apples in one direction separately, completing the task faster. Meanwhile,

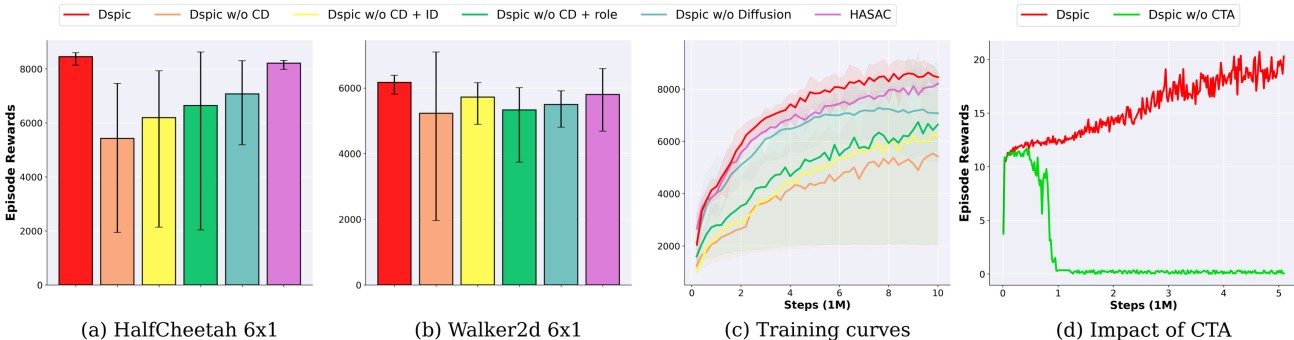

*Figure 4.* Ablation Studies. (a) and (b) demonstrate the average convergence performance and its $95\%$ confidence interval in HalfCheetah 6x1 and Walker2d 6x1 of ablation methods.(c) shows the training curves in HalfCheetah 6x1 to compare sample efficiency between methods. (d) shows the training curves of CTA ablation study on 3s5z_vs_3s6z task.

Fig. 5 (b) shows that Dspic can learn a solution of length 5 (which is theoretically optimal obviously), while shared policy can only learn a solution of length 8 at best.

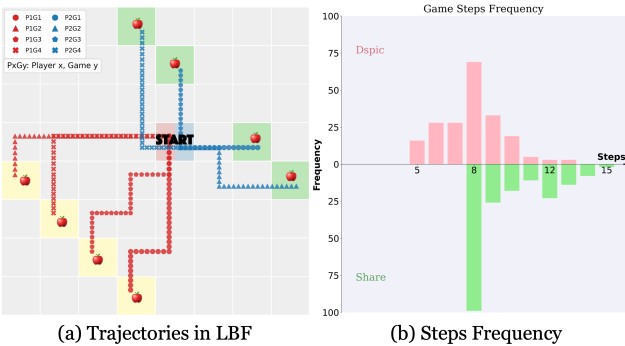

*Figure 5.* Results of case study on LBF. (a) two players' trajectories in four games, where points of the same shape originate from the same game. (b) the frequency of completing game steps with the two methods in 200 tests.

We also provide a case visualization of the more difficult MMM2 task, where 2 Marauders, 7 Marines, and 1 Medivac are scheduled to compete against the same opponent. The visualization in Fig. 3 shows that the role embedding $z_i$ does indeed contain role semantic information, and $\mathbf{P}_i$ quickly converges to a solution that can distinguish the agent's observation space. Furthermore, clustering results show that it also possesses role semantics. Once $\mathbf{P}_i$ converges, both the game reward and win rate increase rapidly, demonstrating higher sample efficiency than the baseline.

### 5.3. Ablation Studies

**The Role of CD and Diffusion.** To demonstrate respective roles of CD and diffusion policy, along with the necessity of their coexistence, we test the performance of Dspic without CD (i.e., directly share policy) and Dspic without Diffusion on HalfCheetah 6x1 and Walker2d 6x1. We also tested the performance of concatenating IDs or role embedding $z_i$ based on Dspic without CD. As shown in Fig. 4, we first

present the convergence performance of various methods in three random experiments in (a) and (b) and the training curves of HalfCheetah 6x1 in (c). For Dspic without CD, its average performance on HalfCheetah 6x1 is significantly lower than that of HASAC and Dspic, but similar on Walker2d 6x1. However, this approach exhibits unacceptable stability in all environments due to conflicts arising from direct parameter sharing between agents. By concatenating ID or role embedding, the average performance on both tasks are improved, but it still suffers from stability issues that sometimes resulted in policy collapse practically. This is because most dimensions of the states in these methods still highly overlap. The performance degradation of concatenating $z_i$ implies that spatial complete division is indeed a more fundamental approach compared to directly providing roles in observation, otherwise, we could achieve similar results to Dspic with this method.

On the other hand, Dspic without diffusion achieves similar performance to HASAC in both environments, indicating that CD can guarantee optimality in practice. However, due to the excessive representational complexity of simultaneously modeling policies for $n$ agents, it also faces the issue of training instability. The training curve in Fig. 4 (c) also reveals this, especially in the later stages of training. Diffusion policy can improve stability and asymptotic performance due to its powerful expressiveness.

**The Necessity of** *Conservative Temperature Adaptation* **(CTA).** To demonstrate the necessity of CTA in Eqn. 18, we conduct experiments on the 3s5z_vs_3s6z task. As shown in Fig. 4(d), training without CTA resulted in significant policy collapse, leading to policy optimization failure. Conversely, clipping loss with CTA made training more robust, ensuring convergence to a better policy. In conclusion, by introducing CTA, the Dspic does not exhibit policy collapse, significantly improving the training stability. Additional comparative experiments and ablation studies are available in App. D.2 and D.3.

# 6. Conclusion and Future Work

To balance the trade-off between the knowledge sharing and policy customization in MARL and improve the learning efficiency, we first theoretically analyze the suboptimal problem of shared policies and propose CD for the observation space to guarantee optimality. Next, we introduce diffusion policy to ensure the policy's expressiveness as the observation space becomes denser, and propose the Dspic. Extensive experimental results demonstrate that our approach outperforms a broad baseline in various tasks, demonstrating its efficacy in complex heterogeneous agents problems. Frankly, constructing CDs with fewer indivisible regions is desirable and could potentially further improve model performance. Meanwhile, our method now still focuses on cyber space, the application of it in real-world scenarios and multi-agent embodied tasks (Liu et al., 2025; Feng et al., 2025) is an important direction for future work.

## Acknowledgments

This work was supported by the National Natural Science Foundation of China under Grants 62506159, and U24A20324; the Natural Science Foundation of Jiangsu under Grants BK20241199 and BK20243039; the "111 Center" (No. B26023); and the Fundamental and Interdisciplinary Disciplines Breakthrough Plan of the Ministry of Education of China (No.JYB2025XDXM118). And Jianhong Wang is supported by the Engineering and Physical Sciences Research Council (EPSRC) [Grant Ref: EP/Y028732/1]. Guanghao Li would also like to thank Peng Xie and Xinhao Chen for their support of this work.

## Impact Statement

The goal of the work presented in this paper is to advance the development of heterogeneous MARL. The proposed framework is intended to enhance the exploration of agent teams, providing an effective approach for future research on complex MARL cooperation. Furthermore, the work presented does not raise any additional ethical concerns, and thus no special discussion on ethical issues is required.

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

# A. Detailed Preliminaries

## A.1. Heterogeneous-Agent Soft Policy Iteration

QRE policy is a generalization of Nash equilibrium policy to nondeterministic policies, and its rigorous definition is given here.

**Definition A.1** (**QRE Policy**). A joint policy $\pi_{QRE} \in \mathbf{\Pi}$ is a QRE if none of the agents can increase the maximum entropy objective (Eqn. 1) by only altering its policy, i.e.,

$$\forall i \in \mathcal{N}, \forall \pi^i \in \Pi^i, J\left(\pi^i, \pi_{QRE}^{-i}\right) \le J\left(\pi_{QRE}\right).$$

where $\Pi^i \triangleq \{\times_{s \in \mathcal{S}} \pi^i(\cdot|s_t)|\forall s \in \mathcal{S}\}$ denotes the policy space of agent $i$, and $\mathbf{\Pi} \triangleq (\Pi^1, \Pi^2, ..., \Pi^n)$ denotes the joint policy space.

To get the QRE policy, Heterogeneous-Agent Soft Policy Iteration (HASPI) framework consists of two core steps, policy evaluation and policy improvement. Given a joint policy $\pi$, policy evaluation aims to evaluate the value of $\pi$ with a soft Bellman backup operator $\mathcal{T}^{\pi}$ by Eqn. 3 and Eqn. 4, i.e. $Q_{k+1} = \mathcal{T}^{\pi} Q_k$ . Therefore, $Q_k$ eventually converges with $k \to \infty$.

To explain the policy update, we first define multi-agent soft Q-function as follows.

**Definition A.2.** Let $i_{1:m} = \{i_1, \ldots, i_m\} \subseteq \mathcal{I}$ be an ordered subset of agents, and let $-i_{1:m}$ refer to its complement. We write $i_k$ when we refer to the $k^{\text{th}}$ agent in the ordered subset. Correspondingly, the multi-agent soft $Q$-function is defined as

$$Q_{\pi}^{i_{1:m}}\left(s, \boldsymbol{a}^{i_{1:m}}\right) \triangleq \mathbb{E}_{\boldsymbol{a}^{-i_{1:m}} \sim \pi^{-i_{1:m}}}\left[Q_{\pi}\left(s, \boldsymbol{a}^{i_{1:m}}, \boldsymbol{a}^{-i_{1:m}}\right) + \alpha \sum_{i \in -i_{1:m}} \mathcal{H}\left(\pi^i(\cdot|s)\right)\right]. \tag{19}$$

Specifically, we have $Q_{\pi}^{i_{1:n}}\left(s, \boldsymbol{a}^{i_{1:n}}\right) = Q_{\pi}(s, \boldsymbol{a})$ if $m = n$ and the function represents the soft value function $V_{\pi}(s)$ when $m = 0$, i.e., $i_{1:m} = \emptyset$.

Policy improvement of HASPI is a sequential process that randomly generates an agent permutation $i_{1:n} \in \text{Sym}(n)$ then update the agents in this order with:

$$\pi_{\text{new}}^{i_m} = \operatorname*{argmin}_{\pi^{i_m} \in \Pi^{i_m}} D_{\text{KL}}\left(\pi^{i_m}(\cdot^{i_m}|s) \middle\| \frac{\exp\left(\mathbb{E}_{\boldsymbol{a}^{i_{1:m-1}} \sim \pi_{\text{new}}^{i_{1:m-1}}}\left[\frac{1}{\alpha}Q_{\pi_{\text{old}}}^{i_{1:m}}(s, \boldsymbol{a}^{i_{1:m-1}}, \cdot^{i_m})\right]\right)}{\mathbb{E}_{\boldsymbol{a}^{i_{1:m-1}} \sim \pi_{\text{new}}^{i_{1:m-1}}}\left[Z_{\pi_{\text{old}}}(s, \boldsymbol{a}^{i_{1:m-1}})\right]}\right), \tag{20}$$

where $Z_{\pi_{\text{old}}}(s, \boldsymbol{a}^{i_{1:m-1}})$ normalizes the distribution. Through individual updates sequentially, the joint policy obtained by HASPI simultaneously satisfies

$$\pi_{\text{new}} = \operatorname*{argmin}_{\pi \in \mathbf{\Pi}} D_{\text{KL}}\left(\pi(\cdot|s) \middle\| \frac{\exp\left(\frac{1}{\alpha}Q_{\pi_{\text{old}}}(s, \cdot)\right)}{Z_{\pi_{\text{old}}}(s)}\right). \tag{21}$$

The monotonicity of HASPI are guaranteed with $Q_{\pi_{\text{new}}}(s, \boldsymbol{a}) \ge Q_{\pi_{\text{old}}}(s, \boldsymbol{a})$ for all $(s, \boldsymbol{a}) \in \mathcal{S} \times \mathcal{A}$ and $J\left(\pi_{\text{new}}\right) \ge J\left(\pi_{\text{old}}\right)$, and the joint policy eventually converges to the QRE policy $\pi_{\star}^{\text{HA}}$.

## A.2. Multi-Agent Advantage Function

With multi-agent soft Q-function defined in A.2, we define the multi-agent soft advantage function correspondingly,

**Definition A.3.** Let $i_{1:m} = \{i_1, \ldots, i_m\} \subseteq \mathcal{I}$ and $j_{1:k} = \{j_1, \ldots, j_k\} \subseteq \mathcal{I}$ be disjoint ordered subsets of agents, and let $Q_{\pi}^{i_{1:m}}(s, \boldsymbol{a}^{i_{1:m}})$ be the multi-agent soft $Q$-function. Then the multi-agent soft advantage function is defined as

$$A_{\pi}^{i_{1:m}}\left(s, \boldsymbol{a}^{j_{1:k}}, \boldsymbol{a}^{i_{1:m}}\right) \triangleq Q_{\pi}^{j_{1:k}, i_{1:m}}\left(s, \boldsymbol{a}^{j_{1:k}}, \boldsymbol{a}^{i_{1:m}}\right) - Q_{\pi}^{j_{1:k}}\left(s, \boldsymbol{a}^{j_{1:k}}\right). \tag{22}$$

We need to use the relevant *Multi-Agent Advantage Decomposition Lemma* to support our subsequent proof.

**Lemma A.4** (Multi-Agent Advantage Decomposition). *Let $\boldsymbol{\pi}$ be a joint policy, and $i_1, \ldots, i_m$ be an arbitrary ordered subset of agents. Then, for any state $s$ and joint action $\boldsymbol{a}^{i_{1:m}}$,*

$$A_{\boldsymbol{\pi}}^{i_{1:m}}\left(s, \boldsymbol{a}^{i_{1:m}}\right) = \sum_{j=1}^{m} A_{\boldsymbol{\pi}}^{i_j}\left(s, \boldsymbol{a}^{i_{1:j-1}}, a^{i_j}\right). \tag{23}$$

*Proof.* (The lemma is proposed in (Kuba et al., 2021) and is generalized to the soft advantage function in (Liu et al., 2024).) Note that,

$$\begin{aligned} A_{\boldsymbol{\pi}}^{i_{1:m}}\left(s, \boldsymbol{a}^{i_{1:m}}\right) &= Q_{\boldsymbol{\pi}}^{i_{1:m}}\left(s, \boldsymbol{a}^{i_{1:m}}\right) - V_{\boldsymbol{\pi}}(s) \\ &= \sum_{j=1}^{m}\left[Q_{\boldsymbol{\pi}}^{i_{1:j}}\left(s, \boldsymbol{a}^{i_{1:j}}\right) - Q_{\boldsymbol{\pi}}^{i_{1:j-1}}\left(s, \boldsymbol{a}^{i_{1:j-1}}\right)\right] = \sum_{j=1}^{m} A_{\boldsymbol{\pi}}^{i_j}\left(s, \boldsymbol{a}^{i_{1:j-1}}, a^{i_j}\right). \end{aligned}$$

$\square$

## A.3. Denoising Diffusion Probabilistic Models

Consider a data $\boldsymbol{x}_0$ from the original data distribution $q(\boldsymbol{x}_0)$, the forward process of DDPM progressively perturbs it at each diffusion step $k = 1, 2, ..., K$:

$$q(\boldsymbol{x}_k|\boldsymbol{x}_{k-1}) := \mathcal{N}(\boldsymbol{x}_k; \sqrt{1-\beta_k}\boldsymbol{x}_{k-1}, \beta_k\mathbf{I}), \tag{24}$$

where $\beta_k > 0$ and $q(\boldsymbol{x}_k|\boldsymbol{x}_0)$ can be deduced to :

$$q(\boldsymbol{x}_k|\boldsymbol{x}_0) = \mathcal{N}(\boldsymbol{x}_k; \sqrt{\bar{\alpha}_k}\boldsymbol{x}_0, (1-\bar{\alpha}_k)\mathbf{I}), \tag{25}$$

where $\alpha_k = 1 - \beta_k$ and $\bar{\alpha}_k = \prod_{s=0}^{k} \alpha_s$. Eventually, $\boldsymbol{x}_K$ converges to $\mathcal{N}(\mathbf{0}, \mathbf{I})$ when $K \to \infty$. Thus, in the reverse process of DDPM, we first sample $x_K \sim \mathcal{N}(\mathbf{0}, \mathbf{I})$ and then gradually denoises it by unknown transition $p_\theta(\boldsymbol{x}_{k-1}|\boldsymbol{x}_k) = \mathcal{N}(\boldsymbol{x}_{k-1}; \mu_\theta(\boldsymbol{x}_k, k), \Sigma_\theta(\boldsymbol{x}_k, k))$ to estimate $q(\boldsymbol{x}_t|\boldsymbol{x}_{k-1}, \boldsymbol{x}_0)$, where $\mu_\theta$ and $\Sigma_\theta$ represent the predicted average value and covariance matrix parameterized by $\theta$. The training objective of DDPM is to maximize the *Variational **LO**wer Bound* (**VLO**) defined as $\mathcal{L}_{\text{VLO}} = \mathbb{E}_{q(\boldsymbol{x}_{0:K})}\left[\log \frac{p_\theta(\boldsymbol{x}_{0:K})}{q(\boldsymbol{x}_{1:K}|\boldsymbol{x}_0)}\right]$. Finally, with $\epsilon \sim \mathcal{N}(\mathbf{0}, \mathbf{I})$, the loss in DDPM takes the form of:

$$\mathbb{E}_{k,\boldsymbol{x}_0,\epsilon}\left[\left\|\epsilon - \epsilon_\theta(\sqrt{\bar{\alpha}_k}\boldsymbol{x}_0 + \sqrt{1-\bar{\alpha}_k}\epsilon_k, k)\right\|^2\right]. \tag{26}$$

## A.4. Diffusion-Based Maximum Entropy Reinforcement Learning

In this section, we will supplement the proof of the formulas in Sec. 3.2, which will also be very helpful in understanding our work.

First, we focus on Eqn. 5, i.e.

$$J\left(\bar{\boldsymbol{\pi}}\right) \geq \hat{J}\left(\bar{\boldsymbol{\pi}}\right) = \mathbb{E}_{s_{0:T}, \boldsymbol{a}_{0:T}}\left[\sum_{t=0}^{T} \gamma^t\left(r_t + \alpha\ell(\bar{\boldsymbol{\pi}}\left(\cdot|s_t\right))\right)\right],$$

where

$$\ell(\bar{\boldsymbol{\pi}}\left(\cdot|s\right)) = \mathbb{E}_{\mathbf{u}_{0:K} \sim \bar{\boldsymbol{\pi}}_{0:K}}\left[\log \frac{\vec{\boldsymbol{\pi}}_{1:K|0}(\mathbf{u}_{1:K}|\mathbf{u}_0, s)}{\overleftarrow{\boldsymbol{\pi}}_{0:K}(\mathbf{u}_{0:K}|s)}\right].$$

*Proof.* (The proof is cited from (Celik et al., 2025)) Fisrtly, we have

$$
\begin{aligned}
\mathcal{H}(\overleftarrow{\boldsymbol{\pi}}\,(\cdot|s)) &= -\mathbb{E}_{u_{0:K}\sim\overleftarrow{\boldsymbol{\pi}}_{0:K}}\left[\log\frac{\overleftarrow{\boldsymbol{\pi}}_{0:K}(u_{0:K}|s)}{\overrightarrow{\boldsymbol{\pi}}_{1:K|0}(u_{1:K}|s,u_0)}\right] \\
&= -\mathbb{E}_{u_{0:K}\sim\overleftarrow{\boldsymbol{\pi}}_{0:K}}\left[\log\frac{\overleftarrow{\boldsymbol{\pi}}_{0:K}(u_{0:K}|s)\overrightarrow{\boldsymbol{\pi}}_{1:K|0}(u_{1:K}|s,u_0)}{\overrightarrow{\boldsymbol{\pi}}_{1:K|0}(u_{1:K}|s,u_0)\overrightarrow{\boldsymbol{\pi}}_{1:K|0}(u_{1:K}|s,u_0)}\right] \\
&= \mathbb{E}_{u_{0:K}\sim\overleftarrow{\boldsymbol{\pi}}_{0:K}}\left[\log\frac{\overrightarrow{\boldsymbol{\pi}}_{1:K|0}(u_{1:K}|s,u_0)}{\overleftarrow{\boldsymbol{\pi}}_{0:K}(u_{0:K}|s)}\right] + \mathbb{E}_{u_{0:K}\sim\overleftarrow{\boldsymbol{\pi}}_{0:K}}\left[\log\frac{\overrightarrow{\boldsymbol{\pi}}_{1:K|0}(u_{1:K}|s,u_0)}{\overrightarrow{\boldsymbol{\pi}}_{1:K|0}(u_{1:K}|s,u_0)}\right] \\
&\text{with } \overleftarrow{\boldsymbol{\pi}}_0(u_0|s) = \frac{\overleftarrow{\boldsymbol{\pi}}_{0:K}(u_{0:K}|s)}{\overleftarrow{\boldsymbol{\pi}}_{1:K|0}(u_{1:K}|s,u_0)} \\
&= \mathbb{E}_{u_{0:K}\sim\overleftarrow{\boldsymbol{\pi}}_{0:K}}\left[\log\frac{\overrightarrow{\boldsymbol{\pi}}_{1:K|0}(u_{1:K}|s,u_0)}{\overleftarrow{\boldsymbol{\pi}}_{0:K}(u_{0:K}|s)}\right] + \mathbb{E}_{u_0\sim\overleftarrow{\boldsymbol{\pi}}_0}\left[D_{\mathrm{KL}}\left(\overleftarrow{\boldsymbol{\pi}}_{1:K|0}(u_{1:K}|s,u_0)\,\|\,\overrightarrow{\boldsymbol{\pi}}_{1:K|0}(u_{1:K}|s,u_0)\right)\right] \\
&\geq \mathbb{E}_{u_{0:K}\sim\overleftarrow{\boldsymbol{\pi}}_{0:K}}\left[\log\frac{\overrightarrow{\boldsymbol{\pi}}_{1:K|0}(u_{1:K}|s,u_0)}{\overleftarrow{\boldsymbol{\pi}}_{0:K}(u_{0:K}|s)}\right] \\
&= \mathbb{E}_{u_{0:K}\sim\overleftarrow{\boldsymbol{\pi}}_{0:K}}\left[\sum_{k=1}^{K}\overrightarrow{\boldsymbol{\pi}}_{k|k-1}(u_k|u_{k-1},s) - \sum_{k=1}^{K}\log\overleftarrow{\boldsymbol{\pi}}_{k-1|k}(u_{k-1}|u_k,s) - \log\overleftarrow{\boldsymbol{\pi}}_K(u_K|s)\right] \\
&= -\mathbb{E}_{u_{0:K}\sim\overleftarrow{\boldsymbol{\pi}}_{0:K}}\left[\log\overleftarrow{\boldsymbol{\pi}}_K(u_K|s) + \sum_{k=1}^{K}\log\frac{\overleftarrow{\boldsymbol{\pi}}_{k-1|k}(u_{k-1}|u_k,s)}{\overrightarrow{\boldsymbol{\pi}}_{k|k-1}(u_k|u_{k-1},s)}\right].
\end{aligned}
$$

Directly, we replace $\mathcal{H}(\overleftarrow{\boldsymbol{\pi}}\,(\cdot|s_t))$ with $\ell(\overleftarrow{\boldsymbol{\pi}}\,(\cdot|s_t))$ in $J(\overleftarrow{\boldsymbol{\pi}}\,)$,

$$
J\left(\overleftarrow{\boldsymbol{\pi}}\,\right) \geq \hat{J}\left(\overleftarrow{\boldsymbol{\pi}}\,\right) = \mathbb{E}_{s_{0:T},\boldsymbol{a}_{0:T}}\left[\sum_{t=0}^{T}\gamma^t\left(r_t + \alpha\ell(\overleftarrow{\boldsymbol{\pi}}\,(\cdot|s_t))\right)\right].
$$

$\square$

At the same time, $\ell(\cdot)$ can be written in a more easily computable form,

$$
\begin{aligned}
\ell(\overleftarrow{\boldsymbol{\pi}}\,(\cdot\mid s)) &= \mathbb{E}_{u_{0:K}\sim\overleftarrow{\boldsymbol{\pi}}_{0:K}}\left[\log\frac{\overrightarrow{\boldsymbol{\pi}}_{1:K|0}(u_{1:K}|u_0,s)}{\overleftarrow{\boldsymbol{\pi}}_{0:K}(u_{0:K}|s)}\right] \\
&= -\mathbb{E}_{u_{0:K}\sim\overleftarrow{\boldsymbol{\pi}}_{0:K}}\left[\log\overleftarrow{\boldsymbol{\pi}}_K(u_K|s) + \sum_{k=1}^{K}\log\frac{\overleftarrow{\boldsymbol{\pi}}_{k-1|k}(u_{k-1}|u_k,s)}{\overrightarrow{\boldsymbol{\pi}}_{k|k-1}(u_k|u_{k-1},s)}\right].
\end{aligned}
$$

As for target Eqn. 7, we show that,

*Proof.* Let

$$
\hat{Q}_{\overleftarrow{\boldsymbol{\pi}}}\left(s_t,a_t\right) = r_t + \mathbb{E}_{\boldsymbol{a}_{t+1:T},s_{t+1:T}}\left[\sum_{l=t+1}^{T}\gamma^{l-t}\left(r_l + \alpha\ell(\overleftarrow{\boldsymbol{\pi}}\,(\cdot|s_l))\right)\right].
$$

Then the forward joint policy is given by Boltzmann distribution, i.e.,

$$
\overrightarrow{\boldsymbol{\pi}}(\boldsymbol{a}_0|s_0) = \frac{\exp\left(\alpha^{-1}\hat{Q}_{\overleftarrow{\boldsymbol{\pi}}}\left(s_0,\boldsymbol{a}_0\right)\right)}{Z_{\overleftarrow{\boldsymbol{\pi}}}\left(s_0\right)},
$$

where, $Z_{\bar{\pi}}(s_0)$ normalizes the distribution. Firstly, we rewrite the expression for $\hat{J}(\bar{\pi})$,

$$
\begin{aligned}
\hat{J}(\bar{\pi}) &= \mathbb{E}_{s_{0:T}\sim\rho_{\bar{\pi}}^{0:T},\boldsymbol{a}_{0:T}\sim\bar{\pi}}\left[\sum_{t=0}^{T}\gamma^t\left(r(s_t,\boldsymbol{a}_t)+\alpha\sum_{i=1}^{n}\ell(\bar{\pi}(\cdot\mid o_t^i))\right)\right] \\
&= \mathbb{E}_{s_{0:T}\sim\rho_{\bar{\pi}}^{0:T},\boldsymbol{a}_{0:T}\sim\bar{\pi}}\left[\sum_{t=0}^{T}\gamma^t\left(r(s_t,\boldsymbol{a}_t)+\alpha\ell(\bar{\pi}(\cdot\mid s_t))\right)\right] \\
&= \mathbb{E}_{s_0\sim\rho_{\bar{\pi}}^0}\left[\alpha\ell(\bar{\pi}(\cdot|s_0))\right]+\mathbb{E}_{s_0\sim\rho_{\bar{\pi}}^0,\boldsymbol{a}_0\sim\bar{\pi}}\left[\hat{Q}_{\bar{\pi}}(s_0,\boldsymbol{a}_0)\right] \\
&\quad\text{with } \vec{\pi}(\boldsymbol{a}_0|s_0)=\frac{\exp\left(\alpha^{-1}\hat{Q}_{\bar{\pi}}(s_0,\boldsymbol{a}_0)\right)}{Z_{\bar{\pi}}(s_0)} \\
&= \mathbb{E}_{s_0\sim\rho_{\bar{\pi}}^0}\left[\alpha\ell(\bar{\pi}(\cdot|s_0))\right]+\alpha\mathbb{E}_{s_0\sim\rho_{\bar{\pi}}^0,\boldsymbol{a}_0\sim\bar{\pi}}\left[\log\vec{\pi}(\boldsymbol{a}_0|s_0)+\log Z_{\bar{\pi}}(s_0)\right] \\
&= -\alpha\mathbb{E}_{s_0\sim\rho_{\bar{\pi}}^0}\left[D_{\mathrm{KL}}\left(\bar{\pi}(\mathbf{u}_{0:K}|s_0)\|\vec{\pi}(\mathbf{u}_{0:K}|s_0)\right)\right]+\alpha\mathbb{E}_{s_0\sim\rho_{\bar{\pi}}^0}\left[\log Z_{\bar{\pi}}(s_0)\right].
\end{aligned}
$$

Because the normalizer $Z_{\bar{\pi}}(s_0)$ is a constants independent of $\mathbf{u}_{0:K}$, we have

$$
\max_{\bar{\pi}\in\tilde{\Pi}}\hat{J}(\bar{\pi})\Leftrightarrow\min_{\bar{\pi}\in\tilde{\Pi}}\mathbb{E}_s\left[D_{\mathrm{KL}}\left(\bar{\pi}(\mathbf{u}_{0:K}|s)\|\vec{\pi}(\mathbf{u}_{0:K}|s)\right)\right].
$$

$\square$

It is worth mentioning that the objective can be rewrite to

$$
\max_{\bar{\pi}\in\tilde{\Pi}}\hat{J}(\bar{\pi})\Leftrightarrow\max_{\bar{\pi}\in\tilde{\Pi}}\mathbb{E}_s\left[\mathbb{E}_{\boldsymbol{a}\sim\bar{\pi}}\left[\hat{Q}_{\boldsymbol{\pi}}(s,\boldsymbol{a})\right]+\alpha\ell(\bar{\pi}(\cdot|s))\right]. \tag{27}
$$

# B. Implementation Details

## B.1. Details of Role Embeddings

As for the role embedder, We use two fully connected layers with a hidden size of 64 as the encoder of the VAE and three fully connected layers with a hidden size of 64 as the decoder of the VAE. We set the dimension of the latent space to 10. The VAE with $\varphi$ as parameter is trained after the on-policy learning warmup, which we typically set to 10000 timesteps by random exploration. We obtain all transition pairs in the replay buffer as $(s_t, \boldsymbol{a}_t, s_{t+1}, r_t)$ and then train for 10 epochs with a batch size of 128. The training loss of the VAE can be written as

$$
\mathcal{L}_{\mathrm{VAE}}(\varphi)=\mathbb{E}_{i\in\mathcal{I}}\left[\mathrm{MSE}(\hat{r}_t, r_t)+\mathrm{MSE}(\hat{s}_{t+1}, s_{t+1})+D_{\mathrm{KL}}(p_\varphi(z_i\mid i)|p(z_i))\right], \tag{28}
$$

where we set $p(z_i)=\mathcal{N}(\mathbf{0},\mathbf{I})$, $\mathrm{MSE}(\hat{y}, y)=\|\hat{y}-y\|_2^2$ and $\hat{r}_t, \hat{s}_{t+1}$ is the prediction value of the decoder. In the case of partial observability, we can obtain different observations for different agents denoted as $o_t^i$, and the transition pairs can be written as $(\mathbf{o}_t^{1:n}, \boldsymbol{a}_t, \mathbf{o}_{t+1}^{1:n}, r_t)$. Then we train the VAE with

$$
\mathcal{L}_{\mathrm{PO\text{-}VAE}}(\varphi)=\mathbb{E}_{i\in\mathcal{I}}\left[\mathrm{MSE}(\hat{r}_t, r_t)+\mathrm{MSE}(\hat{o}_{t+1}^i, o_{t+1}^i)+D_{\mathrm{KL}}(p_\varphi(z_i\mid i)|p(z_i))\right], \tag{29}
$$

where $\hat{o}_{t+1}^i$ is the prediction of next observation by agent $i$. Note that, this architecture is similar to SePS (Christianos et al., 2021), but we believe that the role embedding $z_i$ should have more useful semantic information, rather than simply achieving policy sharing through clustering.

After VAE training is complete, we input the one-hot ID into its Encoder to obtain the latent vector $\{z_i\}_{i=1}^n$. After generation, we normalize the role embeddings, mathematically, for the $j$-th dimension of the $i$-th agent's embedding, we transform via $\frac{z_{ij}-\min_k z_{kj}}{\max_k z_{kj}-\min_k z_{kj}+\epsilon}\to z_{ij}$. And $g_\psi(\cdot)$ is a a two-layer fully connected neural network with a hidden size of 64. Typically, we obtain a vector with dimension $d_s\times d'$ as its output, and then reshape it to size $(d_s, d')$ as $\mathbf{P}_i$. Finally we normalize all $\mathbf{P}_i$ by dividing them by their Frobenius norm $\|\mathbf{P}_i\|_F$.

**Discussion.** Frankly, our architecture is largely inspired by the design of SePS (Christianos et al., 2021), as it is both simple and effective for role information extraction. At a deeper level, our core arguments are: (**1**) Classifying agents solely based on clustering methods can improve model capabilities, but requires prior knowledge of the environment to determine the number of clusters. (**2**) Embedding roles into a low-dimensional latent space implies the assumption that role information can be fully expressed within it. However, our experiments in Sec. 5.3 show that simply concatenating role information does not effectively improve model capabilities and still faces stability issues, **indicating that directly using role embedding is flawed.** Our method, Dspic, further embeds role information into a higher-dimensional matrix space and trains dynamically, balancing the dynamic characteristics of roles with deeper semantics. Furthermore, we believe that using role information to divide the observation space addresses the issues of shared policies from a more fundamental perspective. (**3**) Since we chose Kaleidoscope, a method that outperforms SePS in various environments, we did not compare the performance of SePS in the comparative experiments. Furthermore, SePS was initially only applied in tasks with obvious roles, such as SMAC. In tasks like MaMuJoCo, it is difficult to heuristically determine the number of clusters, and we prefer to compare methods that are applicable to various tasks.

**Comparison with CTR (Li et al., 2024b).** We demonstrate the differences between CTR and Dspic from two perspectives, although they have a similar observation that input overlap can lead to failure in learning shared policies and try to solve it by roles. **For role learning methods**, CTR employs contrastive learning with other agents as negative samples to foster diversity when learning roles. In contrast, Dspic learns roles by VAE from trajectories and encodes it into orthogonal matrix space. **For algorithmic mechanism**, CTR is a soft-constraint method that provides contrast loss as diversity regularization, while Dspic is a hard-constraint method by projecting the input directly into disjoint space. Projection provides an isolation of information flow and yields better performance.

**Comparison with ADMN (Yu et al., 2024).** ADMN is still a soft constraint approach, and it actually expresses the knowledge shared by agents through the idea of dynamic routing. Our method, on the other hand, directly explores the principle of shared suboptimality and solves the problem from there.

**Comparison with GradPS (Qin et al., 2025).** GradPS is a neuron-level study that determines which neurons influence policy sharing by measuring the degree of gradient perturbation. While it is also a form of hard constraint, the biggest difference between our approach and GradPS is that we address the problem directly at the input, whereas GradPS analyzes gradients during training to solve the problem.

## B.2. The Q-network

We use distributional-$Q$ following (Bellemare et al., 2017), we model the $Q$-function as a distribution of $b$-bins, denoted as $q_\theta(\cdot \mid s, \boldsymbol{a})$. Then we can compute the prediction value with $Q_\theta(s, \boldsymbol{a}) = \mathbb{E}_{b \sim q_\theta(\cdot \mid s, \boldsymbol{a})}[b]$. Follow (Celik et al., 2025), we train it by using the entropy-regularized cross-entropy loss $\mathcal{L}(\theta) = -\sum q_{target} \log q_\theta - 0.005 \sum q_\theta \log q_\theta$, where the target probabilities $q_{target}$ is calculated by using the bellman backup operator for diffusion models from Eqn. 12 and Eqn. 13, alone with the bin values $b$ following (Bellemare et al., 2017). Then we use two $Q$-network with parameters $\theta_1$ and $\theta_2$, we take the minimum of the two values when we need to obtain the $Q$-value, just following (Liu et al., 2024) and (Haarnoja et al., 2018). These two networks correspond to two target networks, with $\bar{\theta}_1$ and $\bar{\theta}_2$ as parameters, respectively.

## B.3. The Denoising Network

The denoising network $\bar{\pi}_\phi^{\mathrm{CD}}(\cdot \mid u_k^i, h_\psi^i)$ for agent $i$ takes $u_k^i$ as input while $h_\psi^i$ and $k$ as conditions in for the $k$-th diffusion step. By using gated attention, we can better utilize this condition information instead of directly concatenating it, i.e.,

$$
\begin{aligned}
u_{k-1}^i &\sim \mathrm{Linear}((\mathbf{1} - g_\phi) \odot q_\phi + g_\phi \odot v_\phi) \\
q_\phi &= \mathbf{W}_\phi^Q u_k^i \quad v_\phi = \mathbf{W}_\phi^V(h_\psi^i, k)
\end{aligned}
\tag{30}
$$

where $g_\phi$ comes from a two-layer fully connected network with hidden size of 64 by input $(q_\phi, v_\phi)$, and $\odot$ means element-wise product. Our approach is essentially a simplified version of the attention mechanism, where we take conditional information as key and value inputs, and the diffusion latent variable as the query. This method has demonstrated excellent results in *Latent Diffusion Model* (Rombach et al., 2022).

## B.4. Details of Actors

**For Continuous Action Space.** We assume that each dimension of the action space is finite, which is often satisfied in practice. At this point, we can let diffusion predict values in $[-1, 1]$, and then scale them to the actual space. Specifically, after obtaining $u_0^i$ of agent $i$ through $\overleftarrow{\pi}_\phi^{\text{CD}}(\cdot|h_\psi^i)$, the final action $a^i$ calculated by

$$a^i = \tanh(u_0^i). \tag{31}$$

Then the log probability of $a^i$ is expressed as

$$\log p_\phi(a^i|h_\psi^i) = \log \overleftarrow{\pi}_\phi^{\text{CD}}(u_0^i|h_\psi^i) + \sum_{j=1}^{D} \log\left(1 - \tanh^2(u_{0,j}^i)\right), \tag{32}$$

where $D$ is the dimension of action space $\mathcal{A}$ and $u_{0,j}^i$ denotes the $j$-th dimension of $u_0^i$. At the same time, we use *Action Selection*, which means sampling multiple actions and selecting the one with the largest $Q$-value to execute. Formulaically,

$$\overleftarrow{\pi}_K^{\text{CD}}(\cdot|s) \triangleq \underset{\boldsymbol{a}\sim\{\boldsymbol{a}_1,\ldots,\boldsymbol{a}_K\sim\overleftarrow{\pi}_\phi^{\text{CD}}(\cdot|s)\}}{\operatorname{argmax}} Q(s, \boldsymbol{a}), \tag{33}$$

where $K > 0$ is then number of samples. Follow QVPO (Ding et al., 2024), we choose the action selection number $K_b$ for the behavior policy, and a smaller action selection number $K_t < K_b$ for the target policy. In practice, we set $K_b = 4$ and $K_t$ for all environment.

**For Discrete Action Space.** We assume $|\mathcal{A}| = m$ then for every agent, there are $m$ selectable actions after padding. We let Diffusion generate an unbounded $m$-dimensional vector, representing the logits for each action, denoted as $l^i$ for agent $i$, i.e. $l^i = u_0^i$. Then the action is sampled with $a^i \sim \text{softmax}(l^i)$. To ensure the gradient, we use *Gumbel softmax* (Jang et al., 2017) for approximation, which provides a differentiable continuous relaxation for discrete sampling:

$$p(a_j^i|l^i) = \frac{\exp((l_j^i + g_j)/\tau)}{\sum_{k=1}^{m} \exp((l_k^i + g_k)/\tau)}, \tag{34}$$

where $a_j^i$ is the $j$-th action for agent $i$, $g_j = -\log(-\log(x_j))$ represents i.i.d. noise sampled from the Gumbel$(0, 1)$ distribution with $x_j \sim \text{Uniform}(0, 1)$, and $\tau$ is a temperature parameter. This mechanism allows the categorical samples to be approximated by a continuous distribution that converges to a one-hot categorical distribution as $\tau \to 0$. In this case, we typically balance the entropy from the diffusion model and softmax sampling for estimating the true entropy, i.e.,

$$\mathcal{H}(a^i|s) \simeq \log p(a_j^i|l^i) + \beta\ell(\overleftarrow{\pi}^{\text{CD}}(\cdot|h_\psi^i)), \tag{35}$$

where we set $\beta = 1$ for LBF, and $\beta = 0$ for SMAC and SMACv2. Note that we do not use *Action Selection* as for discrete action space, i.e., $K_b = K_t = 1$. At the same time, we do not use distributional-$Q$, as its value is small and more suitable for the direct numerical prediction.

## C. Pseudocode of Dspic

---

**Algorithm 1** Dspic

---

1: **Input:** Polyak coefficient $\tau$, batch size $B$, VAE batch size $B_{\text{VAE}}$, number of agents $n$, episodes $K$, warmup episode $K_{\text{warmup}}$, steps per episode $T$, VAE epoch $e_{\text{VAE}}$, mini-epochs $e$, learning rates $\boldsymbol{\eta} = (\eta_\varphi, \eta_\theta, \eta_\phi, \eta_\psi)$, weight coefficient $\lambda$;

2: **Initialize:** the critic networks $\theta_1$ and $\theta_2$, policy networks $\phi$ and role embedder $\psi$, role embedder $\varphi$, temperature $\alpha_i$ for agent $i$ ($i \in \mathcal{I}$) and $\alpha_{\text{critic}}$ for critic, replay buffer $\mathcal{B}$, Set target parameters equal to main parameters $\bar{\theta}_1 \leftarrow \theta_1, \bar{\theta}_2 \leftarrow \theta_2$, $\bar{\phi} \leftarrow \phi$;

3: **for** $k = 0, 1, ..., K_{\text{warmup}}$ **do**

4:     Observe state $o_t^i$, select action $a_t^i$ randomly and execute it in the environment;

5:     Observe next state $o_{t+1}$, reward $r_t$, then push transitions $\{(o_t^i, a_t^i, o_{t+1}^i, r_t), \forall i \in \mathcal{N}, t \in T\}$ into $\mathcal{B}$;

6: **end for**

7: **for** $l = 0, 1, ..., e_{\text{VAE}}$ **do**

8:     Divide transitions in $\mathcal{B}$ into $M$ batches with batch size $B_{\text{VAE}}$ randomly;

9:     **for** $d = 0, 1, ..., |\mathcal{B}|/B_{\text{VAE}}$ **do**

10:         Update VAE with the $d$-th batch in $\mathcal{B}$

$$\varphi \leftarrow \varphi - \eta_\varphi \nabla_\varphi \mathcal{L}_{\text{PO-VAE}}(\varphi); \quad \text{(With Eqn. 29)}$$

11:     **end for**

12: **end for**

13: Generate $\{z_i\}_{i=1}^n$ for all agents by VAE Encoder;

14: **for** $k = K_{\text{warmup}}, K_{\text{warmup}} + 1, \ldots, K - 1$ **do**

15:     Observe state $o_t^i$ and select action $a_t^i \sim \bar{\boldsymbol{\pi}}_\phi^{\text{CD}}(\cdot|h_{\psi,t}^i)$ with $h_{\psi,t}^i = (g_\psi(z_i)o_t^i, o_t^i)$;

16:     execute $a_t^i$ in the environment;

17:     Observe next state $o_{t+1}$, reward $r_t$;

18:     Push transitions $\{(o_t^i, a_t^i, o_{t+1}^i, r_t), \forall i \in \mathcal{N}, t \in T\}$ into $\mathcal{B}$;

19:     Sample a random minibatch of $B$ transitions from $\mathcal{B}$;

20:     Compute the critic targets

$$y_t = r_t + \gamma \left[ \min_{i=1,2} Q_{\bar{\theta}_i}(s_{t+1}, \boldsymbol{a}_{t+1}) - \alpha_{\text{critic}} \ell \left( \bar{\boldsymbol{\pi}}_{\bar{\phi}}^{\text{CD}}(\boldsymbol{a}_{t+1}|s_{t+1}) \right) \right],$$

$$\text{where} \quad \boldsymbol{a}_{t+1} \sim \bar{\boldsymbol{\pi}}_{\bar{\phi}}^{\text{CD}}(\cdot|s_{t+1});$$

21:     Compute TD error as loss

$$\mathcal{L}(\theta_i) = \frac{1}{2B} \sum \left[ Q_{\theta_i}(s_t, \boldsymbol{a}_t) - y_t \right]^2 \quad \theta_i \leftarrow \theta_i - \eta_\theta \nabla_{\theta_i} \mathcal{L}(\theta_i) \quad \text{for } i = 1, 2;$$

22:     Draw a permutation of agents $i_{1:n}$ at random;

23:     **for** agent $i_m = i_1, \ldots, i_n$ **do**

24:         Compute $\mathcal{L}(\phi, \psi)$ as for agent $i_m$, then update

$$\phi \leftarrow \phi - \eta_\phi \nabla_\phi \mathcal{L}(\phi, \psi), \psi \leftarrow \psi - \eta_\psi \nabla_\psi \mathcal{L}(\phi, \psi); \quad \text{(With Eqn. 17)}$$

25:         Update temperature $\alpha_{i_m}$ for agent $i_m$ with $\mathcal{L}(\alpha)$ Eqn. 18;

26:         with $e$ mini-epochs of policy gradient ascent;

27:     **end for**

28:     Update temperature $\alpha_{\text{critic}}$ for critic with $\mathcal{L}(\alpha)$ Eqn. 18;

29:     Update the target critic network smoothly

$$\bar{\phi} \leftarrow \tau\bar{\phi} + (1 - \tau)\phi, \quad \bar{\theta}_i \leftarrow \tau\bar{\theta}_i + (1 - \tau)\theta_i \quad \text{for } i = 1, 2;$$

30: **end for**

---

# D. Additional Experiment

## D.1. Environment Settings

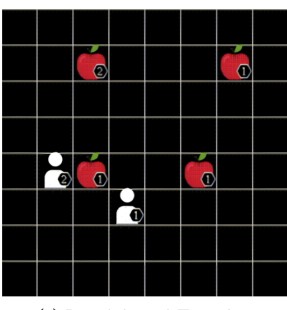 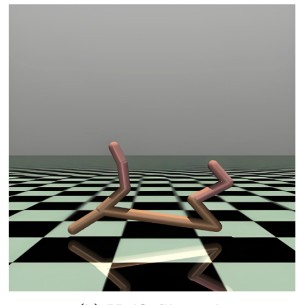 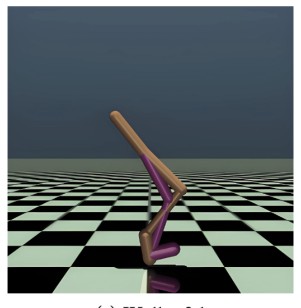 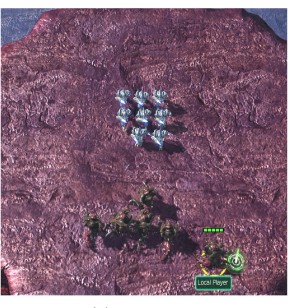

| (a) Level−based Foraging | (b) Half−Cheetah | (c) Walker2d | (d) 6h_vs_8z |

*Figure 6.* Environment demonstration. We show a selection of tasks from our chosen benchmarks. Specifically, in LBF, we selected the 2p4f task (a); in MaMuJoCo, we selected the HalfCheetah (b) and Walker2d (c) tasks; and in SMAC, we selected the 6h_vs_8z (d) task.

| Training Environment | Observation Space Dim. | State Space Dim. | Action Space Dim. | num. agents |
|---|---|---|---|---|
| HalfCheetah-v4 2x3 | 19 | 17 | 3 | 2 |
| HalfCheetah-v4 3x2 | 20 | 17 | 2 | 3 |
| HalfCheetah-v4 6x1 | 23 | 17 | 1 | 6 |
| Walker2d-v4 2x3 | 19 | 17 | 3 | 2 |
| Walker2d-v4 6x1 | 23 | 17 | 1 | 6 |
| Ant-v4 2x4 | 29 | 27 | 4 | 2 |
| Ant-v4 4x2 | 32 | 27 | 2 | 4 |
| Ant-v4 8x1 | 35 | 27 | 1 | 8 |
| HumanoidStandup-v4 9\|8 | 378 | 376 | [9,8] | 2 |
| 3s5z | 252 | 299 | 14 | 8 |
| 3s5z_vs_3s6z | 268 | 318 | 15 | 8 |
| 6h_vs_8z | 172 | 213 | 14 | 6 |
| MMM2 | 370 | 435 | 18 | 10 |
| terran_5_vs_5 | 82 | 120 | 11 | 5 |
| protoss_10_vs_11 | 191 | 327 | 17 | 10 |
| LBF2p2f | 12 | 12 | 6 | 2 |
| LBF2p3f | 15 | 15 | 6 | 2 |

*Table 1.* Action space dimension, state space dimension, observation space dimension and the number of agents for various training environments

**Level-based Foraging**  Level-based foraging (LBF) (Christianos et al., 2020) is a partially observable grid world game, where agents and foods are initialized with random skill levels. The action space of each agent consists of the movement in four directions, loading food and a "none" action. A group of agents can collect the food if they all choose the loading food action and the summation of their levels is greater than or equal to the level of the food. Then agents will receive a reward correlated to the level of the food. The goal of agents is to maximize the global return in a limited horizon, and the maximized return is normalized to one.

**Multi-agent MuJoCo**  Multi-Agent MuJoCo (MaMuJoCo) (de Witt et al., 2020; Peng et al., 2021) is a comprehensive benchmark suite designed for cooperative multi-agent reinforcement learning (MARL) in continuous action spaces. Built upon the standard single-agent MuJoCo robotic control framework, MaMuJoCo transforms classic locomotion tasks into multi-agent challenges by partitioning a single robot's body joints and segments into distinct sub-graphs, with each sub-graph controlled by an independent agent. This setup inherently models decentralized partially observable Markov decision

processes where agents must coordinate their high-dimensional continuous actions based on local observations to achieve a global objective (e.g., walking or swimming). With its configurable agent partitions and scalable scenarios such as ManyAgent Ant and Swimmer, MaMuJoCo provides a more complex and physically realistic platform for evaluating coordination and scalability in continuous MARL compared to traditional toy environments. And all our experiments are conducted on MaMuJuCo-v4.

**StarCraft Multi-Agent Challenge and SMACv2**   The StarCraft Multi-Agent Challenge (SMAC) (Samvelyan et al., 2019) is a widely-used benchmark for cooperative multi-agent reinforcement learning (MARL). Based on the popular real-time policy game StarCraft II, SMAC focuses specifically on decentralized micromanagement tasks where each allied unit is controlled by an independent agent. These agents operate under partial observability, meaning they can only perceive information within a limited circular sight range. The environment require agents to learn complex coordinated behaviors—such as focus fire and kiting—to defeat an opposing army controlled by the game's built-in scripted AI. SMACv2 (Ellis et al., 2023) is an extension based on SMAC, offering more diverse environments but also presenting a greater learning challenge.

To illustrate the game's mechanics, we will show action space dimension, state space dimension, observation space dimension and the number of agents for different tasks in Tab. 1. (For the discrete action space, we present the number of selectable actions as the dimension.)

## D.2. More Comparative Experiments

We have supplemented this section with more comprehensive comparative experiments, including five MaMuJoCo tasks, two SMAC tasks, one SMACv2 task, and one LBF task. The experimental results are shown in Fig. 7. These experiments further corroborate our analysis in Sec. 5.2, demonstrating the excellent performance of our algorithm.

Similar to Sec 5.2, Dspic achieves SOTA or better performance on all tasks, including MaMuJoCo, SMAC, and LBF, demonstrating excellent sample efficiency. Particularly in the Half-Cheetah 3x2 task, Ant 2x4 task and LBF task, our algorithm demonstrates a significant improvement in sample efficiency compared to other baselines. On the other hand, although Kaleidoscope achieves impressive performance on the SMACv2 protoss_10_vs_11 task, Dspic still surpasses the baseline for learning explicit policy functions. Furthermore, our algorithm exhibits excellent stability across all tasks.

We simultaneously compared the performance with role-based baseline ROMA (Wang et al., 2020a) and RODE (Wang et al., 2021b), as shown in the Tab. 2.

|  | Dspic (ours) | ROMA | RODE |
|---|---|---|---|
| 6h_vs_8z | $\mathbf{19.04 \pm 0.65}$ | $15.00 \pm 1.44$ | $18.82 \pm 0.28$ |
| MMM2 | $\mathbf{19.51 \pm 0.20}$ | $15.23 \pm 0.97$ | $19.48 \pm 0.76$ |
| terran_5_vs_5 | $\mathbf{15.06 \pm 1.63}$ | $14.58 \pm 1.54$ | $5.64 \pm 0.40$ |
| LBF2p3f | $\mathbf{0.97 \pm 0.03}$ | $0.59 \pm 0.03$ | $0.47 \pm 0.23$ |

*Table 2.* Performance comparison of our algorithm with ROMA and RODE. We recorded the mean and variance of convergence performance across three experiments, and bolded the best performance.

## D.3. More Ablation Studies

**Hyperparameter Influence.** We investigate the effects of four hyperparameters: temperature parameter $\alpha$, diffusion step number `diff_step`, projection dimension $d'$, and weight coefficient $\lambda$. These parameters control the exploration-exploitation trade-off, the accuracy of the diffusion model, the information provided by the projected observation space, and the trade-off between the learning role embedder and the policy network. The first two were experimented on in Half-Cheetah 2x3, while the latter was conducted in Half-Cheetah 6x1 because it was more challenging to distinguish agents. We examined the impact of different hyperparameter values on the final round's game reward, and the results are shown in Fig. 8.

Experimental results show that too high $\alpha$ (i.e. 2) will cause the agent to continuously explore without learning, while too small $\alpha$ (i.e. 2e-4) will cause it to converge to a suboptimal policy. Furthermore, the method of automatically adopting $\alpha$ achieves optimal performance, which is the setting we recommend. Regarding $\lambda$, we found that our algorithm is not very sensitive to it, providing a wide range of usability (i.e. $5 \sim 500$). Of course, too large $\lambda$ value (i.e. 5000) will still cause it to converge prematurely to a suboptimal solution, while too small $\lambda$ value (i.e. 0.5), although having similar sample efficiency

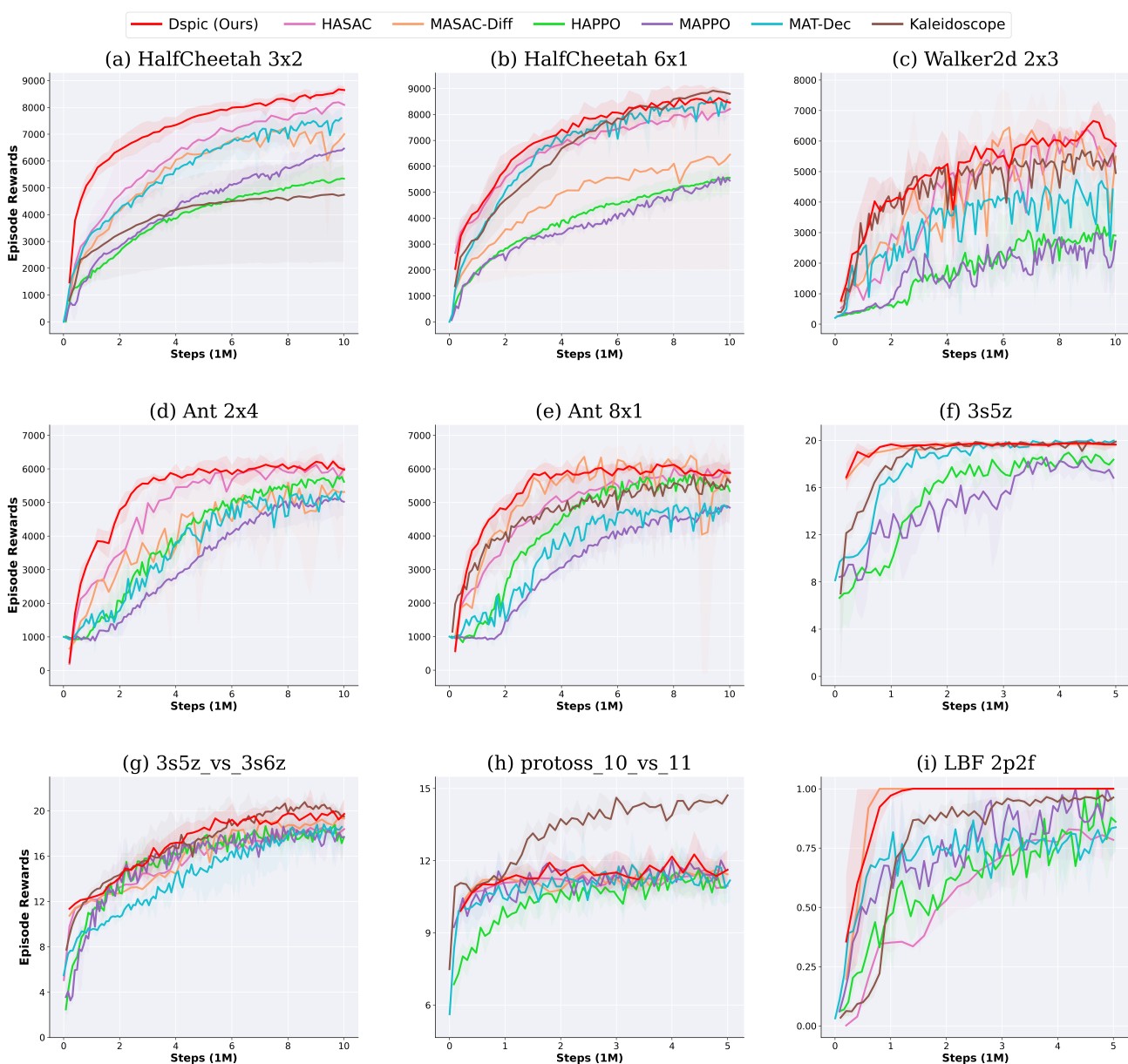

*Figure 7.* Additional comparative experiments. We conduct more comparative experiments on 5 MaMuJoCo tasks, 2 SMAC tasks, 1 SMACv2 task, and 1 LBF task. The curves show the average performance under 3 random seeds, and shade the area corresponding to the 95% confidence interval of our experiments.

to a suitable value in the early stages, hinders policy learning and convergence due to its slow learning speed. Too small `diff_step` (i.e. 2 and 4) can prevent the learning of better policies due to limitations in expressive power. We generally choose `diff_step` = 8 ∼ 16 for different environments, because too large `diff_step` (i.e. 32) leads to unnecessary overhead. The same applies to $d'$: too small value (i.e. 2) can make it difficult to distinguish agents, while too high value (i.e. 64) leads to an unnecessary increase in model size and also results in a slight loss of sample efficiency. Therefore, we recommend setting $d' = 32$ for all the tasks we tested.

**Effect of Denoising Network with Attention.** To test the effect of the attention gate we use in the denoising network as B.3, we conduct an experiment in HalfCheetah 2x3 and Ant 4x2. As shown in Fig. 9 (a) and (b), not using attention may still cause the network to have difficulty capturing character information, resulting in unstable training in both tasks. However, using an attention gate solves this problem.

**Effect of Distributional-$Q$.** To test the effect of Distributional-$Q$ mentioned in App. B.2, we conduct an experiment in HalfCheetah 2x3 and Ant 4x2. As shown in Fig. 9 (c) and (d), using Distributional-$Q$ further improves model performance on both two tasks, and is more pronounced on the Ant 4x2 task. This improvement stems from the fact that these tasks typically have large $Q$ values (generally exceeding 500), meaning that traditional $Q$-networks often have large TD errors. In contrast, Distributional-$Q$ is unaffected by orders of magnitude and is more robust to environmental settings. This also explains why we chose a more lightweight direct prediction method in SMAC tasks, where $Q$-values are generally small (usually not more than 10).

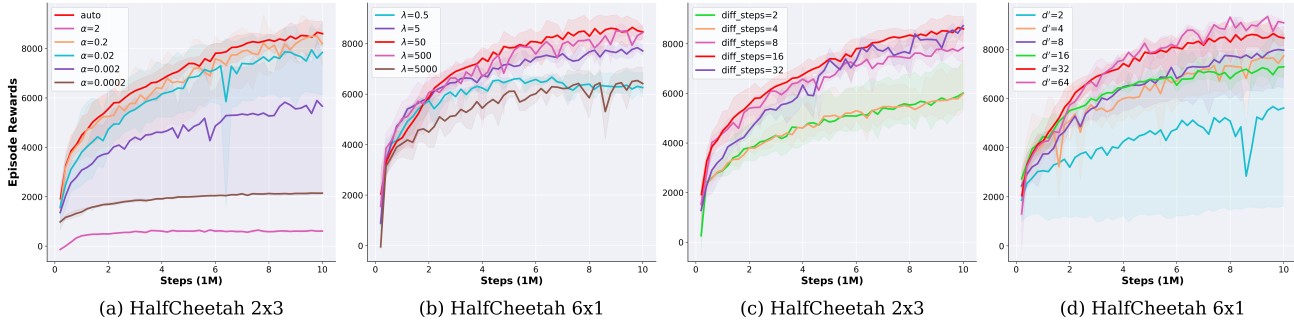

|     |     |     |     |
| --- | --- | --- | --- |
| (a) HalfCheetah 2x3 | (b) HalfCheetah 6x1 | (c) HalfCheetah 2x3 | (d) HalfCheetah 6x1 |

*Figure 8.* Hyperparameter impact experiments. For different hyperparameter settings, we tested them with 3 random seeds in the corresponding environment. Specifically, we tested the impact of the hyperparameter $\alpha$, $\lambda$, `diff_step` and $d'$ on the algorithm's performance.

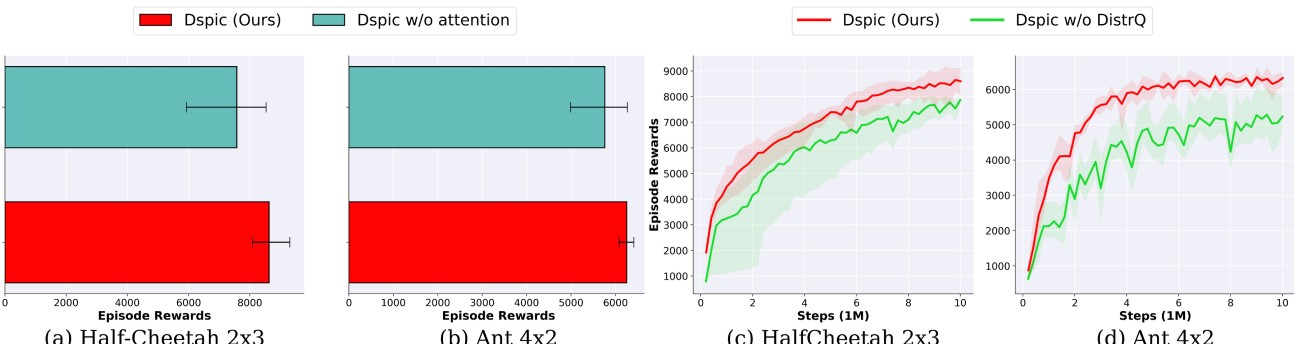

|     |     |     |     |
| --- | --- | --- | --- |
| (a) Half-Cheetah 2x3 | (b) Ant 4x2 | (c) HalfCheetah 2x3 | (d) Ant 4x2 |

*Figure 9.* Additional ablation experiments. (a) and (b) show the ablation experiments on the attention gate on HalfCheetah 2x3 and Ant 4x2 respectively. To demonstrate stability, we use bar charts to show the average asymptotic performance and 95% confidence interval under 3 randomized seeds. (c) and (d) show the ablation with and without Distributional-$Q$. We also show the average performance curves and 95% confidence interval under 3 randomized seeds on HalfCheetah 2x3 and Ant 4x2.

# E. Time Analysis

A key concern regarding our method is that the use of Diffusion introduces more time consumption. To address this, we present the average time taken to train or inference with HASAC and Dspic for 1M timesteps across all benchmarks and show them in the Tab. 3. All testing was conducted on a single NVIDIA RTX 4090 GPU.

| Environment | Train Wall-clock of HASAC | Train Wall-clock of Dspic | Inference Wall-clock of HASAC | Inference Wall-clock of Dspic |
| --- | --- | --- | --- | --- |
| MaMuJoCo-v4 | 41.5 min | 82.2 min (1.98 ×) | 281.7s | 347.8s (1.23 ×) |
| SMAC | 108.8 min | 140.5 min (1.29 ×) | 1542.3s | 1670.2s (1.08×) |
| SMACv2 | 95.5 min | 150.5 min (1.58 ×) | 2846.0s | 3278.6s (1.15 ×) |
| LBF | 34.0 min | 43.0 min (1.26 ×) | 266.2s | 317.1s (1.19 ×) |
| Overall | 62.7 min | 99.3 min (1.58 ×) | 878.2s | 1000.1s (1.14×) |

*Table 3.* Comparison of training time and inference time for 1M timesteps between HASAC and Dspic.

As shown in Tab. 3 , while the use of diffusion-based policies entails an inherent computational overhead, this increase is manageable. Specifically, the training efficiency does not exceed twice that of HASAC, while the inference efficiency does not exceed 1.25 times. Despite the time disadvantage, considering that our algorithm has shown better performance than it on all baselines, we believe it has achieved a **trade-off** between efficiency and effectiveness

## F. List of Hyperparameters

Please note that we have used the official code releases of the respective baseline methods for training. For parameters not listed in the baseline, we use the default values from their reference implementation. Although **actor lr** is environment-related in some baselines, we still display it in the common parameter table using ranges, and show it in detail in Tab. 7. All the baselines we selected are open source.

| | **Dspic** | **HASAC** | **MASAC-Diff** | **HAPPO** | **MAPPO** | **MAT-Dec** | **Kaleidoscope** |
|---|---|---|---|---|---|---|---|
| actor lr | [3e-4, 1e-3] | [3e-4, 1e-3] | 5e-4 | 5e-4 | 5e-4 | 5e-4 | [3e-4, 1e-3] |
| critic lr | 2.4e-3 | 1e-3 | 2.4e-3 | 5e-4 | 5e-4 | 5e-4 | 5e-4 |
| batch size | 1000 | 1000 | 1000 | N/A | N/A | N/A | 1000 |
| buffer size | 1e6 | 1e6 | 1e6 | N/A | N/A | N/A | 1e6 |
| $\tilde{\mathcal{H}}$ | $-2\dim(\mathcal{A})$ | $-\dim(\mathcal{A})$ | $-2\dim(\mathcal{A})$ | N/A | N/A | N/A | N/A |
| alpha lr | 3e-4 | 3e-4 | 3e-4 | N/A | N/A | N/A | N/A |
| critic hidden size | 2048 | 256 | 2048 | 128 | 128 | 64 | 256 |
| actor hidden size | 256 | 256 | 256 | 128 | 128 | 64 | 256 |
| num. critic atoms | 150 | N/A | 150 | N/A | N/A | N/A | N/A |
| optimizer | Adam | Adam | Adam | Adam | Adam | Adam | Adam |
| warmup steps | 1e4 | 1e4 | 1e4 | N/A | N/A | N/A | 1e4 |
| policy frequency | 1 | 2 | 1 | N/A | N/A | N/A | 2 |
| ppo epoch | N/A | N/A | N/A | 5 | 5 | 5 | N/A |
| ppo clip | N/A | N/A | N/A | 0.1 | 0.1 | 0.1 | N/A |

*Table 4.* Hyperparameter settings for all algorithms in MaMuJoCo-v4.

| | **Dspic** | **HASAC** | **MASAC-Diff** | **HAPPO** | **MAPPO** | **MAT-Dec** | **Kaleidoscope** |
|---|---|---|---|---|---|---|---|
| actor lr | 3e-4 | 3e-4 | 3e-4 | 3e-4 | 3e-4 | 3e-4 | 3e-4 |
| critic lr | 5e-4 | 5e-4 | 5e-4 | 5e-4 | 5e-4 | 5e-4 | 5e-4 |
| batch size | 1000 | 1000 | 1000 | N/A | N/A | N/A | 128 |
| buffer size | 1e6 | 1e6 | 1e6 | N/A | N/A | N/A | 5e5 |
| $\tilde{\mathcal{H}}$ | $0.08\|\mathcal{A}\|$ | $0.98\dim(\mathcal{A})$ | $0.08\|\mathcal{A}\|$ | N/A | N/A | N/A | N/A |
| alpha lr | 3e-4 | 3e-4 | 3e-4 | N/A | N/A | N/A | N/A |
| critic hidden size | 2048 | 256 | 2048 | 128 | 128 | 64 | 256 |
| actor hidden size | 256 | 256 | 256 | 128 | 128 | 64 | 256 |
| optimizer | Adam | Adam | Adam | Adam | Adam | Adam | Adam |
| warmup steps | 1e4 | 1e4 | 1e4 | N/A | N/A | N/A | 1e4 |
| policy frequency | 1 | 1 | 1 | N/A | N/A | N/A | N/A |
| ppo epoch | N/A | N/A | N/A | 5 | 5 | 5 | N/A |
| ppo clip | N/A | N/A | N/A | 0.2 | 0.2 | 0.05 | N/A |

*Table 5.* Hyperparameter settings for all algorithms in SMAC and SMACv2.

| | Dspic | HASAC | MASAC-Diff | HAPPO | MAPPO | MAT-Dec | Kaleidoscope |
|---|---|---|---|---|---|---|---|
| actor lr | 5e-4 | 5e-4 | 5e-4 | 5e-4 | 5e-4 | 5e-4 | 5e-4 |
| critic lr | 5e-4 | 5e-4 | 5e-4 | 5e-4 | 5e-4 | 5e-4 | 5e-4 |
| batch size | 128 | 128 | 128 | N/A | N/A | N/A | 128 |
| buffer size | 1e6 | 1e6 | 1e6 | N/A | N/A | N/A | 5e5 |
| $\widetilde{\mathcal{H}}$ | $0.08|\mathcal{A}|$ | $0.98\dim(\mathcal{A})$ | $0.08|\mathcal{A}|$ | N/A | N/A | N/A | N/A |
| alpha lr | 3e-4 | 3e-4 | 3e-4 | N/A | N/A | N/A | N/A |
| critic hidden size | 2048 | 256 | 2048 | 128 | 128 | 64 | 256 |
| actor hidden size | 256 | 256 | 256 | 128 | 128 | 64 | 256 |
| optimizer | Adam | Adam | Adam | Adam | Adam | Adam | Adam |
| warmup steps | 1e4 | 1e4 | 1e4 | N/A | N/A | N/A | 1e4 |
| policy frequency | 1 | 1 | 1 | N/A | N/A | N/A | N/A |
| ppo epoch | N/A | N/A | N/A | 5 | 5 | 5 | N/A |
| ppo clip | N/A | N/A | N/A | 0.2 | 0.2 | 0.05 | N/A |

*Table 6.* Hyperparameter settings for all algorithms in LBF.

| | actor lr | n_step | $\alpha_{\text{init}}$ | $\gamma$ | $v_{\text{max}}$ | $v_{\text{min}}$ | $\epsilon_0$ | $t_0$ | diff_step | $d''$ | $\lambda$ |
|---|---|---|---|---|---|---|---|---|---|---|---|
| HalfCheetah-v4 | 3e-4 | 10 | 0.2 | 0.99 | 1000 | 0 | 1e-4 | 1e6 | 16 | 32 | 50 |
| Walker2d-v4 | 5e-4 | 20 | 0.2 | 0.99 | 800 | 0 | 1e-4 | 1e6 | 16 | 32 | 50 |
| Ant-v4 | 3e-4 | 5 | 0.2 | 0.99 | 800 | 0 | 1e-4 | 1e6 | 16 | 32 | 50 |
| HumanoidStandup-v4 | 1e-3 | 10 | 0.2 | 0.99 | 16000 | 0 | 1e-2 | 1e6 | 16 | 32 | 50 |
| 3s5z | 3e-4 | 20 | 0.01 | 0.99 | N/A | N/A | 1e-4 | 1e6 | 16 | 32 | 50 |
| 6h_vs_8z | 3e-4 | 5 | 0.01 | 0.99 | N/A | N/A | 1e-4 | 1e6 | 8 | 32 | 50 |
| 3s5z_vs_3s6z | 3e-4 | 10 | 0.01 | 0.99 | N/A | N/A | 1e-4 | 1e6 | 16 | 32 | 50 |
| MMM2 | 3e-4 | 5 | 0.01 | 0.99 | N/A | N/A | 1e-4 | 1e6 | 10 | 32 | 50 |
| terran_5_vs_5 | 3e-4 | 10 | 0.02 | 0.95 | N/A | N/A | 1e-4 | 1e6 | 8 | 32 | 50 |
| protoss_10_vs_11 | 3e-4 | 10 | 0.02 | 0.95 | N/A | N/A | 1e-4 | 1e6 | 10 | 32 | 50 |
| LBF | 5e-4 | 5 | 0.1 | 0.99 | N/A | N/A | 1e-4 | 1e6 | 16 | 32 | 50 |

*Table 7.* Environment-related hyperparameters for Dspic, HASAC and MASAC-Diff. Note that **actor lr**, **n_step** and **gamma** also apply to Kaleidoscope.

# G. Derivation of Dspic

## G.1. Policy Deviation of Shared Policy

Firstly, we first transform the problem using the following proposition.

**Proposition G.1** (**Joint KL Divergence Decomposition**). *For two arbitrary policies under CTDE setting, denoted as* $\boldsymbol{\pi}_\circ(\boldsymbol{a}|s) = \prod_{i=1}^n \pi_\circ^i(a^i|s)$ *and* $\boldsymbol{\pi}_\bullet(\boldsymbol{a}|s) = \prod_{i=1}^n \pi_\bullet^i(a^i|s)$, *we have*

$$D_{KL}\left(\boldsymbol{\pi}_\circ(\boldsymbol{a}|s) \| \boldsymbol{\pi}_\bullet(\boldsymbol{a}|s)\right) = \sum_{i=1}^n D_{KL}\left(\pi_\circ^i(\boldsymbol{a}|s) \| \pi_\bullet^i(\boldsymbol{a}|s)\right). \tag{36}$$

*Proof.*

$$\begin{aligned}
D_{\mathrm{KL}}(\boldsymbol{\pi}_\circ(\cdot \mid s) \| \boldsymbol{\pi}_\bullet(\cdot \mid s)) &= \int_{\boldsymbol{a}} \boldsymbol{\pi}_\circ(\boldsymbol{a} \mid s) \log \frac{\prod_{i=1}^n \pi_\circ^i(a^i|s)}{\prod_{i=1}^n \pi_\bullet^i(a^i|s)} \mathrm{d}\boldsymbol{a} \\
&= \int_{\boldsymbol{a}} \prod_{i=1}^n \pi_\circ^i(a^i|s) \left(\sum_{i=1}^n \log \frac{\pi_\circ^i(a^i|s)}{\pi_\bullet^i(a^i|s)}\right) \mathrm{d}\boldsymbol{a} \\
&= \sum_{i=1}^n \int_{\boldsymbol{a}} \boldsymbol{\pi}_\circ^{-i}(\boldsymbol{a}^{-i}|s) \left(\pi_\circ^i(a^i|s) \log \frac{\pi_\circ^i(a^i|s)}{\pi_\bullet^i(a^i|s)}\right) \mathrm{d}\boldsymbol{a} \\
&= \sum_{i=1}^n \int_{a^i} \pi_\circ^i(a^i|s) \log \frac{\pi_\circ^i(a^i|s)}{\pi_\bullet^i(a^i|s)} \mathrm{d}a^i \\
&= \sum_{i=1}^n D_{\mathrm{KL}}\left(\pi_\circ^i(\boldsymbol{a}|s) \| \pi_\bullet^i(\boldsymbol{a}|s)\right),
\end{aligned}$$

which finishes the proof. $\square$

Next, we will consider the two cases separately.

**Proposition G.2** (**QRE Representation**). *The QRE policy, i.e.,* $\boldsymbol{\pi}_\star^{HA}$ *is given by*

$$\forall i \in \mathcal{I}, \pi_\star^i\left(a^i|s\right) = \frac{\exp\left(\alpha^{-1}\mathbb{E}_{\boldsymbol{a}^{-i}\sim\boldsymbol{\pi}_\star^{-i}}\left[Q_{\boldsymbol{\pi}_\star^{HA}}\left(s, a^i, \boldsymbol{a}^{-i}\right)\right]\right)}{\sum_{b^i\in\mathcal{A}^i}\exp\left(\alpha^{-1}\mathbb{E}_{\boldsymbol{a}^{-i}\sim\boldsymbol{\pi}_\star^{-i}}\left[Q_{\boldsymbol{\pi}_\star^{HA}}\left(s, b^i, \boldsymbol{a}^{-i}\right)\right]\right)}. \tag{37}$$

*Proof.* (The proof is cited from (Liu et al., 2024)) Considering the definition of maximum entropy, for an agent $i$, given any $s \in \mathcal{S}$, we study the optimality of $\pi^i(\cdot \mid s)$. The optimization problem can be written as:

$$\begin{aligned}
\max_{\pi^i(\cdot|s)} \quad & \mathbb{E}_{a^i\sim\pi^i(\cdot|s),\boldsymbol{a}^{-i}\sim\boldsymbol{\pi}^{-i}(\cdot|s)}\left[Q_{\boldsymbol{\pi}}(s,\boldsymbol{a})\right] - \alpha \sum_{j=1}^n \sum_{a^j\in\mathcal{A}^j} \pi^j\left(a^j|s\right) \log \pi^j\left(a^j|s\right), \\
\text{s.t.} \quad & \sum_{a^i\in\mathcal{A}^i} \pi^i\left(a^i|s\right) = 1.
\end{aligned}$$

Next, we will write out the Lagrangian function corresponding to this optimization problem:

$$\begin{aligned}
\mathcal{L}\left(\pi^i, \lambda\right) &= \mathbb{E}_{a^i\sim\pi^i(\cdot|s),\boldsymbol{a}^{-i}\sim\boldsymbol{\pi}^{-i}(\cdot|s)}\left[Q_{\boldsymbol{\pi}}(s,\boldsymbol{a})\right] - \alpha \sum_{j=1}^n \sum_{a^j\in\mathcal{A}^j} \pi^j\left(a^j|s\right) \log \pi^j\left(a^j|s\right) + \lambda\left(\sum_{a^i\in\mathcal{A}^i} \pi^i\left(a^i|s\right) - 1\right) \\
&= \sum_{\boldsymbol{a}\in\boldsymbol{\mathcal{A}}} \prod_{j=1}^n \pi^j\left(a^j|s\right) Q_{\boldsymbol{\pi}}(s,\boldsymbol{a}) - \alpha \sum_{j=1}^n \sum_{a^j\in\mathcal{A}^j} \pi^j\left(a^j|s\right) \log \pi^j\left(a^j|s\right) + \lambda\left(\sum_{a^i\in\mathcal{A}^i} \pi^i\left(a^i|s\right) - 1\right).
\end{aligned}$$

To find its extrema with respect to $\pi^i\left(a^i|s\right)$, we differentiate $\mathcal{L}\left(\pi^i, \lambda\right)$,

$$\frac{\partial \mathcal{L}\left(\pi^i, \lambda\right)}{\partial \pi^i\left(a^i|s\right)} = \sum_{\boldsymbol{a}^{-i}\in\boldsymbol{\mathcal{A}}^{-i}} \prod_{j\neq i} \pi^j\left(a^j|s\right) Q_{\boldsymbol{\pi}}\left(s, a^i, \boldsymbol{a}^{-i}\right) - \alpha \log \pi^i\left(a^i|s\right) - \alpha + \lambda.$$

Let $\frac{\partial \mathcal{L}\left(\pi^i, \lambda\right)}{\partial \pi^i\left(a^i|s\right)} = 0$, then we can obtain the analytical solution of $\pi^i_\star(\cdot \mid s)$, i.e.,

$$\alpha \log \pi^i\left(a^i|s\right) = \sum_{\boldsymbol{a}^{-i}\in\boldsymbol{\mathcal{A}}^{-i}} \prod_{j\neq i} \pi^j\left(a^j|s\right) Q_{\boldsymbol{\pi}}\left(s, a^i, \boldsymbol{a}^{-i}\right) - \alpha + \lambda$$

$$\pi^i_\star\left(a^i|s\right) = \exp\left(\alpha^{-1}\mathbb{E}_{\boldsymbol{a}^{-i}\sim\boldsymbol{\pi}^{-i}}\left[Q_{\boldsymbol{\pi}}\left(s, a^i, \boldsymbol{a}^{-i}\right)\right]\right)\exp\left(\frac{\lambda}{\alpha}-1\right).$$

Noting that $\sum_{a^i\in\mathcal{A}^i} \pi^i_\star\left(a^i|s\right) = 1$, we can obtain that the optimal Lagrangian multiplier $\lambda_\star$ satisfies that

$$\exp\left(1 - \frac{\lambda_\star}{\alpha}\right) = \sum_{a^i\in\mathcal{A}^i} \exp\left(\alpha^{-1}\mathbb{E}_{\boldsymbol{a}^{-i}\sim\boldsymbol{\pi}^{-i}}\left[Q_{\boldsymbol{\pi}}\left(s, a^i, \boldsymbol{a}^{-i}\right)\right]\right),$$

so we can get

$$\lambda_\star = \alpha\left(1 - \log \sum_{a^i\in\mathcal{A}^i} \exp\left(\alpha^{-1}\mathbb{E}_{\boldsymbol{a}^{-i}\sim\boldsymbol{\pi}^{-i}}\left[Q_{\boldsymbol{\pi}}\left(s, a^i, \boldsymbol{a}^{-i}\right)\right]\right)\right).$$

Therefore,

$$\pi^i_\star\left(a^i|s\right) = \exp\left(\alpha^{-1}\mathbb{E}_{\boldsymbol{a}^{-i}\sim\boldsymbol{\pi}^{-i}}\left[Q_{\boldsymbol{\pi}}\left(s, a^i, \boldsymbol{a}^{-i}\right)\right]\right)\exp\left(\frac{\lambda_\star}{\alpha}-1\right)$$

$$= \exp\left(\alpha^{-1}\mathbb{E}_{\boldsymbol{a}^{-i}\sim\boldsymbol{\pi}^{-i}}\left[Q_{\boldsymbol{\pi}}\left(s, a^i, \boldsymbol{a}^{-i}\right)\right]\right)\left(\sum_{b^i\in\mathcal{A}^i} \exp\left(\alpha^{-1}\mathbb{E}_{\boldsymbol{a}^{-i}\sim\boldsymbol{\pi}^{-i}}\left[Q_{\boldsymbol{\pi}}\left(s, b^i, \boldsymbol{a}^{-i}\right)\right]\right)\right)^{-1}$$

$$= \frac{\exp\left(\alpha^{-1}\mathbb{E}_{\boldsymbol{a}^{-i}\sim\boldsymbol{\pi}^{-i}}\left[Q_{\boldsymbol{\pi}}\left(s, a^i, \boldsymbol{a}^{-i}\right)\right]\right)}{\sum_{b^i\in\mathcal{A}^i} \exp\left(\alpha^{-1}\mathbb{E}_{\boldsymbol{a}^{-i}\sim\boldsymbol{\pi}^{-i}}\left[Q_{\boldsymbol{\pi}}\left(s, b^i, \boldsymbol{a}^{-i}\right)\right]\right)}.$$

In summary, we see the QRE policy take the following form,

$$\forall i \in \mathcal{N}, \pi^i_\star\left(a^i|s\right) = \frac{\exp\left(\alpha^{-1}\mathbb{E}_{\boldsymbol{a}^{-i}\sim\boldsymbol{\pi}^{-i}_\star}\left[Q_{\boldsymbol{\pi}^{\text{HA}}_\star}\left(s, a^i, \boldsymbol{a}^{-i}\right)\right]\right)}{\sum_{b^i\in\mathcal{A}^i} \exp\left(\alpha^{-1}\mathbb{E}_{\boldsymbol{a}^{-i}\sim\boldsymbol{\pi}^{-i}_\star}\left[Q_{\boldsymbol{\pi}^{\text{HA}}_\star}\left(s, b^i, \boldsymbol{a}^{-i}\right)\right]\right)}.$$

$\square$

Similarly, we can solve for the form of $\boldsymbol{\pi}^{\text{share}}_\star$.

**Proposition G.3 (Optimal Shared Policy).** *The optimal shared policy, i.e., $\boldsymbol{\pi}^{share}_\star$ is given by*

$$\pi_\star\left(a|s\right) = \frac{\exp\left(\frac{1}{n\alpha}M_{\pi_\star}(s, a)\right)}{\sum_{b\in\mathcal{A}} \exp\left(\frac{1}{n\alpha}M_{\pi_\star}(s, b)\right)}, \tag{38}$$

*for any agents with*

$$M_\pi(s, a) = \sum_{k=1}^{n} \mathbb{E}_{\forall j\in[-k], a^j\sim\pi}\left[Q_{share}(s, a^k = a, \boldsymbol{a}^{-k})\right], \tag{39}$$

*where we use $[-k]$ to denote the set of agents other than agent $k$ and $Q_{share}$ to denote the soft Q-function under $\boldsymbol{\pi}^{share}_\star$.*

*Proof.* In shared case, we can directly solve for the global optimum, given any $s \in \mathcal{S}$:

$$\max_{\pi} \quad \mathbb{E}_{\forall j, a^j \sim \pi(\cdot|s)} \left[Q_{\text{share}}(s, \boldsymbol{a})\right] - n\alpha \sum_{a \in \mathcal{A}} \pi(a|s) \log \pi(a|s),$$

$$\text{s.t.} \quad \sum_{a \in \mathcal{A}} \pi(a|s) = 1.$$

We get its Lagrangian function,

$$\mathcal{L}(\pi, \lambda) = \mathbb{E}_{\forall j, a^j \sim \pi(\cdot|s)} \left[Q_{\text{share}}(s, \boldsymbol{a})\right] - n\alpha \sum_{a \in \mathcal{A}} \pi(a|s) \log \pi(a|s) + \lambda \left(\sum_{a \in \mathcal{A}} \pi(a|s) - 1\right)$$

$$= \sum_{\boldsymbol{a} \in \mathcal{A}} \prod_{j=1}^{n} \pi(a^j|s) Q_{\text{share}}(s, \boldsymbol{a}) - n\alpha \sum_{a \in \mathcal{A}} \pi(a|s) \log \pi(a|s) + \lambda \left(\sum_{a \in \mathcal{A}} \pi(a|s) - 1\right).$$

Differentiating $\mathcal{L}(\pi, \lambda)$ with respect to $\pi(a \mid s)$, we get,

$$\frac{\partial \mathcal{L}(\pi, \lambda)}{\partial \pi(a|s)} = M_\pi(s, a) - n\alpha \log \pi(a|s) - n\alpha + \lambda.$$

where,

$$M_\pi(s, a) = \frac{\partial}{\partial \pi(a \mid s)} \sum_{\boldsymbol{a} \in \mathcal{A}} \prod_{j=1}^{n} \pi(a^j \mid s) Q_{\text{share}}(s, \boldsymbol{a}).$$

In fact, we can simplify it,

$$M_\pi(s, a) = \frac{\partial}{\partial \pi(a \mid s)} \sum_{\boldsymbol{a} \in \mathcal{A}} \prod_{j=1}^{n} \pi(a^j \mid s) Q_{\text{share}}(s, \boldsymbol{a})$$

$$= \sum_{\boldsymbol{a} \in \mathcal{A}} Q_{\text{share}}(s, \boldsymbol{a}) \frac{\partial}{\partial \pi(a|s)} \left(\prod_{j=1}^{n} \pi(a^j|s)\right)$$

$$= \sum_{\boldsymbol{a} \in \mathcal{A}} Q_{\text{share}}(s, \boldsymbol{a}) \sum_{k=1}^{n} \left(\frac{\partial \pi(a^k|s)}{\partial \pi(a|s)} \prod_{j \neq k} \pi(a^j|s)\right)$$

Introducing indicator function $\mathbb{I}(\cdot)$

$$= \sum_{\boldsymbol{a} \in \mathcal{A}} Q_{\text{share}}(s, \boldsymbol{a}) \sum_{k=1}^{n} \left(\mathbb{I}(a^k = a) \prod_{j \neq k} \pi(a^j|s)\right)$$

$$= \sum_{k=1}^{n} \sum_{\boldsymbol{a} \in \mathcal{A}} \mathbb{I}(a^k = a) \left(\prod_{j \neq k} \pi(a^j|s)\right) Q_{\text{share}}(s, \boldsymbol{a})$$

$$= \sum_{k=1}^{n} \sum_{\boldsymbol{a}^{-k} \in \mathcal{A}^{-k}} \left(\prod_{j \neq k} \pi(a^j|s)\right) Q_{\text{share}}(s, a^k = a, \boldsymbol{a}^{-k})$$

$$= \sum_{k=1}^{n} \mathbb{E}_{\forall j \in [-k], a^j \sim \pi} \left[Q_{\text{share}}(s, a^k = a, \boldsymbol{a}^{-k})\right].$$

This formula is almost identical to the form used in solving heterogeneous policies, but it involves a process of calculating the expectation cross agents.

$$n\alpha \log \pi\,(a|s) = M_\pi(s,a) - n\alpha + \lambda$$

$$\pi_\star\,(a|s) = \exp\left(\frac{1}{n\alpha}\sum_{k=1}^{n}\mathbb{E}_{\forall j\in[-k],a^j\sim\pi_\star}\left[Q_{\text{share}}(s,a^k = a, \boldsymbol{a}^{-k})\right]\right)\exp\left(\frac{\lambda}{n\alpha} - 1\right).$$

Noting that $\sum_{a\in\mathcal{A}}\pi_\star\,(a|s) = 1$, we can obtain that the optimal Lagrange multiplier $\lambda_\star$ satisfies

$$\exp\left(1 - \frac{\lambda_\star}{n\alpha}\right) = \sum_{a\in\mathcal{A}}\exp\left(\frac{1}{n\alpha}\sum_{k=1}^{n}\mathbb{E}_{\forall j\in[-k],a^j\sim\pi_\star}\left[Q_{\text{share}}(s,a^k = a, \boldsymbol{a}^{-k})\right]\right),$$

so we can get

$$\lambda_\star = n\alpha\left(1 - \log\sum_{a^i\in\mathcal{A}}\exp\left(\frac{1}{n\alpha}\sum_{k=1}^{n}\mathbb{E}_{\forall j\in[-k],a^j\sim\pi_\star}\left[Q_{\text{share}}(s,a^k = a^i, \boldsymbol{a}^{-k})\right]\right)\right).$$

Therefore,

$$\pi_\star\,(a|s) = \frac{\exp\left(\frac{1}{n\alpha}M_{\pi_\star}(s,a)\right)}{\sum_{b\in\mathcal{A}}\exp\left(\frac{1}{n\alpha}M_{\pi_\star}(s,b)\right)}$$

$$= \frac{\exp\left(\frac{1}{n\alpha}\sum_{k=1}^{n}\mathbb{E}_{\forall j\in[-k],a^j\sim\pi_\star}\left[Q_{\text{share}}(s,a^k = a, \boldsymbol{a}^{-k})\right]\right)}{\sum_{b\in\mathcal{A}}\exp\left(\frac{1}{n\alpha}\sum_{k=1}^{n}\mathbb{E}_{\forall j\in[-k],a^j\sim\pi_\star}\left[Q_{\text{share}}(s,a^k = b, \boldsymbol{a}^{-k})\right]\right)}.$$

In summary, we see the optimal shared policy take the following form,

$$\pi_\star\,(a|s) = \frac{\exp\left(\frac{1}{n\alpha}M_{\pi_\star}(s,a)\right)}{\sum_{b\in\mathcal{A}}\exp\left(\frac{1}{n\alpha}M_{\pi_\star}(s,b)\right)}.$$

$\square$

Formally, the analytical solutions of $\pi_\star^{\text{HA}}$ and $\pi_\star^{\text{share}}$ are similar but not equal. We will prove that mathematically and point out the cases where they are equal.

**Proposition G.4** (Deviation of Shared Policy). $D_{KL}(\pi_\star^{HA}(\cdot|s)\|\pi_\star^{share}(\cdot|s)) > 0$ holds **unless** the environment setting is **homogeneous**.

*Proof.* First, if agents are in homogeneous setting, then the optimization objective is

$$J(\pi^i) = \mathbb{E}_{a^i\sim\pi^i(\cdot|s),\boldsymbol{a}^{-i}\sim\boldsymbol{\pi}^{-i}(\cdot|s)}\left[Q_{\boldsymbol{\pi}}(s,\boldsymbol{a})\right] - \alpha\sum_{j=1}^{n}\sum_{a^j\in\mathcal{A}^j}\pi^j\left(a^j|s\right)\log\pi^j\left(a^j|s\right).$$

According to the definition of homogeneous setting, we know that exchanging agent actions only produces constant term differences. This indicates that the QRE policy must satisfy that

$$\pi_\star^1 = \pi_\star^2 = ... = \pi_\star^n.$$

This actually leads to

$$\boldsymbol{\pi}_\star^{\text{HA}}(\cdot\mid s) = \boldsymbol{\pi}_\star^{\text{share}}(\cdot\mid s),$$

i.e.,

$$D_{\text{KL}}(\boldsymbol{\pi}_\star^{\text{HA}}(\cdot\mid s)\|\boldsymbol{\pi}_\star^{\text{share}}(\cdot\mid s)) = 0.$$

However, for a more general environment setting, we demonstrate the policy deviation by contradiction. Let's assume that,

$$D_{\text{KL}}(\boldsymbol{\pi}_\star^{\text{HA}}(\cdot\mid s)\|\boldsymbol{\pi}_\star^{\text{share}}(\cdot\mid s)) = 0.$$

Then we have $Q_{\text{share}}(s, \boldsymbol{a}) = Q_{\boldsymbol{\pi}_\star^{\text{HA}}}(s, \boldsymbol{a})$ for any $s \in \mathcal{S}$ and $a \in \mathcal{A}$. Meanwhile, according to the definition of homogeneous setting, there must exist $\exists i \in \mathcal{I}$, and $a_u, a_v \in \mathcal{A}$ such that

$$\frac{1}{n} \sum_{k=1}^{n} \mathbb{E}_{\forall j \in [-k], a^j \sim \pi_\star} \left[ Q_{\text{share}}(s, a^k = a_u, \boldsymbol{a}^{-k}) \right] - \mathbb{E}_{\boldsymbol{a}^{-i} \sim \boldsymbol{\pi}^{-i}} \left[ Q_{\text{share}} \left( s, a^i = a_u, \boldsymbol{a}^{-i} \right) \right]$$

$$\neq \frac{1}{n} \sum_{k=1}^{n} \mathbb{E}_{\forall j \in [-k], a^j \sim \pi_\star} \left[ Q_{\text{share}}(s, a^k = a_v, \boldsymbol{a}^{-k}) \right] - \mathbb{E}_{\boldsymbol{a}^{-i} \sim \boldsymbol{\pi}^{-i}} \left[ Q_{\text{share}} \left( s, a^i = a_v, \boldsymbol{a}^{-i} \right) \right],$$

i.e.,

$$\frac{1}{n} \sum_{k=1}^{n} \mathbb{E}_{\forall j \in [-k], a^j \sim \pi_\star} \left[ Q_{\text{share}}(s, a^k = a_u, \boldsymbol{a}^{-k}) \right] - \mathbb{E}_{\boldsymbol{a}^{-i} \sim \boldsymbol{\pi}^{-i}} \left[ Q_{\boldsymbol{\pi}} \left( s, a^i = a_u, \boldsymbol{a}^{-i} \right) \right]$$

$$\neq \frac{1}{n} \sum_{k=1}^{n} \mathbb{E}_{\forall j \in [-k], a^j \sim \pi_\star} \left[ Q_{\text{share}}(s, a^k = a_v, \boldsymbol{a}^{-k}) \right] - \mathbb{E}_{\boldsymbol{a}^{-i} \sim \boldsymbol{\pi}^{-i}} \left[ Q_{\boldsymbol{\pi}} \left( s, a^i = a_v, \boldsymbol{a}^{-i} \right) \right].$$

Considering the properties of the Boltzmann distribution with Proposition G.2 and Proposition G.3, we can obtain that,

$$\pi_\star^i(\cdot \mid s) \neq \pi_\star^{\text{share}}(\cdot \mid s),$$

then

$$D_{\text{KL}}(\pi_\star^i(\cdot|s) \| \pi_\star(\cdot|s)) > 0.$$

Follow Proposition G.1,

$$D_{\text{KL}}(\boldsymbol{\pi}_\star^{\text{HA}}(\cdot \mid s) \| \boldsymbol{\pi}_\star^{\text{share}}(\cdot \mid s)) = \sum_{j=1}^{n} D_{\text{KL}}(\pi_\star^j(\cdot|s) \| \pi_\star(\cdot|s))$$

$$\geq D_{\text{KL}}(\pi_\star^i(\cdot|s) \| \pi_\star(\cdot|s))$$

$$> 0.$$

This leads to a contradiction, which finishes the proof.

**Discussion.** Note that the expression for $M_\pi(s, a)$ requires gradient calculation for terms containing the product of shared policies. This results in many higher-order polynomial terms $\pi(a|s)^k$, where $k = 1, 2, \ldots, n$. This means that even after calculating the gradient of $\pi(a|s)$, $M_\pi(s, a)$ still contains terms of $\pi(a|s)^{k-1}$. This causes $M_\pi(s, a)$ to deviate from heterogeneous inputs (or when the agents' inputs are completely disjoint), leading to policy deviation. This shows that policy deviation is caused by input overlap, which also motivates us to design Complete Division.

$\square$

### G.2. Equivalence Optimality with Complete Division

Just like in App. G.1, we discuss the form of QRE policy with CD.

**Proposition G.5** (**QRE Representation with CD**). *Let us consider a **Complete Division** denoted as $\{f_i\}_{i=1}^n$. The QRE policy with it is given by*

$$\forall i \in \mathcal{I}, \pi_\star^{CD}\left(a^i|h^i\right) = \frac{\exp\left(\alpha^{-1}\mathbb{E}_{\boldsymbol{a}^{-i} \sim \boldsymbol{\pi}_{CD}^{-i}}\left[Q_{CD}\left(s, a^i, \boldsymbol{a}^{-i}\right)\right]\right)}{\sum_{b^i \in \mathcal{A}^i} \exp\left(\alpha^{-1}\mathbb{E}_{\boldsymbol{a}^{-i} \sim \boldsymbol{\pi}_{CD}^{-i}}\left[Q_{CD}\left(s, b^i, \boldsymbol{a}^{-i}\right)\right]\right)}, \tag{40}$$

*where $h^i = f_i(s)$ and $Q_{CD}$ is the soft Q-function under $\boldsymbol{\pi}_\star^{CD}$.*

*Proof.* Since $f_i(s)$ is injective, for any $v_i \in \mathcal{V}_i$, there is a unique $s \in \mathcal{S}$ corresponding to it. Therefore, for any given $s \in \mathcal{S}$ and $i \in \mathcal{I}$, we denote $h_i = f_i(s)$, and we can similarly write the corresponding optimization objective like the proof of

Proposition G.2:

$$\max_{\pi^{\text{CD}}(\cdot|h^i)} \quad \mathbb{E}_{a^i \sim \pi^{\text{CD}}(\cdot|h^i), \boldsymbol{a}^{\text{CD}} \sim \boldsymbol{\pi}^{-i}(\cdot|\mathbf{h}^{-i})} \left[ Q_{\text{CD}}(s, \boldsymbol{a}) \right] - \alpha \sum_{j=1}^{n} \sum_{a^j \in \mathcal{A}} \pi^{\text{CD}} \left( a^j | h^j \right) \log \pi^{\text{CD}} \left( a^j | h^j \right),$$

$$\text{s.t.} \quad \sum_{a^i \in \mathcal{A}} \pi^{\text{CD}} \left( a^i | h^i \right) = 1.$$

Then write out the Lagrangian function corresponding to this optimization problem,

$$\mathcal{L} \left( \pi^{\text{CD}}, \lambda \right) = \mathbb{E}_{a^i \sim \pi^{\text{CD}}(\cdot|h^i), \boldsymbol{a}^{\text{CD}} \sim \boldsymbol{\pi}^{-i}(\cdot|\mathbf{h}^{-i})} \left[ Q_{\text{CD}}(s, \boldsymbol{a}) \right] - \alpha \sum_{j=1}^{n} \sum_{a^j \in \mathcal{A}} \pi^{\text{CD}} \left( a^j | h^j \right) \log \pi^{\text{CD}} \left( a^j | h^j \right) + \lambda \left( \sum_{a^i \in \mathcal{A}} \pi^{\text{CD}} \left( a^i | h^i \right) - 1 \right)$$

$$= \sum_{\boldsymbol{a} \in \mathcal{A}} \prod_{j=1}^{n} \pi^{\text{CD}} \left( a^j | h^j \right) Q_{\text{CD}} \left( s, \boldsymbol{a} \right) - \alpha \sum_{j=1}^{n} \sum_{a^j \in \mathcal{A}} \pi^{\text{CD}} \left( a^j | h^j \right) \log \pi^{\text{CD}} \left( a^j | h^j \right) + \lambda \left( \sum_{a^i \in \mathcal{A}} \pi^{\text{CD}} \left( a^i | h^i \right) - 1 \right).$$

We differentiate $\mathcal{L} \left( \pi^{\text{CD}}, \lambda \right)$ w.r.t. $\pi^{\text{CD}}(a^i \mid h^i)$,

$$\frac{\partial \mathcal{L} \left( \pi^{\text{CD}}, \lambda \right)}{\partial \pi^{\text{CD}}(a^i \mid h^i)} = \sum_{\boldsymbol{a}^{-i} \in \mathcal{A}^{-i}} \prod_{j \neq i} \pi^{\text{CD}} \left( a^j | h^j \right) Q_{\text{CD}} \left( s, a^i, \boldsymbol{a}^{-i} \right) - \alpha \log \pi^{\text{CD}} \left( a^i | h^i \right) - \alpha + \lambda.$$

Let $\frac{\partial \mathcal{L} \left( \pi^{\text{CD}}, \lambda \right)}{\partial \pi^{\text{CD}}(a^i|h^i)} = 0$, then we can obtain the analytical solution of $\pi_\star^{\text{CD}}(\cdot \mid h^i)$,

$$\alpha \log \pi^{\text{CD}} \left( a^i | h^i \right) = \sum_{\boldsymbol{a}^{-i} \in \mathcal{A}^{-i}} \prod_{j \neq i} \pi^{\text{CD}} \left( a^j | h^j \right) Q_{\text{CD}} \left( s, a^i, \boldsymbol{a}^{-i} \right) - \alpha + \lambda$$

$$\pi^{\text{CD}} \left( a^i | h^i \right) = \exp \left( \alpha^{-1} \mathbb{E}_{\boldsymbol{a}^{-i} \sim \boldsymbol{\pi}_{\text{CD}}^{-i}} \left[ Q_{\text{CD}} \left( s, a^i, \boldsymbol{a}^{-i} \right) \right] \right) \exp \left( \frac{\lambda}{\alpha} - 1 \right).$$

Noting that $\sum_{a^i \in \mathcal{A}} \pi_\star^{\text{CD}} \left( a^i | h^i \right) = 1$, we can obtain that the optimal Lagrange multiplier $\lambda_\star$ satisfies

$$\exp \left( 1 - \frac{\lambda_\star}{\alpha} \right) = \sum_{a^i \in \mathcal{A}} \exp \left( \alpha^{-1} \mathbb{E}_{\boldsymbol{a}^{-i} \sim \boldsymbol{\pi}_{\text{CD}}^{-i}} \left[ Q_{\text{CD}} \left( s, a^i, \boldsymbol{a}^{-i} \right) \right] \right),$$

where $\boldsymbol{\pi}_{\text{CD}}^{-i}(a^{-i}|s) = \prod_{j \in [-i]} \pi_\star^{\text{CD}}(a^j|h^j)$, so

$$\lambda_\star = \alpha \left( 1 - \log \sum_{a^i \in \mathcal{A}} \exp \left( \alpha^{-1} \mathbb{E}_{\boldsymbol{a}^{-i} \sim \boldsymbol{\pi}^{-i}} \left[ Q_{\text{CD}} \left( s, a^i, \boldsymbol{a}^{-i} \right) \right] \right) \right).$$

Finally, we get that,

$$\pi_\star^{\text{CD}} \left( a^i | h^i \right) = \exp \left( \alpha^{-1} \mathbb{E}_{\boldsymbol{a}^{-i} \sim \boldsymbol{\pi}_{\text{CD}}^{-i}} \left[ Q_{\text{CD}} \left( s, a^i, \boldsymbol{a}^{-i} \right) \right] \right) \exp \left( \frac{\lambda_\star}{\alpha} - 1 \right)$$

$$= \exp \left( \alpha^{-1} \mathbb{E}_{\boldsymbol{a}^{-i} \sim \boldsymbol{\pi}_{\text{CD}}^{-i}} \left[ Q_{\text{CD}} \left( s, a^i, \boldsymbol{a}^{-i} \right) \right] \right) \left( \sum_{b^i \in \mathcal{A}^i} \exp \left( \alpha^{-1} \mathbb{E}_{\boldsymbol{a}^{-i} \sim \boldsymbol{\pi}_{\text{CD}}^{-i}} \left[ Q_{\text{CD}} \left( s, b^i, \boldsymbol{a}^{-i} \right) \right] \right) \right)^{-1}$$

$$= \frac{\exp \left( \alpha^{-1} \mathbb{E}_{\boldsymbol{a}^{-i} \sim \boldsymbol{\pi}_{\text{CD}}^{-i}} \left[ Q_{\text{CD}} \left( s, a^i, \boldsymbol{a}^{-i} \right) \right] \right)}{\sum_{b^i \in \mathcal{A}^i} \exp \left( \alpha^{-1} \mathbb{E}_{\boldsymbol{a}^{-i} \sim \boldsymbol{\pi}_{\text{CD}}^{-i}} \left[ Q_{\text{CD}} \left( s, b^i, \boldsymbol{a}^{-i} \right) \right] \right)}.$$

$\square$

Next, we will prove Theorem 4.3, i.e.,

**Theorem 4.3 (Zero Policy Deviation).** *For $\forall s \in \mathcal{S}$, Equation $D_{KL}(\pi_\star^{HA}(\cdot|s) \| \pi_\star^{CD}(\cdot|s)) = 0$ always holds.*

*Proof.* Considering under the both two cases, the $Q$-function actually has an equivalent structure, i.e.

$$Q(s, \boldsymbol{a}) = r_{\boldsymbol{\pi}}(s, \boldsymbol{a}) + \gamma \mathbb{E}_{s' \sim P, \boldsymbol{a}' \sim \boldsymbol{\pi}} \left[ Q(s', \boldsymbol{a}') \right],$$

$$\boldsymbol{\pi}(\boldsymbol{a}|s) = \prod_{i=1}^{n} \pi^i(a^i|s),$$

$$\pi^i(a^i \mid s) = \frac{\exp \left( \alpha^{-1} \mathbb{E}_{\boldsymbol{a}^{-i} \sim \boldsymbol{\pi}^{-i}} \left[ Q \left( s, a^i, \boldsymbol{a}^{-i} \right) \right] \right)}{\sum_{b^i \in \mathcal{A}^i} \exp \left( \alpha^{-1} \mathbb{E}_{\boldsymbol{a}^{-i} \sim \boldsymbol{\pi}^{-i}} \left[ Q \left( s, b^i, \boldsymbol{a}^{-i} \right) \right] \right)}.$$

This shows that when we discuss any agent $i$, that is, when the policy of agents in $[-i]$ is fixed, Eqn. 37 and Eqn. 40 are completely isomorphic. Thus,

$$\pi_{\star}^i(a^i \mid s) = \pi_{\star}^{\mathrm{CD}}(a^i \mid h^i),$$

i.e.,

$$D_{\mathrm{KL}}(\pi_{\star}^i(\cdot \mid s) \| \pi_{\star}^{\mathrm{CD}}(\cdot \mid h^i)) = 0.$$

Follow Proposition G.1,

$$D_{\mathrm{KL}}(\boldsymbol{\pi}_{\star}^{\mathrm{HA}}(\cdot \mid s) \| \boldsymbol{\pi}_{\star}^{\mathrm{CD}}(\cdot \mid s)) = \sum_{i=1}^{n} D_{\mathrm{KL}}(\pi_{\star}^i (\cdot|s) \| \pi_{\star}^{\mathrm{CD}} (\cdot \mid h^i)) = 0,$$

which finishes the proof. $\qquad \square$

### G.3. Proof of Orthogonal Construction

To provide practical algorithms, we present Proposition 4.4 about orthogonal construction, i.e.,

**Proposition 4.4 (Orthogonal Construction).** *We define a set of matrices $\{\mathbf{P}_i\}_{i=1}^n$, where $\mathbf{P}_i \in \mathbb{R}^{d_s \times d'}$ ($d_s$ is the dimension of the state space $\mathcal{S}$ and $d'$ is the dimension of projection space). If for any $i \neq j$, $\mathbf{P}_i^{\top} \mathbf{P}_j = \mathbf{O}$ holds, then the mapping set $\{f_i(s) = (\mathbf{P}_i s, s)\}_{i=1}^n$ constitutes a Complete Division of space $\mathcal{S}/\{s \mid s \in \mathcal{N}(\mathbf{P}_i^{\top} \mathbf{P}_i) \cap \mathcal{N}(\mathbf{P}_j^{\top} \mathbf{P}_j), \exists i \neq j\}$, where $\mathcal{N}(\mathbf{P}_i^{\top} \mathbf{P}_i) = \{s \mid \mathbf{P}_i^{\top} \mathbf{P}_i s = \mathbf{0}\}$ is the null space of matrix $\mathbf{P}_i^{\top} \mathbf{P}_i$.*

*Proof.* We first prove that $f_i$ is injective. Suppose there exist $s_1 \neq s_2$ such that $f_i(s_1) = f_i(s_2)$. Then, according to

$$(\mathbf{P}_i s_1, s_1) = (\mathbf{P}_i s_2, s_2) \implies s_1 = s_2$$

contradiction occurs, therefore $f_i$ is injective.

On the other hand, suppose there exist $i \neq j$ such that the value spaces $\mathcal{V}_i$ and $\mathcal{V}_j$ of $f_i$ and $f_j$ intersect. Then suppose $\exists (v, s) \in \mathcal{V}_i \cap \mathcal{V}_j$, where $(v, s) \neq \mathbf{0}$, then there must be

$$\mathbf{P}_i s = \mathbf{P}_j s = v,$$

left-multiply it with $\mathbf{P}_i^{\top}$,

$$\mathbf{P}_i^{\top} \mathbf{P}_i s = (\mathbf{P}_i^{\top} \mathbf{P}_j) s = \mathbf{0}.$$

Similarly,

$$\mathbf{P}_j^{\top} \mathbf{P}_j s = \mathbf{0}.$$

Thus,

$$s \in \{s \mid s \in \mathcal{N}(\mathbf{P}_i^{\top} \mathbf{P}_i) \cap \mathcal{N}(\mathbf{P}_j^{\top} \mathbf{P}_j), \forall i \neq j\}$$

This contradicts $s \in \mathcal{S}/\{s \mid s \in \mathcal{N}(\mathbf{P}_i) \cap \mathcal{N}(\mathbf{P}_j), \forall i \neq j\}$, so

$$\mathcal{V}_i \bigcap \mathcal{V}_j = \emptyset \quad \forall i \neq j,$$

which finishes the proof. $\qquad \square$

## G.4. Correctness of Diffusion Soft Policy Iteration with Complete Division

We firstly prove Lemma 4.5, i,e,

**Lemma 4.5 (Joint Soft Diffusion Policy Evaluation with CD).** Consider the soft Bellman backup operator $\mathcal{T}^{\overleftarrow{\bar{\pi}}^{\mathrm{CD}}}$ and function $\hat{Q}_0 : \mathcal{S} \times \mathcal{A} \to \mathbb{R}$ with $|\mathcal{A}| < \infty$, and define $\hat{Q}_{k+1} = \mathcal{T}^{\overleftarrow{\bar{\pi}}^{\mathrm{CD}}}\hat{Q}_k$. Then the sequence $Q_k$ will converge to the joint soft $Q$-function of $\pi$ as $k \to \infty$.

*Proof.* We define the reward with entropy term as $r_{\overleftarrow{\bar{\pi}}^{\mathrm{CD}}}(s, \boldsymbol{a}) \triangleq r(s, \boldsymbol{a}) + \mathbb{E}_{s' \sim P}\left[\alpha \ell\left(\overleftarrow{\bar{\pi}}^{\mathrm{CD}}(\cdot|s)\right)\right]$. We can then express the update rule as:

$$Q(s, \boldsymbol{a}) \leftarrow r_{\overleftarrow{\bar{\pi}}^{\mathrm{CD}}}(s, \boldsymbol{a}) + \gamma \mathbb{E}_{s' \sim P, \boldsymbol{a}' \sim \overleftarrow{\bar{\pi}}^{\mathrm{CD}}}\left[Q(s', \boldsymbol{a}')\right]$$

and apply the standard convergence results for policy evaluation following (Sutton & Barto, 1998). $\qquad\square$

Then in policy improvement phase, we first prove that the joint policy update can be decomposed into multiplication of sequential local diffusion policy updates, as shown in Proposition 4.6.

**Proposition 4.6 (Joint Soft Diffusion Policy Decomposition with CD).** *Let $\vec{\pi}^{CD}$ and $\overleftarrow{\bar{\pi}}^{CD}$ be the forward and backward processes of a joint soft diffusion policy with CD $\{f_i\}_{i=1}^n$, and $i_{1:n} \in Sym(n)$ be a permutation of $\mathcal{I}$. Suppose for each $s \in \mathcal{S}$ and $m \in \mathcal{I}$, let $h^{i_m} = f_{i^m}(s)$, then*

$$\overleftarrow{\bar{\pi}}_{\mathrm{new}}^{\mathrm{CD}}(\cdot|h^{i_m}) = \underset{\overleftarrow{\bar{\pi}}^{\mathrm{CD}}(\cdot|h^{i_m})}{\operatorname{argmin}} D_{\mathrm{KL}}\left(\overleftarrow{\bar{\pi}}^{\mathrm{CD}}(u_{0:K}^{i_m}|h^{i_m}) \| \vec{\pi}_{\mathrm{old}}^{\mathrm{CD}}(u_{0:K}^{i_m}|h^{i_m})\right), \tag{14}$$

*where the forward policy is given by Boltzmann distribution,*

$$\vec{\pi}_{\mathrm{old}}^{\mathrm{CD}}(u_0^{i_m}|h^{i_m}) \propto \exp \mathbb{E}_{\boldsymbol{a}^{i_{1:m-1}}}\left[\frac{1}{\alpha}\hat{Q}_{\overleftarrow{\bar{\pi}}_{\mathrm{old}}^{\mathrm{CD}}}^{i_{1:m}}(s, \boldsymbol{a}^{i_{1:m-1}}, \cdot^{i_m})\right]$$

$$\text{with} \quad \boldsymbol{a}^{i_{1:m-1}} \sim \prod_{j=1}^{m-1}\overleftarrow{\bar{\pi}}_{\mathrm{new}}^{\mathrm{CD}}(a^{i_j}|h^{i_j}). \tag{15}$$

*Then the joint diffusion policy satisfies the following:*

$$\overleftarrow{\bar{\pi}}_{\mathrm{new}}^{\mathrm{CD}} = \underset{\overleftarrow{\bar{\pi}}^{\mathrm{CD}} \in \bar{\boldsymbol{\Pi}}}{\operatorname{argmin}} D_{\mathrm{KL}}\left(\overleftarrow{\bar{\pi}}^{\mathrm{CD}}(\mathbf{u}_{0:K}|s) \| \vec{\pi}_{\mathrm{old}}^{\mathrm{CD}}(\mathbf{u}_{0:K}|s)\right). \tag{16}$$

*Proof.* First, we use $L_{\overleftarrow{\pi}_{\mathrm{old}}^{\mathrm{CD}}}^{i_m}(\pi(\cdot^{i_m}|h^{i_m}))$ to denote

$$D_{\mathrm{KL}}\left(\overleftarrow{\bar{\pi}}^{\mathrm{CD}}(u_{0:K}^{i_m}|h^{i_m}) \| \vec{\pi}_{\mathrm{old}}^{\mathrm{CD}}(u_{0:K}^{i_m}|h^{i_m})\right).$$

And $L_{\overleftarrow{\bar{\pi}}_{\mathrm{old}}^{\mathrm{CD}}}(\boldsymbol{\pi}(\cdot|s))$ to denote

$$D_{\mathrm{KL}}\left(\overleftarrow{\bar{\pi}}^{\mathrm{CD}}(\mathbf{u}_{0:K}|s) \| \vec{\pi}_{\mathrm{old}}^{\mathrm{CD}}(\mathbf{u}_{0:K}|s)\right).$$

Suppose that there exists a policy $\overleftarrow{\bar{\pi}}_0^{\mathrm{CD}} \neq \overleftarrow{\bar{\pi}}_{\mathrm{new}}^{\mathrm{CD}}$, such that $L_{\overleftarrow{\bar{\pi}}_{\mathrm{old}}^{\mathrm{CD}}}(\overleftarrow{\bar{\pi}}_0^{\mathrm{CD}}(\cdot|s)) < L_{\overleftarrow{\bar{\pi}}_{\mathrm{old}}^{\mathrm{CD}}}(\overleftarrow{\bar{\pi}}_{\mathrm{new}}^{\mathrm{CD}}(\cdot|s))$, we have

$$\mathbb{E}_{\boldsymbol{a} \sim \overleftarrow{\bar{\pi}}_0^{\mathrm{CD}}}\left[Q_{\overleftarrow{\bar{\pi}}_{\mathrm{old}}^{\mathrm{CD}}}(s, \boldsymbol{a})\right] + \alpha \sum_{i=1}^n \ell\left(\overleftarrow{\bar{\pi}}_0^{\mathrm{CD}}(\cdot^i|h^i)\right) > \mathbb{E}_{\boldsymbol{a} \sim \overleftarrow{\bar{\pi}}_{\mathrm{new}}^{\mathrm{CD}}}\left[Q_{\overleftarrow{\bar{\pi}}_{\mathrm{old}}^{\mathrm{CD}}}(s, \boldsymbol{a})\right] + \alpha \sum_{i=1}^n \ell\left(\overleftarrow{\bar{\pi}}_{\mathrm{new}}^{\mathrm{CD}}(\cdot^i|h^i)\right). \tag{41}$$

From Eqn.14, we have $L_{\overleftarrow{\bar{\pi}}_{\mathrm{old}}^{\mathrm{CD}}}^{i_m}(\overleftarrow{\bar{\pi}}_{\mathrm{new}}^{\mathrm{CD}}(\cdot^{i_m}|h^{i_m})) \leq L_{\overleftarrow{\bar{\pi}}_{\mathrm{old}}^{\mathrm{CD}}}^{i_m}(\overleftarrow{\bar{\pi}}_0^{\mathrm{CD}}(\cdot^{i_m}|h^{i_m}))$ for every $m = 1, \dots, n$, i.e.,

$$\mathbb{E}_{\boldsymbol{a}^{i_{1:m-1}} \sim \overleftarrow{\bar{\pi}}_{\mathrm{new}}^{\mathrm{CD}}(\cdot|h^{i_{1:m-1}}), a^{i_m} \sim \overleftarrow{\bar{\pi}}_{\mathrm{new}}^{\mathrm{CD}}(\cdot|h^{i_m})}\left[Q_{\overleftarrow{\bar{\pi}}_{\mathrm{old}}^{\mathrm{CD}}}^{i_{1:m}}\left(s, \boldsymbol{a}^{i_{1:m-1}}, a^{i_m}\right) - \alpha \log \overleftarrow{\bar{\pi}}_{\mathrm{new}}^{\mathrm{CD}}\left(a^{i_m}|h^{i_m}\right)\right]$$

$$\geq \mathbb{E}_{\boldsymbol{a}^{i_{1:m-1}} \sim \overleftarrow{\bar{\pi}}_{\mathrm{new}}^{\mathrm{CD}}(\cdot|h^{i_{1:m-1}}), a^{i_m} \sim \overleftarrow{\bar{\pi}}_0^{\mathrm{CD}}(\cdot|h^{i_m})}\left[Q_{\overleftarrow{\bar{\pi}}_{\mathrm{old}}^{\mathrm{CD}}}^{i_{1:m}}\left(s, \boldsymbol{a}^{i_{1:m-1}}, a^{i_m}\right) - \alpha \log \overleftarrow{\bar{\pi}}_0^{\mathrm{CD}}\left(a^{i_m}|h^{i_m}\right)\right].$$

Subtracting both sides of the inequality by $\mathbb{E}_{\boldsymbol{a}^{i_{1:m-1}} \sim \overleftarrow{\boldsymbol{\pi}}^{\text{CD}}_{\text{new}}(\cdot|h^{i_{1:m-1}})} \left[ Q^{i_{1:m-1}}_{\overleftarrow{\boldsymbol{\pi}}^{\text{CD}}_{\text{old}}} \left( s, \boldsymbol{a}^{i_{1:m-1}} \right) \right]$ with the definition of the multi-agent soft advantage function in App. A.2,

$$
\begin{aligned}
&\mathbb{E}_{\boldsymbol{a}^{i_{1:m-1}} \sim \overleftarrow{\boldsymbol{\pi}}^{\text{CD}}_{\text{new}}(\cdot|h^{i_{1:m-1}}), a^{im} \sim \overleftarrow{\pi}^{\text{CD}}_{\text{new}}(\cdot|h^{im})} \left[ A^{i_{1:m}}_{\overleftarrow{\boldsymbol{\pi}}^{\text{CD}}_{\text{old}}} \left( s, \boldsymbol{a}^{i_{1:m-1}}, a^{im} \right) - \alpha \log \overleftarrow{\pi}^{\text{CD}}_{\text{new}} \left( a^{im}|h^{im} \right) \right] \\
&\geq \mathbb{E}_{\boldsymbol{a}^{i_{1:m-1}} \sim \overleftarrow{\boldsymbol{\pi}}^{\text{CD}}_{\text{new}}(\cdot|h^{i_{1:m-1}}), a^{im} \sim \overleftarrow{\pi}^{\text{CD}}_{0}(\cdot|h^{im})} \left[ A^{i_{1:m}}_{\overleftarrow{\boldsymbol{\pi}}^{\text{CD}}_{\text{old}}} \left( s, \boldsymbol{a}^{i_{1:m-1}}, a^{im} \right) - \alpha \log \overleftarrow{\pi}^{\text{CD}}_{0} \left( a^{im}|h^{im} \right) \right].
\end{aligned}
\tag{42}
$$

Combining this with Lemma A.4 gives

$$
\begin{aligned}
&\mathbb{E}_{\boldsymbol{a} \sim \overleftarrow{\boldsymbol{\pi}}^{\text{CD}}_{\text{new}}} \left[ A_{\overleftarrow{\boldsymbol{\pi}}^{\text{CD}}_{\text{old}}}(s, \boldsymbol{a}) + \alpha \sum_{i=1}^{n} \mathcal{H} \left( \overleftarrow{\pi}^{\text{CD}}_{\text{new}} \left( \cdot^i|h^i \right) \right) \right] \\
&= \sum_{m=1}^{n} \left[ \mathbb{E}_{\boldsymbol{a}^{i_{1:m-1}} \sim \overleftarrow{\boldsymbol{\pi}}^{\text{CD}}_{\text{new}}(\cdot|h^{i_{1:m-1}}), a^{im} \sim \overleftarrow{\pi}^{\text{CD}}_{\text{new}}(\cdot|h^{im})} \left[ A^{i_m}_{\overleftarrow{\boldsymbol{\pi}}^{\text{CD}}_{\text{old}}} \left( s, \boldsymbol{a}^{i_{1:m-1}}, a^{im} \right) - \alpha \log \overleftarrow{\pi}^{\text{CD}}_{\text{new}} \left( a^{im}|h^m \right) \right] \right] \\
&\qquad \text{By Inequality 42} \\
&\geq \sum_{m=1}^{n} \left[ \mathbb{E}_{\boldsymbol{a}^{i_{1:m-1}} \sim \overleftarrow{\boldsymbol{\pi}}^{\text{CD}}_{\text{new}}(\cdot|h^{i_{1:m-1}}), a^{im} \sim \overleftarrow{\pi}^{\text{CD}}_{0}(\cdot|h^{im})} \left[ A^{i_m}_{\overleftarrow{\boldsymbol{\pi}}^{\text{CD}}_{\text{old}}} \left( s, \boldsymbol{a}^{i_{1:m-1}}, a^{im} \right) - \alpha \log \overleftarrow{\pi}^{\text{CD}}_{0} \left( a^{im}|s \right) \right] \right] \\
&= \mathbb{E}_{\boldsymbol{a} \sim \overleftarrow{\boldsymbol{\pi}}^{\text{CD}}_{0}} \left[ \mathcal{A}_{\overleftarrow{\boldsymbol{\pi}}^{\text{CD}}_{\text{old}}}(s, \boldsymbol{a}) + \alpha \sum_{i=1}^{n} \mathcal{H} \left( \overleftarrow{\pi}^{\text{CD}}_{0} \left( \cdot^i|h^i \right) \right) \right].
\end{aligned}
$$

The resulting inequality can be equivalently rewritten as

$$
\mathbb{E}_{\boldsymbol{a} \sim \overleftarrow{\boldsymbol{\pi}}^{\text{CD}}_{0}} \left[ Q_{\overleftarrow{\boldsymbol{\pi}}^{\text{CD}}_{\text{old}}}(s, \boldsymbol{a}) \right] + \alpha \sum_{i=1}^{n} \mathcal{H} \left( \overleftarrow{\pi}^{\text{CD}}_{0} \left( \cdot^i|h^i \right) \right) \leq \mathbb{E}_{\boldsymbol{a} \sim \overleftarrow{\boldsymbol{\pi}}^{\text{CD}}_{\text{new}}} \left[ Q_{\overleftarrow{\boldsymbol{\pi}}^{\text{CD}}_{\text{old}}}(s, \boldsymbol{a}) \right] + \alpha \sum_{i=1}^{n} \mathcal{H} \left( \overleftarrow{\pi}^{\text{CD}}_{\text{new}} \left( \cdot^i|h^i \right) \right).
$$

which contradicts Eqn. 41. Hence, for all $\overleftarrow{\boldsymbol{\pi}}^{\text{CD}} \in \overleftarrow{\check{\boldsymbol{\Pi}}}$, $L_{\overleftarrow{\boldsymbol{\pi}}^{\text{CD}}_{\text{old}}}(\overleftarrow{\boldsymbol{\pi}}^{\text{CD}}_{\text{new}}(\cdot|s)) \leq L_{\overleftarrow{\boldsymbol{\pi}}^{\text{CD}}_{\text{old}}}(\overleftarrow{\boldsymbol{\pi}}^{\text{CD}}(\cdot|s))$, i.e.,

$$
\overleftarrow{\boldsymbol{\pi}}^{\text{CD}}_{\text{new}} = \operatorname*{argmin}_{\overleftarrow{\boldsymbol{\pi}}^{\text{CD}} \in \overleftarrow{\check{\boldsymbol{\Pi}}}} D_{\text{KL}} \left( \overleftarrow{\boldsymbol{\pi}}^{\text{CD}}(\mathbf{u}_{0:K}|s) \| \vec{\boldsymbol{\pi}}^{\text{CD}}_{\text{old}}(\mathbf{u}_{0:K}|s) \right).
$$

which finishes the proof. $\qquad\square$

Next, we focus on explaining the **Diffusion Soft Policy Iteration with CD** (Dspic). To clearly demonstrate the correctness of the algorithm, we first propose *Diffusion Soft Policy Improvement with CD*.

**Lemma G.6** (**Diffusion Soft Policy Improvement with CD**). *For any initial diffusion policy $\overleftarrow{\boldsymbol{\pi}}^{CD}_{0}$ and $\vec{\boldsymbol{\pi}}^{CD}_{0}$ with CD $\{f_i\}_{i=1}^{n}$. Let $i_{1:n} \in Sym(n)$ be an agent permutation and for every $m \in \mathcal{I}$, we repeatedly use Eqn. 15 in Proposition 4.6 to iterate from $\overleftarrow{\pi}^{CD}_{old}(\cdot|f_{i^m}(s))$ to obtain $\overleftarrow{\pi}^{CD}_{new}(\cdot|f_{i^m}(s))$ for all $s \in \mathcal{S}$. Then we have $\hat{Q}_{\overleftarrow{\boldsymbol{\pi}}^{CD}_{new}}(s, \boldsymbol{a}) \geq \hat{Q}_{\overleftarrow{\boldsymbol{\pi}}^{CD}_{old}}(s, \boldsymbol{a})$ for all $(s, \boldsymbol{a}) \in \mathcal{S} \times \mathcal{A}$ and $\hat{J} \left( \overleftarrow{\boldsymbol{\pi}}^{CD}_{new} \right) \geq \hat{J} \left( \overleftarrow{\boldsymbol{\pi}}^{CD}_{old} \right)$.*

*Proof.* According to Proposition 4.6, we obtain:

$$
\overleftarrow{\boldsymbol{\pi}}^{\text{CD}}_{\text{new}} = \operatorname*{argmin}_{\overleftarrow{\boldsymbol{\pi}}^{\text{CD}} \in \overleftarrow{\check{\boldsymbol{\Pi}}}} D_{\text{KL}} \left( \overleftarrow{\boldsymbol{\pi}}^{\text{CD}}(\mathbf{u}_{0:K}|s) \| \vec{\boldsymbol{\pi}}^{\text{CD}}_{\text{old}}(\mathbf{u}_{0:K}|s) \right),
$$

then

$$
D_{\text{KL}} \left( \overleftarrow{\boldsymbol{\pi}}^{\text{CD}}_{\text{new}}(\mathbf{u}_{0:K} \mid s) \| \vec{\boldsymbol{\pi}}^{\text{CD}}_{\text{old}}(\mathbf{u}_{0:K} \mid s) \right) \leq D_{\text{KL}} \left( \overleftarrow{\boldsymbol{\pi}}^{\text{CD}}_{\text{old}}(\mathbf{u}_{0:K} \mid s) \| \vec{\boldsymbol{\pi}}^{\text{CD}}_{\text{old}}(\mathbf{u}_{0:K} \mid s) \right).
$$

Consider that

$$D_{\mathrm{KL}}\left(\overleftrightarrow{\boldsymbol{\pi}}^{\mathrm{CD}}(\mathbf{u}_{0:K}\mid s)\|\vec{\boldsymbol{\pi}}_{\mathrm{old}}^{\mathrm{CD}}(\mathbf{u}_{0:K}\mid s)\right) = -\ell\left(\overleftrightarrow{\boldsymbol{\pi}}^{\mathrm{CD}}(\cdot\mid s)\right) - \mathbb{E}_{\mathbf{u}_0\sim\overleftrightarrow{\boldsymbol{\pi}}^{\mathrm{CD}}}\left[\log\vec{\boldsymbol{\pi}}_{\mathrm{old}}^{\mathrm{CD}}(\mathbf{u}_0\mid s)\right]$$
$$= -\left(\ell\left(\overleftrightarrow{\boldsymbol{\pi}}^{\mathrm{CD}}(\cdot\mid s)\right) + \frac{1}{\alpha}\mathbb{E}_{\boldsymbol{a}\sim\overleftrightarrow{\boldsymbol{\pi}}_{\mathrm{old}}^{\mathrm{CD}}}\left[\hat{Q}_{\overleftrightarrow{\boldsymbol{\pi}}_{\mathrm{old}}^{\mathrm{CD}}}(s,\boldsymbol{a})\right] - Z_{\overleftrightarrow{\boldsymbol{\pi}}_{\mathrm{old}}^{\mathrm{CD}}}(s)\right),$$

we can get

$$\alpha\ell\left(\overleftrightarrow{\boldsymbol{\pi}}_{\mathrm{new}}^{\mathrm{CD}}(\cdot\mid s)\right) + \mathbb{E}_{\boldsymbol{a}\sim\overleftrightarrow{\boldsymbol{\pi}}_{\mathrm{old}}^{\mathrm{CD}}}\left[\hat{Q}_{\overleftrightarrow{\boldsymbol{\pi}}_{\mathrm{old}}^{\mathrm{CD}}}(s,\boldsymbol{a})\right] \geq \alpha\ell\left(\overleftrightarrow{\boldsymbol{\pi}}_{\mathrm{old}}^{\mathrm{CD}}(\cdot\mid s)\right) + \mathbb{E}_{\boldsymbol{a}\sim\overleftrightarrow{\boldsymbol{\pi}}_{\mathrm{old}}^{\mathrm{CD}}}\left[\hat{Q}_{\overleftrightarrow{\boldsymbol{\pi}}_{\mathrm{old}}^{\mathrm{CD}}}(s,\boldsymbol{a})\right] = V_{\overleftrightarrow{\boldsymbol{\pi}}_{\mathrm{old}}^{\mathrm{CD}}}(s). \tag{43}$$

At last, considering the soft Bellman equation, the following holds:

$$\hat{Q}_{\overleftrightarrow{\boldsymbol{\pi}}_{\mathrm{old}}^{\mathrm{CD}}}(s,\boldsymbol{a}) = r(s,\boldsymbol{a}) + \gamma\mathbb{E}_{s'\sim P}\left[V_{\overleftrightarrow{\boldsymbol{\pi}}_{\mathrm{old}}^{\mathrm{CD}}}(s')\right]$$

By Inequality 43

$$\leq r(s,\boldsymbol{a}) + \gamma\mathbb{E}_{s'\sim P}\left[\mathbb{E}_{\boldsymbol{a}\sim\overleftrightarrow{\boldsymbol{\pi}}_{\mathrm{new}}^{\mathrm{CD}}}\left[\hat{Q}_{\overleftrightarrow{\boldsymbol{\pi}}_{\mathrm{old}}^{\mathrm{CD}}}(s',\boldsymbol{a})\right] + \alpha\ell\left(\overleftrightarrow{\boldsymbol{\pi}}_{\mathrm{new}}^{\mathrm{CD}}(\cdot\mid s)\right)\right] \tag{44}$$

$$\vdots$$

$$\leq \hat{Q}_{\overleftrightarrow{\boldsymbol{\pi}}_{\mathrm{new}}^{\mathrm{CD}}}(s,\boldsymbol{a}),$$

where we have repeatedly expanded $\hat{Q}_{\overleftrightarrow{\boldsymbol{\pi}}_{\mathrm{old}}^{\mathrm{CD}}}$ on the RHS by applying the soft Bellman equation and the bound in Inequality 43. This proof is based on the convergence of the joint soft $Q$-function given in Lemma 4.5.

We use it to prove the claim as follows,

$$\hat{V}_{\overleftrightarrow{\boldsymbol{\pi}}_{\mathrm{new}}^{\mathrm{CD}}}(s) = \mathbb{E}_{\boldsymbol{a}\sim\overleftrightarrow{\boldsymbol{\pi}}_{\mathrm{new}}^{\mathrm{CD}}}\left[\hat{Q}_{\overleftrightarrow{\boldsymbol{\pi}}_{\mathrm{new}}^{\mathrm{CD}}}(s,\boldsymbol{a})\right] + \alpha\ell\left(\overleftrightarrow{\boldsymbol{\pi}}_{\mathrm{new}}^{\mathrm{CD}}(\cdot\mid s)\right)$$

By Inequality 44

$$\geq \mathbb{E}_{\boldsymbol{a}\sim\overleftrightarrow{\boldsymbol{\pi}}_{\mathrm{new}}^{\mathrm{CD}}}\left[\hat{Q}_{\overleftrightarrow{\boldsymbol{\pi}}_{\mathrm{old}}^{\mathrm{CD}}}(s,\boldsymbol{a})\right] + \alpha\ell\left(\overleftrightarrow{\boldsymbol{\pi}}_{\mathrm{new}}^{\mathrm{CD}}(\cdot\mid s)\right)$$

By Inequality 43

$$\geq \mathbb{E}_{\boldsymbol{a}\sim\overleftrightarrow{\boldsymbol{\pi}}_{\mathrm{old}}^{\mathrm{CD}}}\left[\hat{Q}_{\overleftrightarrow{\boldsymbol{\pi}}_{\mathrm{old}}^{\mathrm{CD}}}(s,\boldsymbol{a})\right] + \alpha\ell\left(\overleftrightarrow{\boldsymbol{\pi}}_{\mathrm{old}}^{\mathrm{CD}}(\cdot\mid s)\right)$$
$$= V_{\overleftrightarrow{\boldsymbol{\pi}}_{\mathrm{old}}^{\mathrm{CD}}}(s).$$

Subsequently, the monotonic improvement property of the joint maximum entropy return follows naturally, as

$$\hat{J}\left(\overleftrightarrow{\boldsymbol{\pi}}_{\mathrm{new}}^{\mathrm{CD}}\right) = \mathbb{E}_{s\sim d}\left[V_{\overleftrightarrow{\boldsymbol{\pi}}_{\mathrm{new}}^{\mathrm{CD}}}(s)\right] \geq \mathbb{E}_{s\sim d}\left[V_{\overleftrightarrow{\boldsymbol{\pi}}_{\mathrm{old}}^{\mathrm{CD}}}(s)\right] = \hat{J}\left(\overleftrightarrow{\boldsymbol{\pi}}_{\mathrm{old}}^{\mathrm{CD}}\right).$$

$$\square$$

Finally, let's prove Theorem 4.7, i.e.,

**Theorem 4.7 (Diffusion Soft Policy Iteration with CD).** *For any initial diffusion policy $\overleftrightarrow{\boldsymbol{\pi}}_0^{CD}$ and $\vec{\boldsymbol{\pi}}_0^{CD}$ with CD $\{f_i\}_{i=1}^n$. Let $i_{1:n} \in Sym(n)$ be an agent permutation and for every $m \in \mathcal{I}$, we repeatedly use Eqn. 15 in Proposition 4.6 to iterate from $\overleftrightarrow{\boldsymbol{\pi}}_{old}^{CD}(\cdot|f_{i^m}(s))$ to obtain $\overleftrightarrow{\boldsymbol{\pi}}_{new}^{CD}(\cdot|f_{i^m}(s))$ for all $s \in \mathcal{S}$. Then we have (1) $\hat{Q}_{\overleftrightarrow{\boldsymbol{\pi}}_{new}^{CD}}(s,\boldsymbol{a}) \geq \hat{Q}_{\overleftrightarrow{\boldsymbol{\pi}}_{old}^{CD}}(s,\boldsymbol{a})$ for all $(s,\boldsymbol{a}) \in \mathcal{S} \times \mathcal{A}$ and $\hat{J}\left(\overleftrightarrow{\boldsymbol{\pi}}_{new}^{CD}\right) \geq \hat{J}\left(\overleftrightarrow{\boldsymbol{\pi}}_{old}^{CD}\right)$, (2) the diffusion soft policy will eventually converge to a $\overleftrightarrow{\boldsymbol{\pi}}_\star^{CD}$ as the number of iterations approaches infinity, with $\hat{Q}_{\overleftrightarrow{\boldsymbol{\pi}}_\star^{CD}}(s,\boldsymbol{a}) \geq \hat{Q}_{\overleftrightarrow{\boldsymbol{\pi}}^{CD}}(s,\boldsymbol{a})$ and $\hat{J}\left(\overleftrightarrow{\boldsymbol{\pi}}_\star^{CD}\right) \geq \hat{J}\left(\overleftrightarrow{\boldsymbol{\pi}}^{CD}\right)$ for any $\overleftrightarrow{\boldsymbol{\pi}}^{CD} \in \bar{\bar{\boldsymbol{\Pi}}}$.*

*Proof.* We have proven **(1)** in Lemma G.6. Let's prove **(2)** below.

Let $\overleftarrow{\bar{\pi}}_k^{\mathrm{CD}}$ be the policy function at iteration $k$. First, by Lemma 4.5 we have that $Q_{\overleftarrow{\bar{\pi}}_k^{\mathrm{CD}}}(s, \boldsymbol{a}) \leq Q_{\overleftarrow{\bar{\pi}}_{k+1}^{\mathrm{CD}}}(s, \boldsymbol{a})$, and that the soft $Q$-function is upper-bounded by $Q_{\max}$ for all $\overleftarrow{\bar{\pi}}^{\mathrm{CD}} \in \overleftarrow{\bar{\Pi}}$ (both reward and entropy are bounded). Hence, the sequence converges to some limit point $\overleftarrow{\bar{\pi}}_0^{\mathrm{CD}}$.

Then, considering this limit point policy $\overleftarrow{\bar{\pi}}_0^{\mathrm{CD}}$, we can get for any $\overleftarrow{\bar{\pi}}^{\mathrm{CD}} \in \overleftarrow{\bar{\Pi}}$ with Lemma G.6,

$$\hat{Q}_{\overleftarrow{\bar{\pi}}_\star^{\mathrm{CD}}}(s, \boldsymbol{a}) \geq \hat{Q}_{\overleftarrow{\bar{\pi}}_\star^{\mathrm{CD}}}(s, \boldsymbol{a}), \quad \hat{J}\left(\overleftarrow{\bar{\pi}}_\star^{\mathrm{CD}}\right) \geq \hat{J}\left(\overleftarrow{\bar{\pi}}^{\mathrm{CD}}\right)$$

always holds, which finishes the proof. $\square$

# H. Approximation Gap with Ideal CD

Although our algorithm achieved excellent performance in practice, it is undeniable that our method has a certain gap compared to the ideal situation. Specifically, the $\mathcal{L}_{\mathrm{ort}}$ value is often not $0$ at convergence, which means that a certain degree of input overlap can be ignored in terms of macroscopic performance. To make this clearer, we use $\rho$ to denote the ratio of the converged $\mathcal{L}_{\mathrm{ort}}$ value to the initial value, i.e.,

$$\rho = \frac{\mathcal{L}_{\mathrm{ort}, t=T}}{\mathcal{L}_{\mathrm{ort}, t=0}}. \tag{45}$$

Since the magnitudes of $\rho$ differ significantly, we show the average of $\log_{10} \rho$. As shown in Tab. 8, the overlap is already very small ($< 3.5\%$ and $0.17\%$ in average) in both tasks. Therefore, the remaining spatial overlap has a relatively small impact on the final result.

|  | MaMuJoCo-v4 | SMAC | SMACv2 | LBF | Overall |
|---|---|---|---|---|---|
| Average of $\log_{10} \rho$ | -3.14 | -1.46 | -1.48 | -7.44 | -2.78 |
| corresponding $\rho$ | 7.24e-4 | 3.47e-2 | 3.31e-2 | 3.61e-8 | 1.66e-3 |

*Table 8.* The ratio of the converged $\mathcal{L}_{\mathrm{ort}}$ value to the initial value over tasks.

