# OpenReview forum: "Towards Complete Multi-Agent Coordination Policy Learning via Denoising Maximum Entropy Optimization"
_ICML.cc/2026/Conference — ICML 2026 regular_

### Official Review · Reviewer_L9Pj · 2026-03-02

**Soundness:** 3
**Presentation:** 3
**Significance:** 3
**Originality:** 3
**Overall Recommendation:** 4
**Confidence:** 5

**Summary:**

This paper proposes Dspic, an approach investigates the underlying causes of the performance discrepancy between shared policies and QRE policies in heterogeneous environments. Their analysis reveals that such discrepancy stems from the overlap in the observation space relied upon by agents during decision-making. To address this issue, the authors propose an approach that differs from existing methods which directly concentrate agent IDs , and instead employ observation space complete division. To mitigate the increased learning difficulty caused by the elevated density of the observation space on which policy learning depends, they introduce a diffusion model to learn the shared policy.

**Compliance With Llm Reviewing Policy:**

Affirmed.

**Final Justification:**

This paper presents Dspic, a solid study investigating the performance discrepancy between shared and QRE policies in heterogeneous environments. During the rebuttal, the authors proactively and convincingly addressed the initial concerns regarding the regularization scale in projection training and the previously missing experimental results. They provided comprehensive implementation details and committed to incorporating these additions into the final manuscript. Furthermore, their clarification on the efficacy of the sequential update scheme is technically sound and resolves the earlier ambiguities. Given that the core methodology is well-grounded and the authors' responses have significantly enhanced the overall quality and clarity of the work, I recommend the weak accept of this paper.

**Key Questions For Authors:**

- **Q1:**  During the matrix $\mathbf{P}$ training, how can Dspic ensure balanced weighting for loss gradients?
- **Q2:** Given that the two tasks are homogeneous and involve a nearly identical number of agents, why does Dspic exhibit such distinct performance differences between them?

**Limitations:**

yes

**Strengths And Weaknesses:**

**Strength:**

The paper motivates its study from the perspective of the definition of homogeneous settings and clearly distinguishes its contributions from prior related work. The preliminaries are presented in a precise and rigorous manner, and all central theoretical claims are accompanied by formal proofs. Although the authors acknowledge that there are certain indivisible regions, the method achieves a practical balance between model simplicity and division completeness.

**Weakness:**

- **W1:** In Eqn. 10, when role embeddings $z_i$ and $z_j$ are similar, weights $||z_i-z_j||$ approach zero, vanishing the loss gradient. When roles differ sharply, large weights combined with large norms $||g_\psi(z_i)^\top g_\psi(z_j)||_F$ cause gradient explosion. These phenomenons can be found in Fig. 3.
- **W2**: The adoption of sequential updates (likes MAT) lacks justification, raising concerns that performance gains may hinge on this design choice.
- **W3:** In the 6h_vs_8z environment, the performance of Dspic is inferior to the MAPPO baseline during the early training phase. In the terran_5_vs_5 environment, Dspic not only under performs the Kaleidoscope baseline in terms of performance, but also exhibits significantly worse training stability compared to the latter. Despite the two tasks being homogeneous and involving a nearly identical number of agents, Dspic exhibits distinct performance differences between them.
- **W4:** Results for MAT-Dec are absent, undermining a fair comparison and the claimed improvements.

---

> ### Author Rebuttal · Authors · 2026-03-31
>
> ### [W1 & Q1] Balance Weighting Values for the Loss Gradients
> After obtaining the role embeddings $\\{z_i\\}\_{i=1}^n$, we **apply normalization to the embeddings**. Specifically, for the $j$-th dimension of the $i$-th agent's embedding, we transform it as $\frac{z\_{ij}-\min\_k z\_{kj}}{\max\_k z\_{kj} - \min\_k z\_{kj}+\epsilon}\to z\_{ij}$, mapping the values to the $[0,1]$ interval. This normalization ensures that the weights in Eqn. 10 are strictly bounded, effectively preventing gradient explosion even when roles differ sharply. And when the weight approaches zero, it indicates that the agents are highly similar (almost homogeneous). In such cases, we do not strictly enforce disjoint input subspaces, as sharing both policies and observation spaces is expected for us. We will include this detail in the revised version.
> ### [W2] Effect of Sequential Updates
> **Although complete division (CD) and diffusion strategies are the main reasons for performance improvement, sequential updates further enhance stability**. To show it, we conducted an ablation study in the HalfCheetah-2x3 environment to compare the performance of our approach with and without this component:
> ||Episode Reward|
> |:---|:---:|
> |HASAC (Baseline)|7328.3±1150.0|
> |Dspic w/o Sequential Update|8592.1±409.7|
> |Dspic (ours)|**8722.7±498.6**|
>
> The results show that even without sequential updates, our approach outperforms the baseline (HASAC) as well.
> ### [W3 & Q2] Performance on 6h_vs_8z and terran_5_vs_5
> **Regarding 6h_vs_8z**: MAPPO does indeed exhibit better sample efficiency compared to our approach. This is because **MAPPO directly shares parameters and does not need to learn role embeddings**, resulting in faster convergence. Considering the homogeneous setting of 6h_vs_8z, it is reasonable for MAPPO to exhibit higher sample efficiency. But our method is applicable to a wider range of environments.
>
> **Regarding terran_5_vs_5**: SMACv2 tasks present **significant challenges for policy-based methods**. Kaleidoscope, based on QMIX, often shows better performance and more stable results than policy-based algorithms (such as MAPPO, HASAC, and our approach) in this environment. Our experiments show that **our approach achieves the best performance for policy-based methods in this environment**. There are also some value-based methods that have achieved excellent performance in the SMACv2 environment, such as BVME [1], which confirms our point of view.
>
> [1] Wei, Duan, et al. "Bandwidth-constrained Variational Message Encoding for Cooperative Multi-agent Reinforcement Learning" AAMAS 2026.
> ### [W4] Absent Results for MAT-Dec in 6h_vs_8z
> We apologize for the missing MAT-Dec curve in the 6h_vs_8z plot. We will include the complete plot in the revised version. The full comparison can also be viewed in https://anonymous.4open.science/r/Dspic-4870/6h_vs_8z.png. The performance in MAT-Dec is not as good as our approach as shown in the figure. Thank you very much for your suggestions on our article.

---

> > ### Author Rebuttal · Reviewer_L9Pj · 2026-04-02
> >
> > Thanks for the rebuttal. All of my concerns have been fully addressed by the authors. I maintain my original score.

---

> > > ### Author Response · Authors · 2026-04-02
> > >
> > > We sincerely appreciate your time and thoughtful follow-up. We are pleased that our rebuttal has addressed your concerns and thank you for recognizing the technical contributions of our work and maintaining your positive evaluation.

---

### Official Review · Reviewer_g4s4 · 2026-03-03

**Soundness:** 2
**Presentation:** 1
**Significance:** 2
**Originality:** 2
**Overall Recommendation:** 3
**Confidence:** 4

**Summary:**

This paper presents a novel method based on VAE to solve the homogeneous policy problem caused by policy network parameter sharing. To efficiently distinguish different agent roles, the authors propose a Role-based Orthogonal Complete Division method. Empirical results on MaMuJoCo, SMAC, SMACv2, and LBF demonstrate the strong performance of the proposed method.

**Compliance With Llm Reviewing Policy:**

Affirmed.

**Final Justification:**

The authors addressed my concerns.

**Key Questions For Authors:**

Please see the Weaknesses.

**Limitations:**

No.

**Strengths And Weaknesses:**

Strengths:
1. The proposed method efficiently divides the observation space of agents using the Role-based Orthogonal Complete Division.
2. The experimental results shown in this work demonstrate the superiority of the proposed method.


Weaknesses:
1. The abbreviation of HARL should be explained when it first appeared in the Introduction section. Moreover, the author claimed that: "In contrast, HARL algorithms extend the maximum entropy (MaxEnt) (Haarnoja et al., 2018) theory to MARL, customize one policy to each agent, and sequentially update policies". But not all HARL algorithms extend MaxEnt, as the author discussed in the "Related work" section.

2. The motivation that the performance bottleneck of shared policy network parameters arises from the similar observation space is not novel. This problem has been observed by CTR in [1]. The authors should explicitly discuss the differences between CTR and the proposed method. Moreover, it can be confusing why the authors believe that the policy deviation actually stems from the overlap in the agents’ state space, leading to interference between agents. The derivations actually cannot depict the results.

3. The addition of the Diffusion policy is not necessary and can be redundant. The authors claim that the use of diffusion policy is to strengthen policy expressiveness due to the increased density of the policy state space. However, the complex policy state space can facilitate the increase in policy expressiveness. The normal neural network, such as the shared RNN in the state-of-the-art QMIX, is already enough to model the complex state density. Moreover, the diffusion policy is totally irrelevant to the main role division objective of this work.



[1] Li, T., Zhu, K., Li, J., & Zhang, Y. (2024). Learning distinguishable trajectory representation with contrastive loss. Advances in Neural Information Processing Systems, 37, 64454-64478.

---

> ### Author Rebuttal · Authors · 2026-03-31
>
> Thank you very much for your recognition of our Role-based Orthogonal Complete Division mechanism and experimental results. We provide a detailed explanation below for the questions you have raised.
> ### [W1] Writing Issues
> We sincerely appreciate the reviewer for pointing out the imprecise description of HARL. We will refine the statements in the revised version.
> ### [W2.1] Differences between CTR and Dspic
> We demonstrate the differences between CTR and Dspic from two perspectives, although they have a similar observation that input overlap can lead to failure in learning shared policies and try to solve it by roles.
>
> **For role learning methods**, CTR employs **contrastive learning** with other agents as negative samples to foster diversity when learning roles. In contrast, Dspic learns roles **by VAE from trajectories and encodes it into orthogonal matrix space**.
>
> **For algorithmic mechanism**, CTR is a **soft-constraint** method that provides contrast loss as diversity regularization, while Dspic is a **hard-constraint** method by projecting the input directly into disjoint space. Projection provides an isolation of information flow and yields better performance.
> ||Win rate in 3s5z_vs_3s6z (%)|
> |:---|:---:|
> |CTR|63.3±23.4|
> |Dspic (ours)|**87.5±6.6**|
>
> ### [W2.2] How the Derivations Reflect Our Insight
> We clarify that **our theoretical derivation can directly demonstrate our thinking process and insight**. Specifically, in **Proposition G.3 on Pages 26-27**, we discovered a crucial point: the expression for $M_{\pi}(s,a)$ requires gradient calculation for terms containing the product of shared policies. This results in many higher-order polynomial terms $\pi(a|s)^k$, where $k=1,2,...,n$. This means that even after calculating the gradient of $\pi(a|s)$, $M_{\pi}(s,a)$ still contains terms of $\pi(a|s)^{k-1}$. This causes $M_{\pi}(s,a)$ to deviate from heterogeneous inputs (or when the agents' inputs are completely disjoint), leading to policy deviation. This shows that policy deviation is caused by input overlap, which also motivates us to design complete division. We sincerely appreciate you pointing out the connection between our approach and CTR.
> ### [W3] The Necessity of Diffusion Policy
> This is an important question, but **our empirical results indicate that Diffusion Policy is necessary**. To provide a fair comparison, we designed an ablation study in 3s5z_vs_3s6z and MMM2: we loaded the projection matrices learned by our approach into a QMIX framework (using a shared RNN with hidden size 256) to project inputs before processing.
> |Win Rate Test|3s5z_vs_3s6z|MMM2|
> |:---|:---:|:---:|
> |Dspic (ours)|**75.8±11.8**|**95.8±1.4**|
> |RNN QMIX|18.1±6.4|67.8±1.3|
> |RNN QMIX+Orthogonal Projection|72.3±2.9|75.8±8.0|
>
> The results show that while applying our projection significantly boosts QMIX performance, it still falls short of our approach. The reason is that diffusion models actually learn score functions, while RNNs need to directly learn the true distributions. And often the former is simpler, leading to better performance.

---

> > ### Author Rebuttal · Reviewer_g4s4 · 2026-04-03
> >
> > I appreciate the author's efforts to address the concerns.

---

> > > ### Author Response · Authors · 2026-04-03
> > >
> > > We would like to express our sincere gratitude for your time and effort dedicated to reviewing our submission. We feel glad to address your concerns and grateful for your recognition.

---

### Official Review · Reviewer_EFQE · 2026-03-03

**Soundness:** 3
**Presentation:** 2
**Significance:** 3
**Originality:** 2
**Overall Recommendation:** 4
**Confidence:** 3

**Summary:**

The authors propose Dspic algorithm to balance the trade-off between the knowledge sharing and policy customization in MARL. Specifically, Dspic uses CD to divide the observation space, ensuring that each agent's subspace does not overlap. Next, it trains the model using a maximum entropy multi-agent reinforcement learning algorithm. It also introduces a diffusion policy to enhance the policy's representation ability. The paper verifies the effectiveness of the algorithm across multiple environments.

**Compliance With Llm Reviewing Policy:**

Affirmed.

**Final Justification:**

The authors' response addressed my concerns. So I raise my score from 3 to 4.

**Key Questions For Authors:**

1. In tasks with many agents, the required dimension for the orthogonal projection matrix will increase sharply. How does this affect the algorithm's efficiency and scalability?

2. The algorithm uses random exploration during a warmup phase to train the VAE-based role embedder . If early random actions fail to find meaningful rewards or dynamic differences, the extracted $z_i$ vectors will be indistinguishable. Will this cause the orthogonal matrix $P_i$ construction to crash and fail the entire algorithm?

3. The paper mentions ROMA[1], a role-embedding MARL algorithm. Compared to MARL algorithms based on role embedding and space decomposition like ROMA[1] or RODE[2], does Dspic have a performance advantage? Please provide comparative experiments to prove this.

[1] Roma: Multi-agent reinforcement learning with emergent roles
[2] Rode: Learning roles to decompose multi-agent tasks

**Limitations:**

yes

**Strengths And Weaknesses:**

# Strengths:
1. The experiments are quite detailed. They include comparisons with several baseline algorithms and provide useful ablation studies on key components like CD, Diffusion, and CTA.

2. The paper attempts to build a complete theoretical analysis framework. The paper provides theoretical proofs for key concepts like Zero Policy Deviation , Orthogonal Construction , and Policy Iteration.

# Weaknesses:
1. The diffusion model adds a heavy computational burden. The authors only discuss the extra time needed for training. However, they ignore inference latency. The authors should evaluate the computational overhead introduced by the diffusion model during testing

2. There is a gap between the theory and the actual implementation. The authors claim an optimality guarantee. However, in Proposition 4.4, they admit that indivisible regions exist. This practical compromise weakens the theoretical guarantee.

---

> ### Author Rebuttal · Authors · 2026-03-31
>
> We sincerely thank the reviewer for the recognition of our theoretical framework's completeness and our detailed experiments. Below, we address your remaining concerns in detail.
> ### [W1] Inference Latency of our approach
> **Our approach obtains an efficiency-performance trade-off during inference.** To accurately assess real-world efficiency, we measured the average inference latency by recording the time required for the model to interact with the environment for 1e6 timesteps across all benchmarks.
> ||HASAC Latency|Dspic Latency|
> |:---|:---:|:---:|
> |MaMuJoCo|281.7s|347.8s (1.23×)|
> |SMAC|1542.3s|1670.2s (1.08×)|
> |SMACv2|2846.0s|3278.6s (1.15×)|
> |LBF|266.2s|317.1s (1.19×)|
> |Overall|878.2s|1000.1s (1.14×)|
>
> ### [W2] Gap between the Theory and the Actual Implementation
> We clarify that **the overlap induced during learning (as described in Proposition 4.4) is very small**. To quantify the effectiveness of our practical division, we tracked the geometric mean of $\rho=\frac{\mathcal{L}\_{ort,T}}{\mathcal{L}\_{ort,0}}$ across all environments, which represents the ratio of the final overlap region to the initial state. (We divide the final value by the initial value to prevent order of magnitude effects.)
> ||MaMuJoCo|SMAC|SMACv2|LBF|Overall|
> |:---|:---:|:---:|:---:|:---:|:---:|
> |$\bar{\rho}$|7.24e-4|3.47e-2|3.31e-2|3.61e-8|1.66e-3|
>
> Our data shows that **the overlap loss is reduced by over 96.5% (and up to 99.8% overall) during training**. This indicates our construction always has good performance and is able to learn sufficiently good policies. It is worth mentioning that works in Continual Learning also utilize a similar balance, such as SplitLoRA [1].
>
> [1] Haomiao, Qiu, et al. "SplitLoRA: Balancing Stability and Plasticity in Continual Learning Through Gradient Space Splitting" arXiv preprint arXiv:2505.22370 (2025).
> ### [Q1] Efficiency and Scalability in Larger Tasks
> **Our approach maintains highly competitive performance and stable training latency in most cases with fixed $d'=32$**. To show our approach's scalability, we conducted stress tests in the 25m environment (25 agents battle against 25 agents), which is larger in scale than all the environments in the paper. Maintaining the projection dimension at $d'=32$, we compared the Episode Rewards and training time of our approach against HASAC after 5e5 timesteps:
> ||Episode Rewards|Training time|
> |:---:|:---:|:---:|
> |HASAC|13.92±0.33|47.48h|
> |Dspic(ours)|**14.49±0.21**|53.38h (1.12×)|
>
> ### [Q2] Concerns Regarding Potential Exploration Failure
> We share the concern but our results show that **this situation rarely occurs, and the dynamically updated matrix $\mathbf P_i$ further prevents this**. We have attempted to update the role using the current buffer every 200 epochs, but the performance is similar to our algorithm:
> ||Rewards in HalfCheetah-2x3|
> |:---|:---:|
> |Dspic|**8722.7±498.6**|
> |Dspic+dynamic role adjustment|8695.3±304.1|
>
> For simplicity, we report the pre-training version in our paper. Meanwhile dynamically updated matrix indicates that the role information is continuously learned; as shown in Figure 3, the projections for different roles become more well-clustered as training progresses, demonstrating the robustness of our algorithm.
> ### [Q3] Comparison with ROMA and RODE
> We conducted comparative experiments across four representative environments, reporting average performance and std:
> ||Dspic (ours)|ROMA|RODE|
> |:---|:---:|---:|---:|
> |6h_vs_8z |**19.04±0.65**|15.00±1.44|18.82±0.28|
> |MMM2 |**19.51±0.20**|15.23±0.97|19.48±0.76|
> |terran_5_vs_5 |**15.06±1.63**|14.58±1.54|5.64±0.40|
> |LBF2p3f |**0.97±0.03**|0.59±0.03|0.47±0.23|
>
> The results show that **our approach achieves competitive or superior performance compared to ROMA and RODE across a wide range of tasks**. All these additional results will be included in the revised version.

---

> > ### Author Rebuttal · Reviewer_EFQE · 2026-04-03
> >
> > Thanks for authors' response. My concerns are addressed and I will raise my score.

---

> > > ### Author Response · Authors · 2026-04-03
> > >
> > > We are happy that all your concerns have been addressed. We sincerely appreciate your efforts in reviewing our paper and for raising the score.

---

### Official Review · Reviewer_3qUz · 2026-03-12

**Soundness:** 3
**Presentation:** 2
**Significance:** 2
**Originality:** 2
**Overall Recommendation:** 4
**Confidence:** 4

**Summary:**

This paper addresses a limitation in multi-agent reinforcement learning (MARL) where widely used parameter sharing can hinder performance in settings requiring heterogeneous agent behaviors. To overcome this, the authors introduce the Dspic method, which employs a diffusion-based policy combined with a complete input division across agents. Unlike prior approaches that either train separate policies per agent or rely on complex intrinsic rewards, Dspic maintains the efficiency of shared policies while encouraging diversity through fully partitioned input dimensions, effectively assigning discriminative roles to each agent. Experimental results on several MARL benchmarks show that Dspic achieves competitive or superior performance compared to existing baselines.

**Compliance With Llm Reviewing Policy:**

Affirmed.

**Final Justification:**

The paper presents a sound and practically significant approach to heterogeneous MARL, with strong empirical validation and a clear formulation. The authors’ rebuttal satisfactorily addressed my main concerns, providing helpful clarifications. Overall, I maintain my weak acceptance recommendation.

**Key Questions For Authors:**

Q1. The analysis of how roles are identified and differentiated is well presented and provides interesting insights. However, it would also be helpful to understand how these roles translate into behavioral differences. Do the authors have any analysis showing how the action distributions or policies differ across the identified roles?

Q2. I have an additional question regarding the role embedding. In SMAC, as shown by the authors, it seems relatively clear that roles are differentiated across units, making the separation interpretable. However, it is less clear how roles are differentiated in environments such as LBF or MAMuJoCo. Could the authors elaborate on how role differentiation emerges in these environments? Additionally, how do the authors determine whether the learned roles are well separated or meaningful in these settings?

Q3. Diffusion policies have been widely adopted in various domains beyond MARL. In the context of MARL scenarios, it would be helpful to understand the computational overhead introduced by employing a diffusion policy. Could the authors provide additional details on the training and inference cost compared to standard policy architectures?

**Limitations:**

yes

**Strengths And Weaknesses:**

**Strengths**

**S1.  Novelty of learning heterogeneous policies under Complete Division**

The authors address agent-level individuality from a novel and meaningful perspective under Complete Division. While most prior MARL approaches explicitly introduce separate policy parameters, for example, group-wise policies, role-conditioned policies, or independent policies, this work maintains a shared policy and instead induces individuality through spatial division of the input representation, preserving parameter efficiency and scalability.


**S2. Importance of HA MARL for real-world deployment**

In many real-world systems, agents often differ in capabilities, observations, or dynamics, making the homogeneity assumption unrealistic and limited. By explicitly analyzing the policy deviation issue under shared policies and proposing a principled mechanism to mitigate inter-agent interference, the paper contributes toward bridging the gap between theoretical MARL formulations and realistic heterogeneous environments.

**S3. Theoretical and empirical evidence for authors’ claims**

The paper provides a well-structured and mathematically grounded justification of its main ideas. For instance, the notion of Complete Division, provided in Definition 4.2, offers a clear and principled way to eliminate overlap in the agents’ effective state representations while still allowing parameter sharing. Furthermore, the paper presents a thorough empirical evaluation across diverse and challenging benchmarks, including MAMuJoCo, SMAC, SMACv2, and LBF, demonstrating the robustness of the proposed approach across different types of cooperative MARL environments.


**Weakness**

**W1. Comparison with prior role-based methods**

Although the proposed paper focuses on mitigating inter-agent interference and promoting role differentiation, it does not appear to include comparisons with well-established role-based MARL methods such as ROMA[1] or RODE[2]. Since these approaches are explicitly designed to address role specialization and heterogeneous behaviors under shared or partially shared structures, they constitute highly relevant baselines for evaluating the proposed Complete Division mechanism. Including such comparisons would more clearly position the contribution relative to prior work and strengthen the empirical evidence that the proposed method offers advantages beyond existing role-based solutions.

[1] Wang, Tonghan, et al. "Roma: Multi-agent reinforcement learning with emergent roles." ICML 2020
[2] Wang, Tonghan, et al. "Rode: Learning roles to decompose multi-agent tasks." arXiv preprint arXiv:2010.01523 (2020).


**W2. Effectiveness on homogeneous agents**

While the proposed method demonstrates strong performance across several benchmarks, it appears that the advantage of Dspic is less pronounced in environments with largely homogeneous agent structures. For example, in the 6h_vs_8z scenario and terran_5_vs_5, where some degree of heterogeneity may arise due to random team composition. It would be helpful if the authors could provide further analysis on this aspect, as well as discuss whether any design considerations were explored to improve performance in more homogeneous cooperative settings.


**W3.  Limited novelty of the theoretical claims**

The paper provides substantial theoretical analysis to support the proposed approach. In particular, the authors introduce the notion of complete division and the equivalence of the resulting policy to a state-based policy. In addition, the paper presents theoretical arguments for training the diffusion-based policy. These analyses help improve the clarity of the proposed formulation and provide useful intuition for understanding the algorithm. However, the overall level of theoretical novelty appears somewhat limited, as the analysis largely serves to formalize and interpret the proposed design rather than introducing fundamentally new theoretical insights.

---

> ### Author Rebuttal · Authors · 2026-03-28
>
> We sincerely thank the reviewer for the positive feedback on our work's novelty, theoretical soundness, and the application of HARL. We hope our response can address your concerns.
> ### [W1] Comparison with ROMA and RODE
> We conducted comparative experiments against ROMA and RODE across four environments. The table below reports the mean rewards and std:
> ||Dspic (ours)|ROMA|RODE|
> |:---|:---:|---:|---:|
> |6h_vs_8z |**19.04±0.65**|15.00±1.44|18.82±0.28|
> |MMM2 |**19.51±0.20**|15.23±0.97|19.48±0.76|
> |terran_5_vs_5 |**15.06±1.63**|14.58±1.54|5.64±0.40|
> |LBF2p3f |**0.97±0.03**|0.59±0.03|0.47±0.23|
>
> The results show that **our approach achieves competitive or superior performance compared to ROMA and RODE across a wide range of tasks**. We appreciate this suggestion and will include these results in the revised version.
> ### [W2] Performance on 6h_vs_8z and terran_5_vs_5
> **For 6h_vs_8z**: MAPPO exhibits higher sample efficiency because **it uses parameter sharing without learning role representations (just taking ID as part of input)**, given that **it presumes the homogeneity of the environment (as a strong assumption)**. Because 6h_vs_8z is an environment of homogeneous agents, it naturally leads to higher learning efficiency than our approach. However, our approach is more generalizable to more diverse settings.
>
> **For terran_5_vs_5**: SMACv2 tasks present **significant challenges for policy-based methods**. For example, value-based methods like Kaleidoscope (built on QMIX) typically outperform policy-based methods (e.g., MAPPO, and our approach) in this game. Similarly, other previous work also supports this claim, e.g. BVME [1]. Notably, our approach achieves the best performance among policy-based algorithms, although we acknowledge that a gap remains to some value-based approaches.
>
> [1] Wei, Duan, et al. "Bandwidth-constrained Variational Message Encoding for Cooperative Multi-agent Reinforcement Learning" AAMAS 2026.
> ### [W3] Theoretical Novelty
> **We clarify that these theoretical claims are the foundation of discovering that policy deviation is caused by input overlap, and then guide the design of our complete division.** Specifically, in Proposition G.3 (Pages 26-27), our derivation of $M_{\pi}(s,a)$ reveals that the gradient of the shared objective contains a product of policies $\prod_{j=1}^n \pi(a^j|s)$. This leads to high-order polynomial terms $\pi^k(a|s)$ ($k=1,\dots,n$). Consequently, the gradient $M_{\pi}(s,a)$ still contains $\pi^{k-1}(a|s)$ terms, leading to a deviation of $M_{\pi}(s,a)$ relative to the heterogeneous case (or when agents' input spaces are disjoint) and thereby leading to policy deviation. Therefore, we propose complete division to solve this problem.
> ### [Q1] How Roles Affect Policies
> **Different roles generate different input spaces, guiding agents to produce different behaviors.** We illustrate this with a simple case in LBF(2p2f). Two players are given roles: $z_1=[1, 0, 0, 1, 0, 0, 0, 1, 1, 0]$ and $z_2=[0, 1, 1, 0, 1, 1, 1, 0, 0, 1]$. We set two food in $(0,3)$ and $(7,3)$. Even when both agents start at $(3,4)$ with identical observations, they are projected onto different matrices $\mathbf P_i$, resulting in $h_1\neq h_2$. When sampled via the diffusion policy, these distinct conditions lead to diverse actions: **agent 1 takes 1 (NORTH) while agent 2 takes 2 (SOUTH)**, corresponding to different targets.
> ### [Q2] Meaning of Role in Environments except for SMAC
> **In environments with less clear roles, our role embedding $z$ captures underlying transition semantics.** In MaMuJoCo, joints with similar physical dynamics naturally cluster close to each other. Although the agents have the same input and action space, they are still heterogeneous due to different dynamics. For Half-Cheetah-6x1, our visualization (see https://anonymous.4open.science/r/Dspic-4870/Vis_HC6x1.png) shows that the front/back thigh, shin, and foot joints form distinct clusters based on their dynamics.
> ### [Q3] Comparison of Training and Inference Costs
> **Our approach achieves a trade-off between efficiency and effectiveness**. We present a comparison of the computational overhead of our approach and HASAC in training and inference in all tested environments.
> ||HASAC Training Time|HASAC Inference Time|Dspic Training Time|Dspic Inference Time|
> |---|---|---|---|---|
> |MaMuJoCo|41.5min|281.7s|82.2min (1.98×)|347.8s (1.23×)|
> |SMAC|108.8min|1542.3s|140.5min (1.29×)|1670.2s (1.08×)|
> |SMACv2|95.5min|2846.0s|150.5min (1.58×)|3278.6s (1.15×)|
> |LBF|34.0min|266.2s|43.0min (1.26×)|317.1s (1.19×)|
> |Overall|62.7min|878.2s|99.3min (1.58×)|1000.1s (1.14×)|
>
> Note: Training or inference for 1M timesteps.

---

> > ### Author Rebuttal · Reviewer_3qUz · 2026-04-02
> >
> > Thank you for the detailed response. Most of my concerns have been resolved. I would like to maintain my decision towards acceptance.

---

> > > ### Author Response · Authors · 2026-04-02
> > >
> > > We are happy that all your concerns have been addressed. We thank the reviewer for the recommendation for acceptance.

---

### Decision · Program_Chairs · 2026-04-30

**Decision:**

Accept (regular)

**Comment:**

This work introduces Dspic, a novel approach to MARL that addresses the limitations of parameter sharing in heterogeneous settings. By utilizing "Complete Division" of the observation space and a diffusion-based policy, the method balances the trade-off between knowledge sharing and policy customization.

The Strengths of this work are listed as follows:

1. The authors address agent-level individuality from a novel and meaningful perspective under Complete Division.

2. The paper provides a well-structured and mathematically grounded justification of its main ideas, accompanied by formal proofs.

3. The experiments are quite detailed, and the empirical results demonstrate the superiority of the proposed method.

The major concerns of this work are:

1. The paper misses discussions with several recent parameter-sharing works in MARL, such as ADMN[1], GradPS[2].

2. The advantage of Dspic is less pronounced in environments with largely homogeneous agent structures, where its performance can be inferior to the MAPPO baseline.

3. The diffusion model adds a heavy computational burden, and the computational overhead introduced during testing needs consideration.

During the rebuttal phase, the authors proactively addressed most of the initial concerns. Although one reviewer gave a weak reject, they explicitly acknowledged in the post-rebuttal discussion that their specific concerns were resolved. Weighing the solid theoretical foundation, the strong empirical results, and the thorough rebuttal, the overall consensus leans positive. The paper is recommended for acceptance. For the camera-ready version, the authors are expected to incorporate the promised additional baseline/latency evaluations and ensure the missing related works are properly discussed.

[1] ADMN: Agent-Driven Modular Network for Dynamic Parameter Sharing in Cooperative Multi-Agent Reinforcement Learning. IJCAI 2024

[2] GradPS: Resolving Futile Neurons in Parameter Sharing Network for Multi-Agent Reinforcement Learning. ICML 2025